# Exploring Diverse Generation Paths via Inference-time Stiefel Activation Steering

**Dongxuan Zhu***
The Chinese University of Hong Kong
`dxzhu@se.cuhk.edu.hk`

**Ly Tran Ho Khanh***
The Chinese University of Hong Kong
`hokhanhlytran@cuhk.edu.hk`

**Andy Yat-Ming Cheung***
Hong Kong Polytechnic University
`andy.cheung@polyu.edu.hk`

**Man-Chung Yue**
Hong Kong Polytechnic University
`manchung.yue@polyu.edu.hk`

**Viet Anh Nguyen**
The Chinese University of Hong Kong
`nguyen@se.cuhk.edu.hk`
*These authors contributed equally.

## Abstract

Language models often default to a narrow set of high-probability outputs, leaving their generation paths homogeneous and prone to mode collapse. Sampling-based strategies inject randomness but still struggle to guarantee diversity across multiple concurrent generation runs. We address this limitation by introducing STARS (**ST**iefel-based **A**ctivation Steering for Diverse **R**ea**S**oning), a training-free, inference-time intervention method that transforms activation steering into an exploration engine. At each token, STARS collects the hidden activations of concurrent generation runs and optimizes multiple additive steering directions jointly on the Stiefel manifold. STARS maximizes the geometric volume of the steered activations, while the Stiefel manifold induces orthogonality of the steering interventions. This formulation explicitly promotes divergent activation vectors of concurrent generation runs, and implicitly promotes divergent generation trajectories. This manifold optimization formulation can be solved using a Riemannian gradient descent algorithm with convergence guarantees, but this algorithm is too time-consuming for real-time inference. To guarantee low latency, we further design a lightweight one-step update with an aggressive, closed-form stepsize. For test case generation and scientific discovery benchmarks, STARS consistently outperforms standard sampling methods, achieving greater diversity without sacrificing qualitative performance.

## 1 Introduction

Language models (LMs) have achieved impressive performance across a broad spectrum of tasks (Achiam et al., 2023; Nam et al., 2024; Shao et al., 2024; Yang et al., 2025a). A powerful emerging paradigm reframes complex problems as a search problem: an LM generates a diverse pool of candidate solutions, and a selector then identifies the best one. This best-of-$N$ strategy has driven significant gains in reasoning, coding, and planning (Lightman et al., 2023; Ni et al., 2023). However, the diversity of the candidate pool fundamentally bottlenecks its efficacy. When multiple runs, despite stochastic sampling, converge on the same high-probability line of thought, they follow nearly identical latent trajectories. The selector is left with paraphrases, unable to uncover a better solution, and performance saturates regardless of how much the sampling budget is increased.

The consequences of this diversity collapse are profound and far-reaching. For reasoning tasks, maintaining distinct trajectories is crucial for exploring the solution space and avoiding premature convergence on a suboptimal line of argument. In mathematical reasoning, a diversity collapse occurs when a model becomes trapped in a single plausible but flawed line of argument, unable to

backtrack and explore alternative proof strategies. For open-ended applications, such as scientific discovery or creative writing, diverse outputs are not just a means to an end, but are intrinsically valuable, offering genuinely different perspectives rather than mere paraphrases of a single idea. For safety and alignment, a lack of diversity prevents us from discovering varied failure modes, adversarial attacks, or unintended model behaviors during red-teaming. Therefore, increasing generation diversity is not an incremental improvement but a fundamental step toward unlocking the full reasoning and creative potential of LMs.

Existing methods for inducing diversity primarily perturb the token-level sampling process. Techniques such as temperature sampling (Ackley et al., 1985), nucleus sampling (Holtzman et al., 2020), beam search (Xie et al., 2023), and self-speculative decoding (Naik et al., 2024) adjust token probabilities during decoding. However, these decoders operate locally and myopically; they are uncoordinated across parallel runs and lack a global objective for diversity. This often leads to *diversity collapse*, a failure mode where superficially distinct sequences still follow nearly identical underlying chains of thought (Yun et al., 2025; Dang et al., 2025). Other approaches promote diversity during training by modifying the learning objective or data, often with reinforcement learning (e.g., Slocum et al., 2025; Chen et al., 2025a; Lanchantin et al., 2025; Chung et al., 2025). While promising, these methods demand access to the full training pipeline, impose substantial computational costs, and their benefits can be fragile across domains (Ling et al., 2025). Consequently, enhancing generation diversity at inference time, without retraining, remains a critical and practical goal.

Recent advances in activation steering have opened new avenues for controlling LMs' behavior during inference (Turner et al., 2024; van der Weij et al., 2024). Activation vectors represent the model's working memory and thought space, in which it assembles features, subgoals, and circuits over time. If multiple runs occupy nearly the same region in this space, surface-level stochasticity has little impact (Wang et al., 2025a; Troshin et al., 2025). This phenomenon suggests a more structural intervention: actively diversifying the hidden trajectories themselves. Prior work on activation steering has shown that carefully crafted directions can control attributes or concepts without retraining (Subramani et al., 2022; Turner et al., 2024; Wang et al., 2025b). Recent work has investigated adaptive activation-steering techniques, including methods that adjust the steering-strength parameter (Rodriguez et al., 2025a;b) and approaches that optimize the position of the steering token (Hedström et al., 2025). However, existing steering techniques push a single generation run toward a fixed, predefined direction, making them ill-suited for exploration.

A major technical obstacle to inference-time diversification is reconciling this principled objective with practical constraints. An effective inference-time intervention must be lightweight enough to run at every token without prohibitive latency, yet powerful enough to produce meaningful diversity. Furthermore, steering must preserve fluency and factuality, as naively aggressive diversification can degrade generation quality, producing meaningless or irrelevant text. This paper introduces a novel approach that repurposes activation steering, originally designed for targeted control, into a dynamic engine for exploration and diversity. Instead of steering a single run toward a fixed concept, we steer multiple concurrent runs away from each other. Our training-free approach intervenes at each token to push the models' internal states into distinct regions of the activation space, thereby promoting divergent and structurally distinct reasoning paths.

**Contributions.** We summarize our contributions as follows:

- We propose STARS (**St**iefel-based **A**ctivation Steering for Diverse **R**ea**S**oning), a framework that steers LM activations at inference time to diversify reasoning trajectories. At each token, STARS extracts the hidden activations from all sequences, then computes the orthogonal steering directions that maximize the total volume spanned by the intervened activations.

- In Section 4, we cast the volume-maximization objective with orthogonality constraints as a Riemannian optimization problem on the Stiefel manifold. We then propose a Riemannian gradient descent with a theoretical convergence guarantee.

- In Section 5, we develop a practical and lightweight one-step update algorithm. By pairing a specific initialization scheme with the activation matrix as a computationally-free search direction, we replace the intractable exact line search with a well-posed quadratic approximation. This enables us to derive an aggressive, closed-form step size, guaranteeing low latency and allowing steering at every token during inference time.

**Notations.** We use $I$ for the identity matrix of appropriate size. We use $\nabla \ell(V)$ for the Euclidean gradient of the function $\ell$ while $\operatorname{grad} \ell(V)$ is for the Riemannian gradient. For a matrix $A \in \mathbb{R}^{d \times N}$, we use $\|A\|_F \triangleq \sqrt{\operatorname{Tr}[A^\top A]}$ for the Frobenius norm. The spectral norm $\|A\|_2$ is defined as the largest singular value of $A$. We use $\mathbb{R}_{++} \triangleq \{x \in \mathbb{R} : x > 0\}$ for the set of all strictly positive reals.

## 2 RELATED WORKS

**Diversity of LM Generation.** Recent studies underscore the fragility of diversity in LM-generated outputs, with mode collapse increasingly recognized as a recurrent failure mode wherein responses become homogenized over time or across inputs (Shumailov et al., 2024). Empirical evidence indicates that LMs frequently produce similar responses even when the prompts are systematically varied along demographic dimensions designed to generate distinct perspectives (Park et al., 2024). In response, test-time scaling methods have been proposed to enhance output diversity, thereby improving downstream performance. A prevalent approach entails sampling multiple independent candidate solutions, often via Chain-of-Thought prompting and its variants, and subsequently applying selection mechanisms to identify the highest-quality response (Wei et al., 2022; Zhang et al., 2022; Wang & Zhou, 2024). These candidates are filtered or ranked using reward models or heuristic criteria policies such as Best-of-$N$ sampling or beam search, to extract superior outputs from a diverse candidate set (Vijayakumar et al., 2016; Lightman et al., 2023). Complementary prompting-based techniques explicitly incentivize variation in reasoning styles or perspectives, including persona-based prompting (Cheng et al., 2023) and step-by-step recall prompting (Hayati et al., 2023). In parallel, Chung et al. (2025); Ismayilzada et al. (2025); Deshpande et al. (2025) investigate post-training interventions aimed at fostering output diversity.

**Activation Steering** directly intervenes the model's behavior by extracting steering vectors within specific layers of the LM's transformer architecture (Panickssery et al., 2023; Turner et al., 2024; Wang et al., 2025b; Chen et al., 2025b). Typical steering methods are for target behavior or purpose, where steering vectors can be computed directly or contrastively from positive and negative examples. For example, to align model behavior with human preferences, Panickssery et al. (2023) averages the differential residual-stream activations between matched pairs of positively and negatively labeled exemplars of a target behavior. Stolfo et al. (2025), instead, compute the steering vectors as the difference in activations between inputs with and without instruction to guide models to follow constraints even without explicit instructions. Instead of adding the steering vector to the residual stream, Postmus & Abreu (2024) projects the activations to ellipsoidal regions by a conceptor matrix, which captures the principal directions and variances of a set of neural activation vectors. In addition, some works employ sparse autoencoders to identify steering directions from sparse latent representations (O'Brien et al., 2024; He et al., 2025; Yang et al., 2025b).

While a recent paper Khanh et al. (2025) also focuses on inference-time activation steering for diverse generations, this paper differs in three ways. First, this paper enforces orthogonality and formulates volume maximization on the (scaled) Stiefel manifold, whereas Khanh et al. (2025) maximizes activation volume on a product of spheres. Second, this paper performs token-level steering by recomputing directions at each token position, while Khanh et al. (2025) updates only at synchronization anchors. Third, to make token-level steering practical, this paper develops a lightweight one-step update with a closed-form stepsize, thereby creating an algorithmically distinct pathway from the iterative Riemannian optimization routines emphasized in Khanh et al. (2025).

## 3 STIEFEL ACTIVATION STEERING FRAMEWORK FOR INFERENCE-TIME DIVERSE GENERATION

In this section, we present STARS, our proposed framework illustrated in Figure 1. The key idea is to promote diverse generation paths from a language model during inference. Given a query, we generate $N$ output sequences in parallel and extract their final hidden-state vectors from a specified layer. For each path, we compute an additive steering vector and apply it to the corresponding hidden state, resulting in a new set of steered representations that serve as input to the next layer.

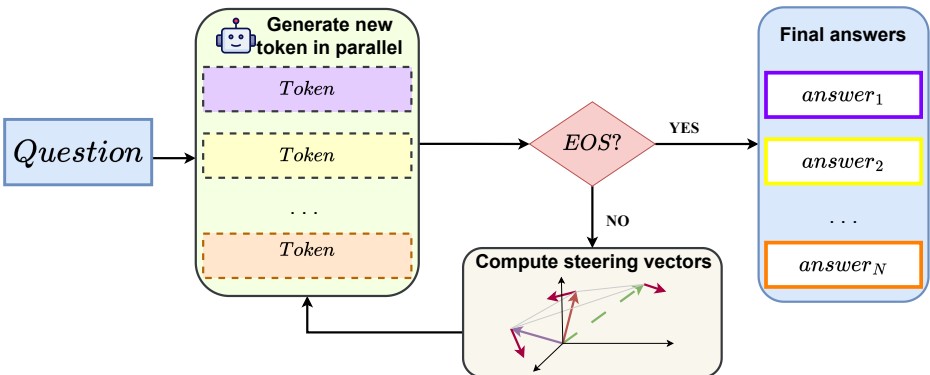

Figure 1: Schematic overview of our STARS framework. Given an input question, the model generates $N$ candidate sequences in parallel. At each decoding step, hidden states from a designated layer are extracted and used to recompute the steering vectors, which then guide the generation of the next token. This iterative process continues until the end-of-sequence (EOS) token is produced. Paths that emit EOS are dropped from the active set.

## 3.1 Attention-Based Steering Intervention

Here, we rigorously introduce where the activation steering is applied in transformer-based LMs. Transformer architectures (Vaswani et al., 2017) have become the foundation of modern LMs, achieving remarkable performance across diverse natural language processing tasks. We consider an LM with $M$ heads per layer, exhibiting the following per-block information flow for the layer $l$:

$$x^{(l+1)} = \text{FFN}(\text{MHA}(x^{(l)})) = \text{FFN}\Big( \bigoplus_{j=1}^{M} W_j^o \big(\text{Attn}_j(x^{(l)})\big) \Big),$$

where $x^{(l)}$ is the input of the specific layer $l$, with $W_j^o$ denoting the output projection matrix and $\text{Attn}_j$ denoting the single-head attention transformation for each head $j = 1, \ldots, M$. Here, FFN, MHA denote the feed-forward and multi-head attention, respectively. The direct sum operator $\bigoplus$ concatenates the outputs from all attention heads before applying the feed-forward transformation. The multi-head attention mechanism is a compelling target for activation steering because its individual heads are functionally specialized. As demonstrated by Clark et al. (2019), specific heads learn to model interpretable linguistic phenomena, such as distinct syntactic dependencies and coreference relations. Following (Li et al., 2023; Chen et al., 2024; Ma et al., 2025), we add the steering vector at the output of each attention head, giving the following edited information flow:

$$x^{(l+1)} = \text{FFN}(\text{MHA}^e(x^{(l)})) = \text{FFN}\left( \bigoplus_{j=1}^{M} W_j^o \left( \text{Attn}_j(x^{(l)}) + v^{(l,j)} \right) \right), \quad (1)$$

where $v^{(l,j)}$ is the steering vector for head $j$ at layer $l$. For notational clarity, we denote the unmodified head output as $h^{(l,j)} = \text{Attn}_j(x^{(l)})$. Each $h^{(l,j)} \in \mathbb{R}^{d_h}$ lives in the head dimension $d_h$, and concatenating across all $M$ heads yields $h^{(l)} = [h^{(l,1)}; \ldots; h^{(l,M)}] \in \mathbb{R}^d$ with $d = d_h \times M$. Similarly, the steering vectors are stacked as $v^{(l)} = [v^{(l,1)}; \ldots; v^{(l,M)}] \in \mathbb{R}^d$.

## 3.2 Orthogonal Steering Framework

At inference, we exploit the mechanism (1) to diversify decoding trajectories by steering at each token. Concretely, we generate $N$ response trajectories and, at each token position $\tau$, collect their hidden states $h_{\tau,1}^{(l)}, \ldots, h_{\tau,N}^{(l)} \in \mathbb{R}^d$ from the layer $l$. For each trajectory, we compute a steering vector $v_{\tau,i}^{(l)}$ and form modified hidden states $h_{\tau,1}^{(l)} + v_{\tau,1}^{(l)}, \ldots, h_{\tau,N}^{(l)} + v_{\tau,N}^{(l)}$, which are then propagated through subsequent layers along their respective paths. We focus on a pre-determined layer $l$ and hence omit the superscript $l$ in what follows; we also omit the token index $\tau$ for notational simplicity.

The central idea of STARS is to maximize the geometric notion of dispersion for the set $\{h_i+v_i\}_{i=1}^N$ in the activation space. A canonical dispersion metric is the volume of the parallelepiped spanned by these vectors. Defining $H = [h_1,\ldots,h_N] \in \mathbb{R}^{d\times N}$ and $V = [v_1,\ldots,v_N] \in \mathbb{R}^{d\times N}$, the volume of the parallelotope by $\{h_i + v_i\}_{i=1}^N$ in the activation space is equal to the square root of the determinant of $(H+V)^\top(H+V)$ (Pyle, 1962). To encourage diversity between different generations, we require the steering vectors to be orthogonal with each other, i.e., $v_i^\top v_j = 0$ when $i \neq j$. The orthogonal constraint regularizes the steering vectors $v_i$, ensuring each run receives a distinct direction and preventing the trivial solution where all $v_i$ align. We further use a user-set magnitude hyperparameter $\alpha > 0$ to control the magnitude of $v_i$, that is, $\|v_i\|_2^2 = \alpha$ for all $i$. A too big $\alpha$ may break the meaningful information in $h_i$, leading to generation collapse. The optimization problem for computing steer vectors is:

$$\min_{V^\top V=\alpha I} \ell(V) \triangleq -\log\det((H+V)^\top(H+V)). \tag{2}$$

For problem (2) to be meaningful, there must exist at least one feasible $V$ with a finite objective value. Since the domain of $-\log\det$ is the set of non-singular matrices, this requires the existence of a $V$ satisfying $V^\top V = \alpha I$ and $(H+V)^\top(H+V)$ is positive definite. The following proposition confirms this, whose proof is constructive and yields a practical initialization for problem (2).

**Proposition 1** (Existence of full-rank solution). *Let $H \in \mathbb{R}^{d\times N}$ be any matrix with rank $r \leq d-N$. Then there exists $V \in \mathbb{R}^{d\times N}$ with $V^\top V = \alpha I$ such that $(H+V)^\top(H+V)$ is positive definite.*

The assumption $d \geq r + N$ is mild in our setting. In practice, the activation dimension $d$ ranges from $1536$ (e.g., QWEN-2.5-1.5B) up to $2^{14}$ (e.g., LLAMA-3.1-405B), whereas the number of responses $N$ is usually small (e.g., 4, 8, or 16). Consequently, $d$ substantially exceeds $r + N$ under typical configurations since $r \leq N$. Proposition 1 also gives us an initialization for $V$, described in Algorithm 1. The computational cost of Algorithm 1 is dominated by the singular value decomposition (SVD) of the matrix $H$. A classical method, such as the Golub–Reinsch algorithm, can compute SVD in $O(4d^2N + 8dN^2 + 9N^3)$ (Golub & van Loan, 2013, Section 8.6.4).

---

**Algorithm 1** Initialization for $V$

---

**Require:** Incumbent activation matrix $H \in \mathbb{R}^{d\times N}$, magnitude $\alpha \in \mathbb{R}_{++}$
 1: Compute the SVD of $H$ as $H = Q\Sigma W^\top$ and $r = \mathrm{rank}(H)$, where $Q \in \mathbb{R}^{d\times d}$, $\Sigma = \mathrm{diag}(\sigma_1,\ldots,\sigma_r,0,\ldots,0) \in \mathbb{R}^{d\times N}$ and $W \in \mathbb{R}^{N\times N}$.
 2: Set $Q_2$ as the last $d - r$ columns of $Q$.
 3: Randomly select $N$ columns $[v_1,\ldots,v_N]$ of $Q_2$.
 4: **return** $V_0 = \sqrt{\alpha}[v_1,\ldots,v_N], \Sigma, W$.

---

## 4   RIEMANNIAN GRADIENT DESCENT OVER STIEFEL MANIFOLD

In this section, we present a Riemannian optimization approach for finding steering vectors based on problem (2). The feasible region of problem (2) is the scaled Stiefel manifold in $\mathbb{R}^{d\times N}$ with magnitude parameter $\alpha > 0$:

$$\mathrm{St}(d, N, \alpha) = \{V \in \mathbb{R}^{d\times N} : V^\top V = \alpha I\}.$$

Since $\mathrm{St}(d, N, \alpha)$ is nonconvex, the classical projected gradient descent theory for convex sets does not apply here. A naive scheme that takes a step along the Euclidean gradient of $\ell(V)$ and then projects back onto the constraint set may actually increase the objective. Even local convergence can be difficult to guarantee, since the projection may distort the step direction enough to slow or stall progress unless the step size is chosen extremely small. To circumvent these issues, we adopt a Riemannian optimization approach to problem (2), thereby exploiting the manifold's differential-geometric structure to design efficient algorithms.

Riemannian gradient descent resembles projected gradient descent in the Euclidean setting, and each iteration consists of two main steps: first, computing a descent direction in the tangent space and choosing a suitable step size to generate a tangent vector; second, retracting this tangent vector back

onto the manifold. At iterate $k$ with incumbent solution $V_k \in \text{St}(d, N, \alpha)$, a descent direction $S_k$ is computed in the tangent space at $V_k$. The tangent space at point $V_k \in \text{St}(d, N, \alpha)$ is

$$T_{V_k}\text{St}(d, N, \alpha) = \{U \in \mathbb{R}^{d \times N} : V_k^\top U + U^\top V_k = 0\}.$$

A tangent space is a subspace of the Euclidean space $\mathbb{R}^{d \times N}$, consisting of tangent vectors that represent the directions of feasible curves passing through the point and remaining on the manifold. We equip the tangent space with the Frobenius inner product $\langle U_1, U_2 \rangle = \text{Tr}[U_1^\top U_2]$ for $U_1, U_2 \in T_{V_k}\text{St}(d, N, \alpha)$. To obtain a descent direction $S_k$, one can first compute the Euclidean gradient $\nabla\ell(V_k) \triangleq -2(H + V_k)[(H + V_k)^\top(H + V_k)]^{-1}$, and then project it onto the tangent space $T_{V_k}\text{St}(d, N, \alpha)$ via the orthogonal projection $\text{Proj}_{T_{V_k}}(U) \triangleq U - V_k \frac{V_k^\top U + U^\top V_k}{2\alpha}$ for $U \in \mathbb{R}^{d \times N}$, which gives the Riemannian gradient $\text{grad}\,\ell(V_k) = \text{Proj}_{T_{V_k}}(\nabla\ell(V_k))$. The negative Riemannian gradient $S_k \triangleq -\text{grad}\,\ell(V_k)$ serves as our descent direction. Indeed, any direction $S_k \in T_{V_k}\text{St}(d, N, \alpha)$ with $\langle S_k, \text{grad}\,\ell(V_k) \rangle < 0$ is also a descent direction at point $V_k$.

The second step is to update the iterate $V_k$ by moving along the direction $S_k$ with a stepsize $\eta > 0$. Because a standard Euclidean update $V_k + \eta S_k$ takes the solution off the manifold, we will use a *retraction*, an operation that generalizes the concept of a straight-line step to a curved space. It translates the tangent vector $\eta S_k$ into a point on the manifold, ensuring the new iterate $V_{k+1}$ remains feasible. We employ the polar decomposition-based retraction

$$\text{R}_V(U) \triangleq \sqrt{\alpha}(V + U)(\alpha I + U^\top U)^{-1/2}. \tag{3}$$

The full update step is $V_{k+1} = \text{R}_{V_k}(\eta S_k)$, where the stepsize $\eta > 0$ is determined using a backtracking line search to ensure a sufficient decrease in the objective function. Ensuring sufficient decrease also prevents $H + V_k$ from becoming rank-deficient, which could render $\ell(V_k)$ ill-posed. We now summarize the complete procedure as Algorithm 2.

---

**Algorithm 2** Riemannian gradient descent with line-search

---

**Require:** Hidden activation matrix $H \in \mathbb{R}^{d \times N}$, max iterations $K$, magnitude $\alpha \in \mathbb{R}_{++}$, line search parameters $\rho \in (0, 1), c \in (0, 1), \bar{\eta} \in \mathbb{R}_{++}$
1: Initialize $V_0$ via Algorithm 1
2: **for** $k = 0$ to $K$ **do**
3:     Compute the Euclidean gradient $\nabla\ell(V_k) \leftarrow -2(H + V_k)[(H + V_k)^\top(H + V_k)]^{-1}$
4:     Compute the Riemannian direction $S_k \leftarrow -\nabla\ell(V_k) + V_k \frac{V_k^\top \nabla\ell(V_k) + (\nabla\ell(V_k))^\top V_k}{2\alpha}$
5:     Initialize $\eta \leftarrow \bar{\eta}$
6:     **while** True **do**
7:         Compute $V_{k+1} \leftarrow \sqrt{\alpha}(V_k + \eta S_k)(\alpha I + \eta^2 S_k^\top S_k)^{-1/2}$
8:         **if** $\ell(V_k) - \ell(V_{k+1}) \geq c\eta\|S_k\|_F^2$ **then break else** $\eta \leftarrow \rho\eta$ **end if**
9:     **end while**
10: **end for**
11: **return** $V_K$

---

Since we require a decrease in function value during each step by line search in Algorithm 2 and initialize $V_0$ to make $H + V_0$ full rank, the iteration sequence $\{V_k\}$ falls into the compact sublevel set $\mathcal{C} \triangleq \{V \in \text{St}(d, N, \alpha) : \ell(V) \leq \ell(V_0)\}$, which is the intersection of a sublevel set and the manifold. Let $\underline{\sigma}$ define the smallest singular value of $H + V$ over $\mathcal{C}$. Since $H + V$ is always full-rank for all $V \in \mathcal{C}$ and $\mathcal{C}$ is a compact set, we can conclude that $\underline{\sigma}$ is bounded away from 0, i.e., $\underline{\sigma} > 0$. Furthermore, we define $\bar{D} \triangleq \max_{V \in \mathcal{C}} \|\nabla\ell(V)\|_F \leq 2\frac{\sqrt{N}}{\underline{\sigma}^2}(\|H\|_2 + \sqrt{\alpha}) < \infty$, which is an upper bound on $\|\nabla\ell(V)\|_F$ over the sublevel set $\mathcal{C}$.

The following theorem shows the convergence guarantee of Algorithm 2:

**Theorem 1** (Convergence Rate of Algorithm 2). *Let $\{V_k\}_{k \geq 1}$ be the sequence generated from Algorithm 2 with $\rho, c \in (0, 1)$ and $0 < \bar{\eta} < \sqrt{\alpha}/\bar{D}$. Then for all $K \geq 1$ we have*

$$\min_{0 \leq k \leq K} \|\text{grad}\,\ell(V_k)\|_F^2 = O(\tfrac{1}{K}).$$

On a Riemannian manifold, $V$ is a stationary point of a smooth function $\ell$ if and only if its Riemannian gradient vanishes, that is, $\mathrm{grad}\,\ell(V) = 0$ (Proposition 4.6 Boumal, 2023). Theorem 1 therefore shows that Algorithm 2 converges to the critical point of problem (2) at rate $O(1/K)$. Despite the favorable theoretical guarantee, Algorithm 2 has multiple practical drawbacks. Firstly, Theorem 1 indicates that the initial stepsize $\bar{\eta}$ depends on the quantity $\bar{D}$, which is challenging to estimate. Secondly, each iteration is expensive because it requires computing matrix inverses, square roots, and inverse square roots, each with complexity $O(N^3)$. Further, if the matrices $H + V_k$ are ill-conditioned, the repeated inversion and square-root computations can be numerically unstable. Additionally, the backtracking line search requires multiple evaluations of the log-determinant objective. When the stepsize shrinks excessively, progress becomes very slow, and poorly chosen parameters can lead to significant computational waste. We will explore a more computationally friendly procedure in the next section.

## 5    ONE-STEP UPDATE WITH GREEDY STEPSIZE

In this section, we propose an efficient method for finding a suitable set of steering vectors. Instead of an iterative algorithm, we employ a one-step update with a carefully designed search direction and a greedy stepsize. For a given point $V \in \mathrm{St}(d, N, \alpha)$ and a search-direction $S \in T_V\mathrm{St}(d, N, \alpha)$, the exact line search determines the stepsize by minimizing the following objective function

$$\min_{\eta \geq 0} \ \mathcal{L}(\eta) \triangleq -\log\det[(H + \mathrm{R}_V(\eta S))^\top (H + \mathrm{R}_V(\eta S))], \tag{4}$$

where $\mathrm{R}_V(\cdot)$ is the polar retraction defined in (3). As solving this problem is intractable due to the nonlinearities in the log-determinant and the retraction, we instead work with a quadratic approximation of $\mathcal{L}(\eta)$.

**Proposition 2** (Quadratic approximation of $\mathcal{L}(\eta)$). *For any activation $H$, steering vectors $V$ and search direction $S$, we define $A = (H + V)^\top (H + V)$, $D_1 = \mathrm{Tr}[A^{-1}(H^\top S + S^\top H)]$ and $D_2 = \frac{1}{\alpha}\mathrm{Tr}[A^{-1}(H^\top V S^\top S + S^\top S V^\top H)] + \mathrm{Tr}[((A^{-1}(H^\top S + S^\top H))^2]$. The function $\mathcal{L}(\eta)$ has the following second-order expansion:*

$$\mathcal{L}(\eta) = \frac{1}{2}D_2\eta^2 - D_1\eta - \log\det A + O(\eta^3).$$

Neglecting the higher-order term $O(\eta^3)$, we consider the quadratically approximated stepsize:

$$\eta^\star \triangleq \arg\min_{\eta \geq 0} \ \frac{1}{2}D_2\eta^2 - D_1\eta. \tag{5}$$

For the minimization problem (5) to be well-posed and yield a finite stepsize, $D_2$ must be positive; otherwise, the approximated objective would be unbounded below as $\eta \to \infty$. When $D_2 > 0$, the optimal stepsize is given by $\eta^\star = \max(0, D_1/D_2)$. To ensure a meaningful update ($0 < \eta^\star < \infty$), we choose a specific combination of $V_0$ and a search direction $S$ as follows: (i) We choose $V_0$ from the output of Algorithm 1; (ii) We choose the search direction as $S = H$.

Next, we show that $H$ is a descent direction in the Riemannian sense at the point $V_0$ chosen above.

**Proposition 3** (Descent direction). *Let $H \in \mathbb{R}^{d \times N}$ and let $V_0$ be given from Algorithm 1. Then $H$ is a descent direction at point $V_0$ in the Riemannian sense, i.e., $\langle H, \mathrm{grad}\,\ell(V_0)\rangle < 0$.*

We now proceed to demonstrate that the above choice of $(V_0, S)$ yields a meaningful stepsize $\eta^\star$. Moreover, we can compute $\eta^\star$ in an analytical form of the singular values of $H$.

**Proposition 4** (Positivesness of $\eta^\star$). *Let $H \in \mathbb{R}^{d \times N}$ and let $V_0$ be given from Algorithm 1. Let $r$ be the rank of $H$ and $\sigma_1, \ldots, \sigma_r$ be the singular values of $H$. Then we have*

$$D_1 = 2\sum_{i=1}^{r} \frac{\sigma_i^2}{\sigma_i^2 + \alpha} > 0 \quad and \quad D_2 = 4\sum_{i=1}^{r} \frac{\sigma_i^4}{(\sigma_i^2 + \alpha)^2} > 0.$$

*In this case, the optimal stepsize is $\eta^\star = \frac{D_1}{D_2} > 0$.*

Now we are ready to propose Algorithm 3, a heuristic one-step update procedure for solving problem (2). Given the activation matrix $H$, Line 1 of the algorithm first calls Algorithm 1 to obtain a feasible initialization $V_0$ along with the SVD of $H$. Then Line 2 computes the stepsize $\eta^\star$ using the singular values of $H$. Finally, Lines 3 implements the polar retraction (3), where the inverse matrix square root is efficiently computed via the SVD of $H$. Algorithm 3 is a one-step update, and it is hopeless to establish any theoretical convergence guarantees for it. However, Appendix B.7 presents extensive empirical results to show that Algorithm 3 achieves the optimality gap of approximately $2\%$ in a single update, while using only about $3\%$ of the run time of Algorithm 2.

---

**Algorithm 3** One-step update with greedy stepsize

---

**Require:** Incumbent activation matrix $H \in \mathbb{R}^{d \times N}$ of rank $r$, magnitude $\alpha \in \mathbb{R}_{++}$
  1: Compute $V_0$, $\Sigma$ and $W$ via Algorithm 1
  2: Compute $D_1 \leftarrow 2 \sum_{i=1}^{r} \frac{\sigma_i^2}{\sigma_i^2 + \alpha}$, $D_2 \leftarrow 4 \sum_{i=1}^{r} [\frac{\sigma_i^2}{\sigma_i^2 + \alpha}]^2$ and $\eta^\star \leftarrow D_1/D_2$
  3: Compute $V_1 \leftarrow \sqrt{\alpha}(V_0 + \eta^\star H)W(\alpha I + (\eta^\star \Sigma)^2)^{-1/2}W^\top$
  4: **return** $V_1$

---

## 6 NUMERICAL EXPERIMENTS

We evaluate our framework STARS on two benchmarks that require generating diverse outputs: the test case generation benchmark TESTEVAL (Wang et al., 2024) and the scientific discovery benchmark LiveIdeaBench (Ruan et al., 2025), allowing us to demonstrate the effectiveness of our approach across different domains that benefit from output diversity. For TESTEVAL, we use $N = 20$ as in the original benchmark (Wang et al., 2024); for LiveIdeaBench, we set $N = 4$. The hyperparameter $\alpha$ in Algorithm 3 is set to $\alpha = C \cdot \|H\|_2^2$ where $C > 0$ is a scaling constant and $\|H\|_2$ is the spectral norm of matrix $H$. We denote configurations by STARS_C to indicate the choice of $C$. All the used prompts are presented in Appendix B. The details of steering layer selection is in Appendix B.2.

### 6.1 TEST CASE GENERATION

Following the TESTEVAL benchmark (Wang et al., 2024), we evaluate the generation of test cases on the overall coverage task. After generation, all test cases must undergo a correctness check. Syntactic correctness determines if the generated test case is free of syntax errors, while execution correctness evaluates if the test case can be executed successfully without any runtime errors. Assertion correctness evaluates whether the generated test case contains correct test assertions. The overall coverage for a program is computed by the proportion of lines/branches in the program that have been covered by at least one test case, while line/branch $cov@k$ measures the line/branch coverage with a subset of the generated test cases of size $k$. The coverage metrics are batch-level by construction: for each program, we consider the union of coverage over the $N$ generated tests and compute the fraction of lines/branches covered by at least one test. The correctness metrics are computed per-test but averaged over the full set of generated tests.

Table 1 demonstrates that the STARS method consistently outperforms the temperature sampling approach (labeled as SAMPLING) across all temperature settings and evaluation metrics on both the GEMMA-1.5-2B-IT and QWEN3-1.7B models. STARS_0.5 (STARS run with $C = 0.5$) achieves particularly strong performance, yielding substantial improvements in overall coverage as well as line- and branch-level coverage at both $k = 1$ and $k = 5$. The trend is consistent across both model families. Notably, on QWEN3-1.7B, STARS improves execution correctness by more than sampling at all temperatures, while also delivering strong coverage gains. Overall, the results indicate that STARS provides a substantially better balance between diversity and functional correctness than temperature sampling, with STARS_0.5 emerging as the most effective configuration.

### 6.2 SCIENTIFIC DISCOVERY

We assess LMs' idea-generation capabilities using the LiveIdeaBench dataset, which contains 118 keywords. Following Guilford's creativity theory (Ruan et al., 2025), our methodology evalu-

Table 1: Experimental results on TESTEVAL across different models and sampling temperatures. All the numbers are in percentages. Best-performing methods are highlighted in bold.

| | T | Method | Correctness ↑ | | | Overall cov. ↑ | | Line $cov@k$ ↑ | | Branch $cov@k$ ↑ | |
|---|---|---|---|---|---|---|---|---|---|---|---|
| | | | syntax | exe | assert | line | branch | $k=1$ | $k=5$ | $k=1$ | $k=5$ |
| GEMMA-1.1-2B-IT | 1.0 | SAMPLING | **89.90** | 1.02 | 0.95 | 5.36 | 5.06 | 5.32 | 5.35 | 4.98 | 5.04 |
| | | STARS_0.1 | 86.48 | 1.43 | 1.17 | 10.03 | 9.32 | 9.97 | 10.02 | 9.12 | 9.25 |
| | | STARS_0.5 | 79.76 | **3.83** | **3.45** | **34.76** | **30.93** | **33.81** | **34.45** | **29.58** | **30.40** |
| | 0.8 | SAMPLING | **91.79** | 1.19 | 0.00 | 3.54 | 3.37 | 3.46 | 3.51 | 3.21 | 3.30 |
| | | STARS_0.1 | 87.83 | 1.45 | 1.12 | 10.77 | 9.88 | 10.64 | 10.72 | 9.65 | 9.78 |
| | | STARS_0.5 | 79.43 | **3.79** | **3.45** | **32.96** | **30.02** | **31.70** | **32.59** | **28.23** | **29.43** |
| | 0.6 | SAMPLING | **93.45** | 1.29 | 0.00 | 3.01 | 2.84 | 2.94 | 2.99 | 2.70 | 2.77 |
| | | STARS_0.1 | 89.19 | 1.71 | 1.43 | 12.10 | 11.22 | 12.03 | 12.10 | 10.99 | 11.15 |
| | | STARS_0.5 | 79.24 | **3.86** | **3.24** | **34.07** | **30.38** | **32.04** | **33.22** | **28.00** | **29.46** |
| | 0.4 | SAMPLING | **95.40** | 1.36 | 1.26 | 2.33 | 2.31 | 2.29 | 2.32 | 2.20 | 2.27 |
| | | STARS_0.1 | 90.43 | 1.48 | 1.43 | 8.44 | 7.69 | 8.41 | 8.44 | 7.58 | 7.63 |
| | | STARS_0.5 | 80.31 | **4.38** | **4.00** | **40.03** | **35.96** | **38.39** | **39.36** | **33.73** | **35.07** |
| | 0.2 | SAMPLING | **95.64** | 1.36 | 1.36 | 1.44 | 1.41 | 1.37 | 1.42 | 1.27 | 1.35 |
| | | STARS_0.1 | 91.52 | 1.45 | 1.36 | 7.28 | 6.50 | 7.23 | 7.25 | 6.36 | 6.40 |
| | | STARS_0.5 | 78.93 | **4.38** | **4.02** | **39.03** | **35.05** | **37.38** | **38.67** | **32.94** | **34.39** |
| QWEN3-1.7B | 1.0 | SAMPLING | 18.07 | 11.88 | 4.19 | 59.95 | 55.05 | 57.18 | 58.66 | 51.31 | 53.39 |
| | | STARS_0.1 | 61.10 | 39.90 | 19.17 | 86.77 | 81.10 | 80.54 | 83.62 | 71.71 | 76.36 |
| | | STARS_0.5 | **72.21** | **41.86** | **30.79** | **91.05** | **86.81** | **82.18** | **87.31** | **73.59** | **81.02** |
| | 0.8 | SAMPLING | 13.86 | 8.17 | 2.83 | 37.06 | 33.73 | 36.15 | 36.47 | 32.43 | 33.04 |
| | | STARS_0.1 | 61.10 | 39.90 | 19.17 | 86.77 | 81.10 | 80.54 | 83.62 | 71.71 | 76.36 |
| | | STARS_0.5 | **71.55** | **41.31** | **29.07** | **92.22** | **87.86** | **83.00** | **88.42** | **74.39** | **82.14** |
| | 0.6 | SAMPLING | 11.69 | 6.19 | 2.26 | 23.92 | 21.79 | 22.76 | 23.24 | 20.35 | 20.89 |
| | | STARS_0.1 | 61.57 | 41.10 | 19.43 | 86.84 | 80.95 | 81.08 | 83.99 | 72.24 | 76.53 |
| | | STARS_0.5 | **72.79** | **43.02** | **31.05** | **90.73** | **86.35** | **82.39** | **87.25** | **73.77** | **80.88** |
| | 0.4 | SAMPLING | 9.00 | 3.83 | 1.52 | 13.56 | 12.26 | 13.29 | 13.43 | 11.86 | 12.08 |
| | | STARS_0.1 | 62.69 | 41.60 | 19.74 | 87.14 | 81.17 | 81.82 | 84.46 | 73.15 | 77.13 |
| | | STARS_0.5 | **73.88** | **42.79** | **30.74** | **91.30** | **87.01** | **82.95** | **87.37** | **74.40** | **81.00** |
| | 0.2 | SAMPLING | 8.17 | 2.69 | 1.17 | 4.71 | 4.16 | 4.70 | 4.71 | 4.13 | 4.14 |
| | | STARS_0.1 | 62.93 | 41.11 | 18.81 | 86.25 | 80.22 | 80.67 | 83.60 | 71.72 | 75.92 |
| | | STARS_0.5 | **73.40** | **41.81** | **29.95** | **91.35** | **87.13** | **82.95** | **87.61** | **74.28** | **81.32** |

ates these five core dimensions, with each metric scored on a 0-10 scale. Flexibility is computed as the 30th percentile of the average scores across the other four metrics. Fluency measures both the quantity and diversity of ideas, capturing the ability to produce diverse responses. All metrics are judged by GPT-4.1-MINI with the specific prompts in Appendix B.1.

Table 2 presents experimental results evaluating the performance of the QWEN2.5-3B-INSTRUCT and LLAMA-3.2-3B-INSTRUCT models on the LiveIdeaBench dataset. The experimental setup extracts hidden states at layer 20, evaluates performance across temperature values ranging from 0.2 to 1.0, and $C \in \{0.1, 0.5\}$. STARS consistently achieves the highest average performance across most temperature settings. In terms of fluency (a metric reflecting diversity), our method achieves 5.09, nearly twice the temperature sampling (2.68) at $T = 0.2$. The results reveal a temperature-dependent pattern: while standard sampling shows declining performance as temperature decreases (from 5.71 at $T = 1.0$ to 5.01 at $T = 0.2$), STARS methods exhibit more stable performance across the temperature range. This suggests that STARS provides more robust text generation capabilities, particularly in low-temperature regimes where standard sampling typically suffers from reduced diversity. We provide a case study in Appendix B.9 that shows that STARS generates more creative ideas than standard temperature sampling. Table 14 in Appendix B.8 reports the average per-question runtime across models and tasks for Algorithm 3 and temperature sampling. The results indicate that Algorithm 3 incurs only a small absolute overhead, averaging no more than 2 seconds per question.

Table 2: Experimental results on `LiveIdeaBench` across different models and sampling temperatures. All values are reported on a scale with a maximum score of 10. Best-performing methods are highlighted in bold.

| | T | Method | Originality ↑ | Feasibility ↑ | Clarity ↑ | Fluency ↑ | Flexibility ↑ | Avg. ↑ |
|---|---|---|---|---|---|---|---|---|
| QWEN2.5-3B-INSTRUCT | 1.0 | SAMPLING | 6.22 | 5.50 | 5.78 | 5.42 | 5.61 | 5.71 |
| | | STARS_0.1 | 6.26 | **5.64** | **5.79** | 5.32 | **5.71** | 5.74 |
| | | STARS_0.5 | **6.30** | 5.45 | 5.75 | **5.81** | 5.70 | **5.80** |
| | 0.8 | SAMPLING | 6.20 | 5.55 | 5.76 | 5.02 | 5.50 | 5.60 |
| | | STARS_0.1 | **6.14** | 5.57 | **5.80** | 5.45 | 5.67 | 5.73 |
| | | STARS_0.5 | 6.13 | **5.62** | 5.70 | **5.96** | **5.74** | **5.83** |
| | 0.6 | SAMPLING | 6.07 | 5.63 | 5.68 | 4.51 | 5.38 | 5.45 |
| | | STARS_0.1 | **6.17** | 5.59 | **5.73** | **5.19** | **5.55** | **5.64** |
| | | STARS_0.5 | 6.01 | **5.68** | 5.72 | 5.17 | 5.48 | 5.61 |
| | 0.4 | SAMPLING | 6.03 | 5.56 | 5.75 | 3.57 | 5.14 | 5.21 |
| | | STARS_0.1 | **6.09** | **5.70** | **5.76** | 4.61 | 5.47 | 5.53 |
| | | STARS_0.5 | 5.99 | **5.70** | 5.73 | **5.02** | **5.50** | **5.59** |
| | 0.2 | SAMPLING | 6.04 | **5.69** | 5.74 | 2.68 | 4.92 | 5.01 |
| | | STARS_0.1 | **6.06** | 5.61 | **5.79** | 4.20 | 5.30 | 5.39 |
| | | STARS_0.5 | 6.04 | 5.62 | 5.68 | **5.09** | **5.52** | **5.59** |
| LLAMA-3.2-3B-INSTRUCT | 1.0 | SAMPLING | 6.51 | **5.34** | **5.89** | 5.41 | 5.66 | 5.76 |
| | | STARS_0.1 | 6.57 | 5.31 | **5.89** | 5.45 | 5.68 | 5.78 |
| | | STARS_0.5 | **6.60** | 5.32 | 5.85 | **5.60** | **5.71** | **5.82** |
| | 0.8 | SAMPLING | 6.47 | **5.39** | **5.92** | 5.07 | 5.58 | 5.69 |
| | | STARS_0.1 | 6.49 | **5.39** | **5.92** | 5.11 | 5.59 | 5.70 |
| | | STARS_0.5 | **6.56** | 5.36 | 5.91 | **5.22** | **5.62** | **5.73** |
| | 0.6 | SAMPLING | 6.43 | 5.42 | 5.99 | 4.59 | 5.46 | 5.58 |
| | | STARS_0.1 | 6.45 | **5.46** | **6.01** | 4.78 | 5.57 | 5.65 |
| | | STARS_0.5 | **6.53** | 5.42 | 5.97 | **4.81** | 5.58 | **5.66** |
| | 0.4 | SAMPLING | 6.41 | 5.42 | 6.00 | 4.17 | 5.35 | 5.47 |
| | | STARS_0.1 | 6.44 | 5.46 | **6.02** | 4.23 | 5.43 | 5.52 |
| | | STARS_0.5 | **6.50** | **5.44** | 5.99 | **4.41** | **5.48** | **5.56** |
| | 0.2 | SAMPLING | 6.42 | 5.45 | **6.03** | 3.27 | 5.19 | 5.27 |
| | | STARS_0.1 | 6.46 | **5.48** | **6.03** | 3.60 | 5.27 | 5.37 |
| | | STARS_0.5 | **6.51** | 5.46 | 6.02 | **4.04** | **5.39** | **5.48** |

## 7 CONCLUSION

We introduced STARS, a training-free, inference-time method that steers activations at every token across multiple concurrent generation paths. By jointly optimizing orthogonal steering directions on the Stiefel manifold, STARS maximizes the geometric volume of the hidden activations from these runs, explicitly promoting divergent internal states and implicitly inducing diverse trajectories. While a full Riemannian gradient descent offers convergence guarantees, it is too slow for real-time use; our lightweight one-step update with a closed-form aggressive stepsize enables low-latency, per-token intervention. Across test case generation and scientific discovery benchmarks, STARS substantially increases diversity without degrading quality, outperforming baselines.

## 8 ACKNOWLEDGEMENT

Man-Chung Yue is supported in part by the Hong Kong Research Grants Council under the GRF project 17309423. Viet Anh Nguyen gratefully acknowledges the support from the CUHK's Improvement on Competitiveness in Hiring New Faculties Funding Scheme, UGC ECS Grant 24210924, and UGC GRF Grant 14208625.

## 9 ETHICS STATEMENT

This work focuses on methods for improving the diversity of outputs from large language models through inference-time activation steering. Our study does not involve collecting or annotating human data, and all benchmarks employed are publicly available (e.g., `TESTEVAL`, `LiveIdeaBench`). The primary contribution is algorithmic, and no new risks of data misuse or privacy violations are introduced.

## 10 REPRODUCIBILITY STATEMENT

Our code implementation is publicly available at `https://github.com/lythk88/STARS`. All experiments use a random seed of 42 and are conducted on NVIDIA RTX A5000 24GB GPUs. We specify experimental details and hyperparameters in Section 6 and Appendix B. Our theoretical contributions are accompanied by complete proofs in Appendix A.

## 11 LLM USAGE STATEMENT

We used large language models exclusively for polishing the writing of this manuscript, specifically to improve clarity and readability. No parts of the technical contributions, mathematical derivations, experimental design, or technical results were generated by language models.

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

# A   PROOFS OF MAIN RESULTS

This section contains the proofs of all technical results presented in the main paper.

*Proof of Proposition 1.* Let $Q_1 \in \mathbb{R}^{d \times r}$ be an orthonormal basis of the column space of $H$, then we can extend $Q_1$ an orthogonal matrix $Q = [Q_1, Q_2] \in \mathbb{R}^{d \times d}$, where the columns of $Q_2 \in \mathbb{R}^{d \times (d-r)}$ span the orthogonal complement of the column space of $H$. Since $d \geq r + N$, we can take $V$ as any $N$ columns of the matrix $\sqrt{\alpha} Q_2$. Then one can verify that $V^\top V = \alpha I$ and $V^\top H = 0$. Also, we have

$$(H + V)^\top (H + V) = H^\top H + (H^\top V + V^\top H) + V^\top V = H^\top H + \alpha I,$$

which means $(H + V)^\top (H + V)$ is positive definite, hence full rank. $\qquad\square$

The following lemma generalizes the second-order boundness of polar retraction from the standard Stiefel manifold to the scaled Stiefel manifold.

**Lemma 1** (Second-order boundness of polar retraction). *For any $V \in \mathrm{St}(d, N, \alpha)$ and $U \in T_V \mathrm{St}(d, N, \alpha)$ with $\|U\|_F \leq \sqrt{\alpha}$, it holds that*

$$\|\mathrm{R}_V(U) - (V + U)\|_F \leq \frac{1}{\sqrt{\alpha}} \|U\|_F^2.$$

*We also have $\|\mathrm{R}_V(U) - \overline{V}\|_F \leq \|V + U - \overline{V}\|_F$ for all $\overline{V} \in \mathrm{St}(d, N, \alpha)$.*

*Proof of Lemma 1.* By the definition of the polar retraction, we have

$$\|\mathrm{R}_V(U) - (V + U)\|_F = \|\sqrt{\alpha}(V + U)(\alpha I + U^\top U)^{-1/2} - (V + U)\|_F$$
$$\leq \|V + U\|_F \|\sqrt{\alpha}(\alpha I + U^\top U)^{-1/2} - I\|_F.$$

Let $U^\top U = S \Sigma S^\top$ be a spectral decomposition of $U^\top U$ with $\Sigma = \mathrm{diag}(\lambda_1, \ldots, \lambda_N)$. For the second term in the product, we have

$$\|\sqrt{\alpha}(\alpha I + U^\top U)^{-1/2} - I\|_F^2 = \|(I + \frac{1}{\alpha} U^\top U)^{-1/2} - I\|_F^2$$
$$= \|(I + \frac{1}{\alpha} S \Sigma S^\top)^{-1/2} - I\|_F^2$$
$$= \sum_{i=1}^N [(1 + \frac{1}{\alpha} \lambda_i)^{-1/2} - 1]^2$$
$$\leq \frac{1}{4\alpha^2} \sum_{i=1}^N \lambda_i^2$$
$$= \frac{1}{4\alpha^2} \|U^\top U\|_F^2.$$

Hence when $\|U\|_F \leq \sqrt{\alpha}$, it holds that

$$\|\mathrm{R}_V(U) - (V + U)\|_F \leq \|V + U\|_F \|\sqrt{\alpha}(\alpha I + U^\top U)^{-1/2} - I\|_F$$
$$\leq \|V + U\|_F \times \frac{1}{2\alpha} \|U\|_F^2$$
$$\leq \frac{1}{\sqrt{\alpha}} \|U\|_F^2,$$

where we use the fact that $\|V + U\|_F \leq \|V\|_F + \|U\|_F \leq 2\sqrt{\alpha}$.

Now we prove the second statement. The convex hull of $\mathrm{St}(d, N, \alpha)$ is given by $\mathrm{co}\, \mathrm{St}(d, N, \alpha) = \{V \in \mathbb{R}^{d \times N} : \|V\|_{\mathsf{op}}^2 \leq \alpha\}$, where $\|V\|_{\mathsf{op}}$ denotes the spectral norm of $V$. We claim that $\mathrm{R}_V(U) = \mathrm{Proj}_{\mathrm{St}(d,N,\alpha)}(V + U) = \mathrm{Proj}_{\mathrm{co}\, \mathrm{St}(d,N,\alpha)}(V + U)$. Since $U \in T_V \mathrm{St}(d, N, \alpha)$, we have $(V +$

$U)^\top(V+U) = \alpha I + U^\top U$, which implies that all singular values of $V+U$ are at least $\sqrt{\alpha}$. Hence, our claim follows. For $V, \overline{V} \in \mathrm{St}(d, N, \alpha)$, and $U \in T_V\mathrm{St}(d, N, \alpha)$, we can write

$$\|\mathrm{R}_V(U) - \overline{V}\|_F = \|\mathrm{Proj}_{\mathrm{co}\,\mathrm{St}(d,N,\alpha)}(V+U) - \mathrm{Proj}_{\mathrm{co}\,\mathrm{St}(d,N,\alpha)}(\overline{V})\|_F$$
$$\leq \|V + U - \overline{V}\|_F$$

since projection onto a closed convex set is non-expansive. $\qquad\square$

The following proposition shows that with a proper stepsize, one can expect a decrease in the objective value with an Riemannian gradient step.

**Proposition 5** (Sufficient descent). *Let $V \in \mathcal{C}$. If $0 < \eta < \sqrt{\alpha}/\bar{D}$, then it holds that*

$$\ell(\mathrm{R}_V(-\eta\,\mathrm{grad}\,\ell(V))) - \ell(V) \leq -C(\eta)\|\mathrm{grad}\,\ell(V)\|_F^2,$$

*where* $C(\eta) = \eta - \frac{\bar{D}}{\sqrt{\alpha}}\eta^2 - \frac{L_2}{2}\eta^2$, $\bar{D} \triangleq \max_{V\in\mathcal{C}}\|\nabla\ell(V)\|_F < \infty$, $L_1 \triangleq$
$\sup_{V\in\mathcal{C}, \|U\|_F=1}\|\nabla^2\ell(V)[U]\|_F < \infty$ *and* $L_2 = L_1 + \bar{D}$.

*Proof of Proposition 5.* For simplicity, we define $P \triangleq H + V$. Let $\underline{\sigma}$ be the smallest singular value of $M$ over the set $\mathcal{C}$. Since $M$ is always full-rank for all $V \in \mathcal{C}$ and $\mathcal{C}$ is a compact set, $\underline{\sigma}$ is bounded away from 0, that is, $\underline{\sigma} > 0$. Noting that

$$\nabla\ell(V) = -2P(P^\top P)^{-1},$$

we must have

$$\bar{D} \triangleq \max_{V\in\mathcal{C}}\|\nabla\ell(V)\|_F \leq 2\max_{V\in\mathcal{C}}\|P\|_F\|(P^\top P)^{-1}\|_2 = 2\frac{1}{\underline{\sigma}^2}\max_{V\in\mathcal{C}}\|P\|_F < \infty.$$

Since $\mathrm{grad}\,\ell(V) = \mathrm{Proj}_{T_V}(\nabla\ell(V))$, it also holds that $\max_{V\in\mathcal{C}}\|\mathrm{grad}\,\ell(V)\|_F \leq \bar{D}$. By basic computation, the Euclidean Hessian of $\ell$ at $V$ evaluated in the direction $U$ is

$$\nabla^2\ell(V)[U] = 2P(P^\top P)^{-1}(U^\top P + P^\top U)(P^\top P)^{-1} - 2U(P^\top P)^{-1}. \qquad(6)$$

One can also show that there exits a $L_1 > 0$ such that for all $V \in \mathcal{C}$,

$$\sup_{\|U\|_F=1}\|\nabla^2\ell(V)[U]\|_F \leq L_1.$$

Thus, $\ell(V)$ is $L_1$-smooth on $\mathcal{C}$. For any $S \in T_V\mathrm{St}(d, N, \alpha)$, Chen et al. (2021, Lemma 2.4) shows that with $L_2 \triangleq L_1 + \bar{D}$, one has

$$\ell(\mathrm{R}_V(S)) \leq \ell(V) + \langle\mathrm{grad}\,\ell(V), \mathrm{R}_V(S) - V\rangle + \frac{L_2}{2}\|\mathrm{R}_V(S) - V\|_F^2$$

$$= \ell(V) + \langle\mathrm{grad}\,\ell(V), S\rangle + \langle\mathrm{grad}\,\ell(V), \mathrm{R}_V(S) - (V+S)\rangle + \frac{L_2}{2}\|\mathrm{R}_V(S) - V\|_F^2$$

$$\leq \ell(V) + \langle\mathrm{grad}\,\ell(V), S\rangle + \|\mathrm{grad}\,\ell(V)\|_F\|\mathrm{R}_V(S) - (V+S)\|_F + \frac{L_2}{2}\|\mathrm{R}_V(S) - V\|_F^2.$$
$$(7)$$

Since $\eta < \sqrt{\alpha}/\bar{D}$ and $\max_{V\in\mathcal{C}}\|\mathrm{grad}\,\ell(V)\|_F \leq \bar{D}$, we must have $\|\eta\,\mathrm{grad}\,\ell(V)\|_F < \sqrt{\alpha}$. Lemma 1 then asserts that

$$\|\mathrm{R}_V(\eta\,\mathrm{grad}\,\ell(V)) - (V + \eta\,\mathrm{grad}\,\ell(V))\|_F \leq \frac{\eta^2}{\sqrt{\alpha}}\|\mathrm{grad}\,\ell(V)\|_F^2,$$

and

$$\|\mathrm{R}_V(\eta\,\mathrm{grad}\,\ell(V)) - V\|_F \leq \eta\|\mathrm{grad}\,\ell(V)\|_F.$$

Then, take $S = -\eta\,\mathrm{grad}\,\ell(V)$ in (7) and we have

$$\ell(\mathrm{R}_V(-\eta\,\mathrm{grad}\,\ell(V))) \leq \ell(V) + \langle\mathrm{grad}\,\ell(V), S\rangle + \|\mathrm{grad}\,\ell(V)\|_F\|\mathrm{R}_V(S) - (V+S)\|_F$$

$$+ \frac{L_2}{2}\|\mathrm{R}_V(S) - V\|_F^2$$

$$\leq \ell(V) - \eta\|\mathrm{grad}\,\ell(V)\|_F^2 + \bar{D}\frac{\eta^2}{\sqrt{\alpha}}\|\mathrm{grad}\,\ell(V)\|_F^2 + \frac{L_2\eta^2}{2}\|\mathrm{grad}\,\ell(V)\|_F^2$$

$$= \ell(V) - C(\eta)\|\mathrm{grad}\,\ell(V)\|_F^2,$$

where $C(\eta) = \eta[1 - (\frac{\bar{D}}{\sqrt{\alpha}} + \frac{L_2}{2})\eta]$. The proof is completed. $\qquad\square$

Now we are ready to prove Theorem 1.

*Proof of Theorem 1.* The backtracking line-search procedure in Algorithm 2 gives that for each $k$ there exits a constant $C_k > 0$ such that

$$\ell(V_k) - \ell(V_{k+1}) \geq C_k \|S_k\|_F^2. \tag{8}$$

We first prove that $C_k$ is bounded away from 0. Since $\operatorname{grad} \ell(V) = \operatorname{Proj}_{T_V}(\nabla \ell(V))$, it also holds that $\max_{V \in \mathcal{C}} \|\operatorname{grad} \ell(V)\|_F \leq \bar{D}$ and thus, $\|\operatorname{grad} \ell(V_k)\|_F \leq \bar{D}$ for all $k$. Hence, $\sqrt{\alpha}/\|\operatorname{grad} \ell(V_k)\|_F$ must has a lower bound $\sqrt{\alpha}/\bar{D}$. Note that $\eta \leq \bar{\eta} \leq \sqrt{\alpha}/\bar{D}$, we always have $\|\eta \operatorname{grad} \ell(V_k)\|_F \leq \sqrt{\alpha}$. Proposition 5 guarantees

$$\ell(V_k) - \ell(V_{k+1}) \geq C(\eta) \|\operatorname{grad} \ell(V_k)\|_F^2.$$

On the other hand, if the backtracking procedure does not terminate for a certain value $\eta$, then

$$\ell(V_k) - \ell(V_{k+1}) < c\eta \|\operatorname{grad} \ell(V_k)\|_F^2.$$

If both are true simultaneously, then we must have $c\eta > C(\eta)$, that is,

$$\eta > \frac{2(1-c)\sqrt{\alpha}}{2\bar{D} + L_2\sqrt{\alpha}}.$$

Thus, the backtracking procedure terminates when $\eta$ is below this value. Hence, we can conclude that the returned $\eta$ satisfies

$$\eta \geq \min\{\bar{\eta}, \frac{2\rho(1-c)\sqrt{\alpha}}{2\bar{D} + L_2\sqrt{\alpha}}\},$$

which implies that $C_k \geq C_{\min} \triangleq c\min\{\bar{\eta}, \frac{2\rho(1-c)\sqrt{\alpha}}{2\bar{D}+L_2\sqrt{\alpha}}\}$ for all $k$.

Based on a standard telescoping sum argument, we get

$$\begin{aligned}
\ell(V_0) - \ell^\star &\geq \ell(V_0) - \ell(V_K) \\
&= \sum_{k=0}^{K-1} [\ell(V_k) - \ell(V_{k+1})] \\
&\geq \sum_{k=0}^{K-1} C_k \|\operatorname{grad} \ell(V_k)\|_F^2 \\
&\geq C_{\min} \sum_{k=0}^{K-1} \|\operatorname{grad} \ell(V_k)\|_F^2 \\
&\geq C_{\min} \times K \min_{k=0,\dots,K-1} \|\operatorname{grad} \ell(V_k)\|_F^2,
\end{aligned}$$

which completes the proof. □

*Proof of Proposition 2.* From the Taylor expansion $(I + X)^{-1/2} = I - \frac{1}{2}X + O(\|X\|^2)$, we have

$$(\alpha I + \eta^2 S^\top S)^{-1/2} = \alpha^{-1/2}(I + \frac{\eta^2}{\alpha}S^\top S)^{-1/2} = \alpha^{-1/2}(I - \frac{\eta^2}{2\alpha}S^\top S + O(\eta^2)),$$

which gives the following expansion of $\mathrm{R}_V(\eta S)$:

$$\mathrm{R}_V(\eta S) = \sqrt{\alpha}(V + \eta S)(\alpha I + \eta^2 S^\top S)^{-1/2} = V + \eta S - \frac{\eta^2}{2\alpha}V S^\top S + O(\eta^3).$$

Thus, we have

$$\begin{aligned}
&(H + \mathrm{R}_V(\eta S))^\top (H + \mathrm{R}_V(\eta S)) \\
=&(H + V + \eta S - \frac{\eta^2}{2\alpha}V S^\top S + O(\eta^3))^\top (H + V + \eta S - \frac{\eta^2}{2\alpha}V S^\top S + O(\eta^3)) \\
=&(H + V)^\top (H + V) + \eta(H^\top S + S^\top H) - \frac{\eta^2}{2\alpha}(H^\top V S^\top S + S^\top S V^\top H) + O(\eta^3).
\end{aligned}$$

Since $\log \det(I + X) = \operatorname{Tr} X - \frac{1}{2}\operatorname{Tr} X^2 + O(\|X\|^3)$, for $\mathcal{L}(\eta)$ it holds that

$$
\begin{aligned}
\mathcal{L}(\eta) &= -\log \det[(H + \mathrm{R}_V(\eta S))^\top (H + \mathrm{R}_V(\eta S))] \\
&= -\log \det[(H + V)^\top (H + V) + \eta(H^\top S + S^\top H) - \frac{\eta^2}{2\alpha}(H^\top V S^\top S + S^\top S V^\top H) + O(\eta^3)] \\
&= -\log \det A - \log \det[I + \eta A^{-1}(H^\top S + S^\top H) - \frac{\eta^2}{2\alpha} A^{-1}(H^\top V S^\top S + S^\top S V^\top H) + O(\eta^3)] \\
&= -\log \det A - \eta \operatorname{Tr}[A^{-1}(H^\top S + S^\top H)] \\
&\quad + \frac{\eta^2}{2\alpha}\operatorname{Tr}[A^{-1}(H^\top V S^\top S + S^\top S V^\top H)] \\
&\quad + \frac{\eta^2}{2}\operatorname{Tr}[\left((A^{-1}(H^\top S + S^\top H))^2\right] + O(\eta^3) \\
&= -\log \det A - D_1 \eta + \frac{1}{2}D_2\eta^2 + O(\eta^3),
\end{aligned}
$$

where $D_1 = \operatorname{Tr}[A^{-1}(H^\top S + S^\top H)]$, $D_2 = \frac{1}{\alpha}\operatorname{Tr}[A^{-1}(H^\top V S^\top S + S^\top S V^\top H)] + \operatorname{Tr}[\left((A^{-1}(H^\top S + S^\top H))^2\right]$. The proof is completed. $\qquad \square$

*Proof of Proposition 3.* With $V_0$ from Algorithm 1, we have $H^\top V_0 = 0$ and $V_0^\top V_0 = \alpha I$. Then,

$$
\operatorname{Proj}_{T_{V_0}}(H + V_0) = H + V_0 - V_0 \frac{(H + V_0)^\top V_0 + V_0^\top (H + V_0)}{2\alpha} = H,
$$

which means that $H$ is a valid tangent vector at $V_0$. Then we have

$$
\begin{aligned}
\langle H, \operatorname{grad}\ell(V_0)\rangle = \langle H, \nabla\ell(V_0)\rangle &= -2\langle H, (H + V_0)\left((H + V_0))(H + V_0)\right)^{-1}\rangle \\
&= -2\langle H, H\left((H + V_0)(H + V_0)\right)^{-1}\rangle < 0,
\end{aligned}
$$

since $H + V_0$ is of full rank. $\qquad \square$

*Proof of Proposition 4.* Let $H = Q\Sigma R^\top$ and $r = \operatorname{rank}(H)$ be the SVD of $H$, where $Q \in \mathbb{R}^{d \times d}$, $\Sigma \in \mathbb{R}^{d \times N}$ and $R \in \mathbb{R}^{N \times N}$. Then we have

$$
\begin{aligned}
A^{-1} &= [(H + V_0)^\top (H + V_0)]^{-1} \\
&= (H^\top H + V_0^\top H + H^\top V_0 + V_0^\top V_0)^{-1} \\
&= (H^\top H + \alpha I)^{-1} \\
&= (R\Sigma^2 R^\top + \alpha I)^{-1} \\
&= R(\Sigma^2 + \alpha I)^{-1} R^\top,
\end{aligned}
$$

which gives that

$$
A^{-1}H^\top H = R(\Sigma^2 + \alpha I)^{-1}R^\top \times R\Sigma^2 R^\top = R(\Sigma^2 + \alpha I)^{-1}\Sigma^2 R^\top,
$$

and

$$
(A^{-1}H^\top H)^2 = R(\Sigma^2 + \alpha I)^{-1}\Sigma^2(\Sigma^2 + \alpha I)^{-1}\Sigma^2 R^\top.
$$

With $S = H$ we have

$$
\begin{aligned}
D_1 &= \operatorname{Tr}[A^{-1}(H^\top S + S^\top H)] = 2\operatorname{Tr}(A^{-1}H^\top H) \\
&= 2\operatorname{Tr}[R(\Sigma^2 + \alpha I)^{-1}\Sigma^2 R^\top] \\
&= 2\operatorname{Tr}[(\Sigma^2 + \alpha I)^{-1}\Sigma^2] \\
&= 2\sum_{i=1}^{r}\frac{\sigma_i^2}{\sigma_i^2 + \alpha} > 0.
\end{aligned}
$$

Noting that $V_0^\top S = V_0^\top H = 0$, for $D_2$ we have

$$
\begin{aligned}
D_2 &= \frac{1}{\alpha} \operatorname{Tr}[A^{-1}(H^\top V_0 S^\top S + S^\top S V_0^\top H)] + \operatorname{Tr}[((A^{-1}(H^\top S + S^\top H))^2] \\
&= 4 \operatorname{Tr}[(A^{-1} H^\top H)^2] \\
&= 4 \operatorname{Tr}[(\Sigma^2 + \alpha I)^{-1} \Sigma^2 (\Sigma^2 + \alpha I)^{-1} \Sigma^2] \\
&= 4 \sum_{i=1}^{r} \frac{\sigma_i^4}{(\sigma_i^2 + \alpha)^2}.
\end{aligned}
$$

The positiveness of $\eta^\star$ comes from $D_1, D_2 > 0$. The proof is completed. $\qquad\square$

## B    EXPERIMENTAL DETAILS

### B.1    EXPERIMENT SETTINGS

#### B.1.1    TEST CASE GENERATION

This is the prompt for generating the test cases.

---

**Test Case Generation Prompt**

Please write a test method for the function `func_name` given the following program under test and function description. Your answer should only contain one test input.
**Program under test:** —- program —-
**Function description for `func_name`:** —- description —-
Your test method should begin with:

```
def test_{func_name}():
    solution = Solution()
```

---

#### B.1.2    SCIENTIFIC DISCOVERY

We use the following prompts to generate ideas and evaluate the quality in the scientific discovery task.

---

**Idea Generation Prompt Example**

I'll be submitting your next responses to a *"Good Scientific Idea"* expert review panel. If they consider your idea to be a good one, you'll receive a reward.
Your assigned keyword is: `keywords`. You may provide background information.
The idea MUST be concisely expressed within 100 words total (including any background information).
*Note: good scientific ideas should be original (novel contribution), feasible (technically implementable), clearly articulated, and address meaningful problems in the field.*

---

---

**Judge Prompt Example**

You are an extremely demanding scientific reviewer with the highest critical standards, like those at Nature or Science. When evaluating scientific ideas, assess them on three key dimensions:

1. **Originality**: Novel contribution to unexplored areas or innovative approaches to existing problems.
2. **Feasibility**: Technical implementation and practicality.
3. **Clarity**: How well-articulated and easy to understand the idea is.

Your response should consist of two parts: a brief text analysis (under 100 words), followed by a JSON score block:

```
{
 "originality": <score_1_to_10>,
 "feasibility": <score_1_to_10>,
 "clarity": <score_1_to_10>
}

-------------------------
{role: user, content: prompt}
```

---

**Fluency Judge Prompt Example**

Here are four ideas submitted to the *"Good Scientific Ideas"* Competition, all related to `keyword`:
**Idea 1:**
{{A}}
**Idea 2:**
{{B}}
**Idea 3:**
{{C}}
**Idea 4:**
{{D}}
**Question:**
Evaluate the similarity between the four ideas that both relate to `keyword`. Please choose the best answer:
A. Completely different ideas addressing different problems, despite relating to the same keyword.
B. Different ideas but addressing similar problems.
C. Similar ideas addressing similar or identical problems.
D. Academically identical ideas with the same core approach and problem statement.
**ONLY ANSWER A/B/C/D, DO NOT EXPLAIN.**

## B.2 EVALUATION ACROSS STEERING LAYERS

We set the random seed to 42 for all experiments. We conduct experiments to investigate the impact of modifying the steering layers on the performance of the proposed algorithm. For simplicity, we sample 10% of each dataset to select the optimal layer for steering.

To identify optimal steering layers, we systematically evaluated layers 0-17 for Gemma-1.1-2b-it and layers 0-27 for Qwen3-1.7b. For each layer, we calculated the mean values across five evaluation metrics: Syntax Correctness, Executable Correctness, Assertion Correctness, Overall Coverage Line, and Overall Coverage Branch. Subsequently, we ranked the layers by their performance on each metric and computed the average rank across all metrics. The layer achieving the lowest average rank was designated as exhibiting superior overall performance and selected for further experiments. See Table 3 and Table 4 for the ranking results.

Table 3: Layer ranking for model Gemma-1.1-2b-it based on multiple evaluation metrics. The Average represents the average ranking across these metrics, with lower values indicating better overall performance.

| Layer | Correctness | | | Overall Coverage | | Average |
|---|---|---|---|---|---|---|
| | Syntax | Executable | Assertion | Line | Branch | |
| 3 | 2 | 1 | 1 | 2 | 2 | 1.6 |
| 16 | 7 | 5 | 5 | 1 | 1 | 3.8 |
| 15 | 6 | 6 | 6 | 4 | 4 | 5.2 |
| 4 | 14 | 2 | 2 | 6 | 6 | 6 |
| 11 | 3 | 8 | 7 | 7 | 10 | 7 |
| 2 | 8 | 4 | 4 | 9 | 10 | 7 |
| 17 | 5 | 14 | 14 | 3 | 3 | 7.8 |
| 8 | 4 | 3 | 3 | 16 | 16 | 8.4 |
| 9 | 1 | 8 | 7 | 14 | 14 | 8.8 |
| 1 | 8 | 12 | 12 | 7 | 6 | 9 |
| 5 | 12 | 13 | 13 | 5 | 5 | 9.6 |
| 0 | 13 | 7 | 9 | 9 | 12 | 10 |
| 7 | 15 | 11 | 11 | 9 | 6 | 10.4 |
| 14 | 11 | 10 | 10 | 15 | 15 | 12.2 |
| 13 | 10 | 15 | 15 | 9 | 12 | 12.2 |
| 6 | 18 | 16 | 16 | 9 | 6 | 13 |
| 10 | 17 | 17 | 17 | 18 | 16 | 17 |
| 12 | 16 | 18 | 18 | 16 | 18 | 17.2 |

Table 5 shows the results when extracting the hidden states from layers 10 and 30 with $C \in \{0.1, 0.5\}$.

Table 4: Layer ranking for model Qwen3-1.7b based on multiple evaluation metrics. The Average represents the average ranking across these metrics, with lower values indicating better overall performance.

| Layer | Syntax | Correctness Executable | Assertion | Overall Coverage Line | Branch | Average |
|---|---|---|---|---|---|---|
| 6 | 4 | 5 | 1 | 2 | 1 | 2.6 |
| 1 | 1 | 1 | 2 | 6 | 7 | 3.4 |
| 2 | 5 | 3 | 4 | 3 | 2 | 3.4 |
| 3 | 6 | 2 | 3 | 8 | 10 | 5.8 |
| 9 | 9 | 11 | 8 | 5 | 4 | 7.4 |
| 24 | 12 | 4 | 7 | 10 | 8 | 8.2 |
| 15 | 7 | 9 | 14 | 7 | 6 | 8.6 |
| 4 | 11 | 6 | 5 | 12 | 13 | 9.4 |
| 12 | 3 | 14 | 12 | 9 | 9 | 9.4 |
| 20 | 14 | 12 | 19 | 1 | 3 | 9.8 |
| 21 | 16 | 8 | 9 | 14 | 11 | 11.6 |
| 0 | 10 | 7 | 6 | 22 | 21 | 13.2 |
| 11 | 2 | 18 | 13 | 17 | 17 | 13.4 |
| 7 | 19 | 10 | 11 | 15 | 12 | 13.4 |
| 10 | 14 | 17 | 9 | 11 | 16 | 13.4 |
| 17 | 13 | 13 | 15 | 16 | 15 | 14.4 |
| 8 | 23 | 21 | 20 | 4 | 5 | 14.6 |
| 26 | 18 | 15 | 18 | 13 | 14 | 15.6 |
| 14 | 8 | 19 | 16 | 19 | 20 | 16.4 |
| 5 | 22 | 16 | 17 | 18 | 18 | 18.2 |
| 23 | 24 | 20 | 22 | 20 | 19 | 21 |
| 16 | 17 | 23 | 21 | 24 | 24 | 21.8 |
| 19 | 25 | 22 | 23 | 23 | 23 | 23.2 |
| 25 | 27 | 24 | 24 | 21 | 22 | 23.6 |
| 22 | 21 | 26 | 26 | 26 | 26 | 25 |
| 18 | 26 | 25 | 25 | 25 | 25 | 25.2 |
| 13 | 20 | 28 | 28 | 28 | 28 | 26.4 |
| 27 | 28 | 27 | 27 | 27 | 27 | 27.2 |

Table 5: Experimental results on 10% of `LiveIdeaBench` dataset across different sampling temperatures on Qwen2.5-3B-Instruct. All values are reported on a scale with a maximum score of 10. Best-performing methods are highlighted in bold.

| T | Method | Originality ↑ | Feasibility ↑ | Clarity ↑ | Fluency ↑ | Flexibility ↑ | Avg. ↑ |
|---|---|---|---|---|---|---|---|
| | Sampling | 6.22 | 5.50 | 5.78 | 5.42 | 5.61 | 5.71 |
| | STARS_0.1 − 10 | **6.34** | 5.58 | 5.80 | 5.17 | 5.60 | 5.70 |
| 1.0 | STARS_0.1 − 30 | 6.21 | **5.60** | **5.83** | 5.27 | 5.63 | 5.71 |
| | STARS_0.5 − 10 | 6.26 | 5.46 | 5.81 | **5.70** | **5.68** | **5.78** |
| | STARS_0.5 − 30 | 6.30 | 5.54 | 5.80 | 5.27 | 5.63 | 5.71 |
| | Sampling | 6.20 | 5.55 | 5.76 | 5.02 | 5.50 | 5.60 |
| | STARS_0.1 − 10 | **6.23** | 5.47 | 5.78 | 5.07 | 5.58 | 5.63 |
| 0.8 | STARS_0.1 − 30 | 6.11 | **5.62** | 5.82 | 4.97 | 5.55 | 5.61 |
| | STARS_0.5 − 10 | 6.21 | 5.51 | 5.80 | **5.17** | **5.60** | **5.66** |
| | STARS_0.5 − 30 | 6.21 | 5.58 | **5.87** | 4.97 | 5.55 | 5.63 |
| | Sampling | 6.07 | **5.63** | 5.68 | 4.51 | 5.38 | 5.45 |
| | STARS_0.1 − 10 | 6.13 | 5.56 | **5.85** | 4.58 | 5.40 | 5.51 |
| 0.6 | STARS_0.1 − 30 | 6.15 | 5.60 | 5.84 | 4.23 | 5.37 | 5.44 |
| | STARS_0.5 − 10 | **6.21** | 5.52 | 5.75 | 4.81 | 5.45 | **5.55** |
| | STARS_0.5 − 30 | 6.14 | 5.55 | 5.71 | **4.86** | **5.47** | **5.55** |
| | Sampling | 6.03 | 5.56 | 5.75 | 3.57 | 5.14 | 5.21 |
| | STARS_0.1 − 10 | 6.08 | 5.54 | 5.80 | 4.15 | 5.29 | 5.37 |
| 0.4 | STARS_0.1 − 30 | 6.06 | **5.71** | 5.82 | 4.03 | 5.26 | 5.37 |
| | STARS_0.5 − 10 | **6.15** | 5.46 | 5.76 | 4.76 | 5.44 | 5.51 |
| | STARS_0.5 − 30 | 6.01 | 5.61 | **5.82** | **4.79** | **5.51** | **5.59** |
| | Sampling | 6.04 | **5.69** | 5.74 | 2.68 | 4.92 | 5.01 |
| | STARS_0.1 − 10 | 6.07 | 5.58 | 5.78 | 3.06 | 5.01 | 5.10 |
| 0.2 | STARS_0.1 − 30 | 6.14 | 5.64 | 5.77 | 3.26 | 5.13 | 5.19 |
| | STARS_0.5 − 10 | 6.05 | 5.54 | 5.78 | 4.00 | 5.25 | 5.32 |
| | STARS_0.5 − 30 | **6.17** | 5.56 | **5.79** | **4.41** | **5.41** | **5.47** |

### B.3 COMPARE INTERVENTION POSITIONS

A critical design choice in activation steering is determining where in the model architecture to apply the intervention. We investigate two natural intervention points within each transformer layer: (1) before the output projection of the attention mechanism (o_proj), which captures the attended representations before they are projected back to the residual dimension, and (2) the output of the MLP block, which represents the processed information after the feed-forward transformation.

Table 6 shows the experimental results on TESTEVAL with Gemma-1.1-2-it across different temperatures and two intervene positions (attention and residual stream).

Table 6: Experimental results on TESTEVAL across different sampling temperatures. All the numbers are in percentages. Best-performing methods are highlighted in bold. STARS_RES means applying STARS to the residual stream.

| | T | Method | Correctness ↑ | | | Overall cov. ↑ | | Line $cov@k$ ↑ | | Branch $cov@k$ ↑ | |
|---|---|---|---|---|---|---|---|---|---|---|---|
| | | | syntax | exe | assert | line | branch | $k=1$ | $k=5$ | $k=1$ | $k=5$ |
| GEMMA-1.1-2B-IT | 1.0 | SAMPLING | **89.90** | 1.02 | 0.00 | 0.00 | 0.00 | 0.00 | 0.00 | 0.00 | 0.00 |
| | | STARS_RES_0.1 | 87.71 | **6.02** | **5.93** | 33.54 | 30.10 | 31.54 | 32.55 | 27.64 | 28.94 |
| | | STARS_RES_0.5 | 26.40 | 0.02 | 0.02 | 0.05 | 0.00 | 0.05 | 0.05 | 0.00 | 0.00 |
| | | STARS_0.1 | 86.48 | 1.43 | 1.17 | 10.03 | 9.32 | 9.97 | 10.02 | 9.12 | 9.25 |
| | | STARS_0.5 | 79.76 | 3.83 | 3.45 | **34.76** | **30.93** | **33.81** | **34.45** | **29.58** | **30.40** |
| | 0.8 | SAMPLING | **91.79** | 1.19 | 0.00 | 1.44 | 1.41 | 1.37 | 1.42 | 1.27 | 1.35 |
| | | STARS_RES_0.1 | 87.71 | **6.02** | **5.93** | **33.54** | **30.10** | 31.54 | 32.55 | 27.64 | 28.94 |
| | | STARS_RES_0.5 | 28.81 | 0.00 | 0.00 | 0.00 | 0.00 | 0.00 | 0.00 | 0.00 | 0.00 |
| | | STARS_0.1 | 87.83 | 1.45 | 1.12 | 10.77 | 9.88 | 10.64 | 10.72 | 9.65 | 9.78 |
| | | STARS_0.5 | 79.43 | 3.79 | 3.45 | 32.96 | 30.02 | **31.70** | **32.59** | **28.23** | **29.42** |
| | 0.6 | SAMPLING | **93.45** | 1.29 | 0.00 | 3.01 | 2.84 | 2.94 | 2.77 | 2.70 | 2.77 |
| | | STARS_RES_0.1 | 90.74 | **6.52** | **6.36** | 33.23 | 29.73 | 31.09 | 32.46 | 26.94 | 28.61 |
| | | STARS_RES_0.5 | 24.83 | 0.00 | 0.00 | 0.00 | 0.00 | 0.00 | 0.00 | 0.00 | 0.00 |
| | | STARS_0.1 | 89.19 | 1.71 | 1.43 | 12.10 | 11.22 | 12.03 | 12.10 | 10.99 | 11.15 |
| | | STARS_0.5 | 79.24 | 3.86 | 3.24 | **34.07** | **30.38** | **32.04** | **33.22** | **28.00** | **29.46** |
| | 0.4 | SAMPLING | **95.40** | 1.36 | 1.26 | 2.33 | 2.31 | 2.29 | 2.32 | 2.20 | 2.27 |
| | | STARS_RES_0.1 | 90.83 | **6.55** | **6.38** | 32.45 | 29.32 | 30.90 | 31.77 | 27.35 | 28.60 |
| | | STARS_RES_0.5 | 27.50 | 0.00 | 0.00 | 0.00 | 0.00 | 0.00 | 0.00 | 0.00 | 0.00 |
| | | STARS_0.1 | 90.43 | 1.48 | 1.43 | 8.44 | 7.69 | 8.41 | 8.44 | 7.58 | 7.63 |
| | | STARS_0.5 | 80.31 | 4.38 | 4.00 | **40.03** | **35.96** | **38.39** | **39.35** | **33.73** | **35.07** |
| | 0.2 | SAMPLING | **95.64** | 1.36 | 1.36 | 1.44 | 1.41 | 1.37 | 1.42 | 1.27 | 1.35 |
| | | STARS_RES_0.1 | 90.67 | **6.33** | **6.21** | 31.09 | 27.62 | 30.21 | 30.60 | 26.33 | 26.92 |
| | | STARS_RES_0.5 | 28.14 | 0.02 | 0.00 | 0.03 | 0.00 | 0.03 | 0.03 | 0.00 | 0.00 |
| | | STARS_0.1 | 91.52 | 1.45 | 1.26 | 7.28 | 6.50 | 7.23 | 7.25 | 6.36 | 6.40 |
| | | STARS_0.5 | 78.93 | 4.38 | 4.02 | **39.03** | **35.05** | **37.38** | **38.67** | **32.94** | **34.39** |

These findings indicate that while MLP-based interventions can be effective at moderate steering strengths for improving execution correctness, attention-based interventions (before o_proj) offer better robustness and scalability for maximizing test coverage. The attention mechanism appears to provide a more stable intervention point that tolerates stronger perturbations without catastrophic degradation in output quality. This may be because intervening before the output projection allows the model's subsequent layers to better adapt and compensate for the perturbation. Based on these results, we adopt attention-based intervention (before o_proj) as our default approach throughout the paper.

## B.4 ADDITIONAL BASELINE METHODS

To further validate the effectiveness of our approach, we compare against additional baseline methods that provide alternative strategies for controlling model behavior during generation. Specifically, we evaluate two natural baselines. The first one is **RAND**, which applies randomly sampled steering vectors to explore the impact of arbitrary directional interventions. The second one is **HMEAN**, which uses steering vectors computed as the mean offset from a mean point to test whether spreading activations uniformly from a central location improves diversity. Specifically, with a set of activations $\{h_i\}_{i=1}^N$, we first compute the mean $\bar{h} = \frac{1}{N}\sum_{i=1}^N h_i$. With a scalar $\alpha > 0$, the steering vector for the $i$-th path is $v_i = \alpha \frac{h_i - \bar{h}}{\|h_i - \bar{h}\|_2}$.

Table 7 shows the experimental results on TESTEVAL with GEMMA-1.1-2-IT across five temperature settings $T \in \{0.2, 0.4, 0.6, 0.8, 1.0\}$. We compare vanilla temperature sampling against RAND and MEAN baselines at two steering strengths (0.1 and 0.5), as well as our proposed STARS method.

Table 7: Experimental results on TESTEVAL across different sampling temperatures. All the numbers are in percentages. Best-performing methods are highlighted in bold.

| | T | Method | Correctness ↑ | | | Overall cov. ↑ | | Line $cov@k$ ↑ | | Branch $cov@k$ ↑ | |
|---|---|---|---|---|---|---|---|---|---|---|---|
| | | | syntax | exe | assert | line | branch | $k=1$ | $k=5$ | $k=1$ | $k=5$ |
| GEMMA-1.1-2B-IT | 1.0 | SAMPLING | 89.90 | 1.02 | 0.00 | 0.00 | 0.00 | 0.00 | 0.00 | 0.00 | 0.00 |
| | | RAND_0.1 | 89.81 | 1.14 | 1.10 | 4.89 | 4.54 | 4.83 | 4.88 | 4.40 | 4.45 |
| | | RAND_0.5 | 89.98 | 1.07 | 0.98 | 6.48 | 6.14 | 6.44 | 6.46 | 6.05 | 6.10 |
| | | MEAN_0.1 | 90.64 | 1.10 | 1.02 | 4.96 | 4.56 | 4.92 | 4.95 | 4.44 | 4.51 |
| | | MEAN_0.5 | **90.76** | 1.12 | 1.07 | 4.51 | 4.20 | 4.43 | 4.49 | 4.00 | 4.13 |
| | | STARS_0.1 | 86.48 | 1.43 | 1.17 | 10.03 | 9.32 | 9.97 | 10.02 | 9.12 | 9.25 |
| | | STARS_0.5 | 79.76 | **3.83** | **3.45** | **34.76** | **30.93** | **33.81** | **34.45** | **29.58** | **30.40** |
| | 0.8 | SAMPLING | 91.79 | 1.19 | 0.00 | 1.44 | 1.41 | 1.37 | 1.42 | 1.27 | 1.35 |
| | | RAND_0.1 | 92.02 | 1.26 | 1.05 | 5.19 | 4.72 | 5.13 | 5.17 | 4.59 | 4.68 |
| | | RAND_0.5 | 91.43 | 1.26 | 1.14 | 4.73 | 4.38 | 4.71 | 4.72 | 4.31 | 4.34 |
| | | MEAN_0.1 | 92.10 | 1.26 | 1.24 | 3.94 | 3.54 | 3.89 | 3.92 | 3.41 | 3.47 |
| | | MEAN_0.5 | **92.43** | 1.14 | 1.07 | 5.34 | 5.07 | 5.29 | 5.33 | 4.94 | 5.02 |
| | | STARS_0.1 | 87.83 | 1.45 | 1.12 | 10.77 | 9.88 | 10.64 | 10.72 | 9.65 | 9.78 |
| | | STARS_0.5 | 79.43 | **3.79** | **3.45** | **32.96** | **30.02** | **31.70** | **32.59** | **28.23** | **29.42** |
| | 0.6 | SAMPLING | 93.45 | 1.29 | 0.00 | 3.01 | 2.84 | 2.94 | 2.77 | 2.70 | 2.77 |
| | | RAND_0.1 | 93.17 | 1.10 | 1.07 | 2.78 | 2.70 | 2.72 | 2.77 | 2.58 | 2.66 |
| | | RAND_0.5 | 93.00 | 0.90 | 0.86 | 2.78 | 2.64 | 2.74 | 2.75 | 2.52 | 2.60 |
| | | MEAN_0.1 | **94.67** | 1.14 | 1.12 | 2.77 | 2.68 | 2.73 | 2.77 | 2.55 | 2.60 |
| | | MEAN_0.5 | 94.33 | 1.12 | 1.10 | 3.21 | 2.96 | 3.18 | 3.19 | 2.90 | 2.92 |
| | | STARS_0.1 | 89.19 | 1.71 | 1.43 | 12.10 | 11.22 | 12.03 | 12.10 | 10.99 | 11.15 |
| | | STARS_0.5 | 79.24 | **3.86** | **3.24** | **34.07** | **30.38** | **32.04** | **33.22** | **28.00** | **29.46** |
| | 0.4 | SAMPLING | 95.40 | 1.36 | 1.26 | 2.33 | 2.31 | 2.29 | 2.32 | 2.20 | 2.27 |
| | | RAND_0.1 | 95.14 | 1.19 | 1.14 | 1.92 | 1.88 | 1.88 | 1.91 | 1.77 | 1.83 |
| | | RAND_0.5 | 94.67 | 1.24 | 1.19 | 2.23 | 2.08 | 2.20 | 2.23 | 1.98 | 2.04 |
| | | MEAN_0.1 | 95.90 | 1.26 | 1.24 | 2.33 | 2.24 | 2.29 | 2.32 | 2.13 | 2.20 |
| | | MEAN_0.5 | **95.93** | 1.07 | 1.07 | 1.41 | 1.36 | 1.40 | 1.41 | 1.29 | 1.32 |
| | | STARS_0.1 | 90.43 | 1.48 | 1.43 | 8.44 | 7.69 | 8.41 | 8.44 | 7.58 | 7.63 |
| | | STARS_0.5 | 80.31 | **4.38** | **4.00** | **40.03** | **35.96** | **38.39** | **39.35** | **33.73** | **35.07** |
| | 0.2 | SAMPLING | 95.64 | 1.36 | 1.36 | 1.44 | 1.41 | 1.37 | 1.42 | 1.27 | 1.35 |
| | | RAND_0.1 | 96.24 | 1.33 | 1.31 | 1.89 | 1.88 | 1.84 | 1.87 | 1.77 | 1.81 |
| | | RAND_0.5 | 95.86 | 1.21 | 1.14 | 1.89 | 1.88 | 1.83 | 1.88 | 1.77 | 1.85 |
| | | MEAN_0.1 | 96.57 | 1.45 | 1.40 | 1.89 | 1.88 | 1.87 | 1.88 | 1.77 | 1.81 |
| | | MEAN_0.5 | **96.74** | 1.29 | 1.29 | 1.41 | 1.36 | 1.41 | 1.41 | 1.31 | 1.33 |
| | | STARS_0.1 | 91.52 | 1.45 | 1.26 | 7.28 | 6.50 | 7.23 | 7.25 | 6.36 | 6.40 |
| | | STARS_0.5 | 78.93 | **4.38** | **4.02** | **39.03** | **35.05** | **37.38** | **38.67** | **32.94** | **34.39** |

Table 8 presents the experimental results on `TESTEVAL` with QWEN3-1.7B across five temperature settings T $\in$ {0.2, 0.4, 0.6, 0.8, 1.0}. We compare vanilla temperature sampling against RAND and MEAN baselines, as well as our proposed STARS method.

Table 8: Experimental results on `TESTEVAL` across different sampling temperatures. All the numbers are in percentages. Best-performing methods are highlighted in bold.

| | T | Method | Correctness ↑ | | | Overall cov. ↑ | | Line $cov@k$ ↑ | | Branch $cov@k$ ↑ | |
|---|---|---|---|---|---|---|---|---|---|---|---|
| | | | syntax | exe | assert | line | branch | $k=1$ | $k=5$ | $k=1$ | $k=5$ |
| QWEN3-1.7B | 1.0 | SAMPLING | 18.07 | 11.88 | 4.19 | 59.95 | 55.05 | 57.18 | 58.66 | 51.31 | 53.39 |
| | | RAND_0.1 | 17.93 | 12.12 | 4.57 | 56.87 | 52.23 | 54.15 | 55.40 | 48.45 | 50.25 |
| | | RAND_0.5 | 22.19 | 15.71 | 6.19 | 71.67 | 66.07 | 68.32 | 70.04 | 61.45 | 63.81 |
| | | MEAN_0.1 | 35.05 | 24.55 | 9.02 | 68.48 | 63.31 | 64.90 | 66.38 | 58.44 | 60.69 |
| | | MEAN_0.5 | **83.05** | **58.14** | 24.21 | 74.30 | 69.24 | 69.53 | 71.62 | 62.10 | 65.23 |
| | | STARS_0.1 | 61.10 | 39.90 | 19.17 | 86.77 | 81.10 | 80.54 | 83.62 | 71.71 | 76.36 |
| | | STARS_0.5 | 72.21 | 41.86 | **30.79** | **91.05** | **86.81** | **82.18** | **87.31** | **73.59** | **81.02** |
| | 0.8 | SAMPLING | 13.86 | 8.17 | 2.83 | 37.06 | 33.73 | 36.14 | 36.47 | 32.43 | 33.04 |
| | | RAND_0.1 | 15.52 | 10.05 | 3.69 | 47.98 | 43.74 | 46.11 | 47.17 | 41.03 | 42.58 |
| | | RAND_0.5 | 20.21 | 14.14 | 5.50 | 67.79 | 62.06 | 64.93 | 66.93 | 58.08 | 60.83 |
| | | MEAN_0.1 | 33.86 | 23.60 | 8.33 | 63.22 | 58.59 | 60.08 | 61.72 | 54.19 | 56.49 |
| | | MEAN_0.5 | **85.95** | **59.98** | 25.43 | 71.53 | 66.64 | 67.29 | 69.14 | 60.24 | 63.14 |
| | | STARS_0.1 | 61.10 | 39.90 | 19.17 | 86.77 | 81.10 | 80.54 | 83.62 | 71.71 | 76.36 |
| | | STARS_0.5 | 71.55 | 41.31 | **29.07** | **92.22** | **87.86** | **83.00** | **88.42** | **74.39** | **82.14** |
| | 0.6 | SAMPLING | 11.69 | 6.19 | 2.26 | 23.90 | 21.79 | 22.76 | 23.24 | 20.35 | 20.89 |
| | | RAND_0.1 | 13.43 | 7.93 | 2.88 | 42.20 | 38.38 | 41.11 | 41.70 | 36.80 | 37.73 |
| | | RAND_0.5 | 20.19 | 13.33 | 4.90 | 63.51 | 57.62 | 61.58 | 62.90 | 54.89 | 56.65 |
| | | MEAN_0.1 | 30.62 | 20.10 | 7.14 | 51.70 | 47.43 | 49.95 | 50.77 | 44.79 | 46.06 |
| | | MEAN_0.5 | **88.10** | **61.00** | 25.43 | 69.92 | 64.79 | 66.17 | 68.08 | 59.13 | 61.84 |
| | | STARS_0.1 | 61.57 | 41.10 | 19.43 | 86.84 | 80.95 | 81.08 | 83.99 | 72.24 | 76.53 |
| | | STARS_0.5 | 72.79 | 43.02 | **31.05** | **90.73** | **86.35** | **82.39** | **87.25** | **73.77** | **80.88** |
| | 0.4 | SAMPLING | 9.00 | 3.83 | 1.52 | 13.56 | 12.26 | 13.29 | 13.43 | 11.86 | 12.08 |
| | | RAND_0.1 | 11.74 | 6.36 | 2.57 | 30.53 | 27.76 | 29.53 | 29.81 | 26.40 | 26.81 |
| | | RAND_0.5 | 19.07 | 12.83 | 4.88 | 59.58 | 53.91 | 58.25 | 58.89 | 51.94 | 52.97 |
| | | MEAN_0.1 | 28.10 | 18.86 | 6.48 | 39.73 | 36.11 | 38.71 | 39.12 | 34.62 | 35.24 |
| | | MEAN_0.5 | **89.36** | **62.40** | 27.05 | 66.51 | 61.23 | 63.89 | 65.28 | 57.18 | 59.34 |
| | | STARS_0.1 | 62.69 | 41.60 | 19.74 | 87.14 | 81.17 | 81.82 | 84.46 | 73.15 | 77.13 |
| | | STARS_0.5 | 73.88 | 42.79 | **30.74** | **91.30** | **87.01** | **82.95** | **87.37** | **74.40** | **81.00** |
| | 0.2 | SAMPLING | 8.17 | 2.69 | 1.17 | 4.41 | 4.16 | 4.70 | 4.71 | 4.12 | 4.14 |
| | | RAND_0.1 | 10.74 | 5.17 | 2.05 | 24.70 | 22.24 | 24.06 | 24.42 | 21.39 | 21.89 |
| | | RAND_0.5 | 18.21 | 12.29 | 4.57 | 58.35 | 52.85 | 57.31 | 58.03 | 51.22 | 52.36 |
| | | MEAN_0.1 | 27.74 | 17.90 | 6.26 | 29.13 | 26.36 | 28.82 | 28.91 | 25.89 | 26.01 |
| | | MEAN_0.5 | **91.10** | **63.00** | 27.60 | 62.30 | 57.07 | 60.88 | 61.54 | 54.87 | 55.85 |
| | | STARS_0.1 | 62.93 | 41.11 | 18.81 | 86.25 | 80.22 | 80.67 | 83.60 | 71.72 | 75.92 |
| | | STARS_0.5 | 73.40 | 41.81 | **29.95** | **91.35** | **87.13** | **82.95** | **87.61** | **74.28** | **81.32** |

Table 9 shows the experimental results on the full `LiveIdeaBench` dataset with QWEN2.5-3B-INSTRUCT across different sampling temperatures T $\in$ {0.2, 0.4, 0.6, 0.8, 1.0}. We compare vanilla temperature sampling against RAND and MEAN baselines, as well as our proposed method, STARS. The activation is extracted at layer 20.

Table 9: Experimental results on full `LiveIdeaBench` across different sampling temperatures on Qwen2.5-3B-Instruct. All values are reported on a scale with a maximum score of 10. Best-performing methods are highlighted in bold.

| T | Method | Originality ↑ | Feasibility ↑ | Clarity ↑ | Fluency ↑ | Flexibility ↑ | Avg. ↑ |
|---|--------|--------------|--------------|-----------|-----------|--------------|--------|
| 1.0 | SAMPLING | **6.13** | 5.55 | 5.81 | 4.83 | 5.46 | 5.56 |
| | RAND_0.1 | 5.82 | **5.97** | 6.18 | 4.79 | 5.57 | 5.67 |
| | MEAN_0.1 | 5.90 | 5.92 | **6.20** | 4.93 | **5.67** | 5.73 |
| | STARS_0.1 | 5.88 | 5.91 | 6.19 | 5.12 | 5.65 | **5.75** |
| | STARS_0.5 | 5.65 | 5.92 | 6.05 | **5.43** | 5.67 | 5.74 |
| 0.8 | SAMPLING | **6.09** | 5.56 | 5.80 | 4.55 | 5.39 | 5.48 |
| | RAND_0.1 | 5.76 | **6.01** | 6.20 | 4.65 | 5.54 | 5.63 |
| | MEAN_0.1 | 5.86 | 5.94 | 6.22 | 4.75 | 5.56 | 5.67 |
| | STARS_0.1 | 5.83 | 5.96 | **6.24** | 4.84 | **5.59** | **5.69** |
| | STARS_0.5 | 5.56 | 5.94 | 6.06 | **5.27** | 5.57 | 5.68 |
| 0.6 | SAMPLING | **6.06** | 5.57 | 5.80 | 4.19 | 5.30 | 5.38 |
| | RAND_0.1 | 5.72 | **5.99** | 6.21 | 4.83 | **5.58** | **5.67** |
| | MEAN_0.1 | 5.81 | 5.93 | 6.22 | 4.30 | 5.45 | 5.54 |
| | STARS_0.1 | 5.79 | 5.96 | **6.23** | 4.57 | 5.52 | 5.61 |
| | STARS_0.5 | 5.51 | 5.97 | 6.08 | **5.01** | 5.50 | 5.62 |
| 0.4 | SAMPLING | 5.76 | 5.97 | 6.20 | 3.59 | 5.27 | 5.36 |
| | RAND_0.1 | 5.71 | **6.03** | 6.23 | 3.75 | 5.31 | 5.41 |
| | MEAN_0.1 | **5.83** | 5.92 | **6.24** | 3.68 | 5.30 | 5.39 |
| | STARS_0.1 | 5.75 | 6.01 | 6.22 | 4.30 | 5.45 | 5.55 |
| | STARS_0.5 | 5.48 | 6.01 | 6.10 | **4.96** | **5.55** | **5.62** |
| 0.2 | SAMPLING | 5.74 | 6.02 | 6.23 | 2.61 | 5.03 | 5.13 |
| | RAND_0.1 | 5.72 | **6.04** | 6.24 | 3.27 | 5.19 | 5.29 |
| | MEAN_0.1 | **5.83** | 5.94 | **6.25** | 2.71 | 5.05 | 5.16 |
| | STARS_0.1 | 5.75 | 5.97 | 6.22 | 3.89 | 5.35 | 5.44 |
| | STARS_0.5 | 5.44 | 5.99 | 6.08 | **4.77** | **5.44** | **5.54** |

Moreover, we compare our STARS method against nucleus sampling (Holtzman et al., 2020), a widely-used technique for controlling output diversity by dynamically truncating the sampling distribution. Table 10 shows results on `TESTEVAL` using Gemma-1.1-2b-it at temperature 0.8 with two nucleus sampling configurations (top_p = 0.9 and 0.95). While nucleus sampling achieves 92-93% syntax correctness, its line coverage reaches only 4-5%. In contrast, STARS achieves 35% line coverage and 30-31% branch coverage, though syntax correctness drops to 78-79%. These results demonstrate that traditional sampling techniques, such as nucleus sampling, which operate solely at the token-level distribution, are insufficient for tasks that require systematic exploration of diverse program behaviors. Our activation-based steering approach enables fundamentally different exploration capabilities by directly manipulating the model's internal representations.

Table 10: Experimental results on `TESTEVAL` to compare with nucleus sampling. All the numbers are in percentages. Best-performing methods are highlighted in bold.

| Top_p | Method | Correctness ↑ | | | Overall cov. ↑ | | Line $cov@k$ ↑ | | Branch $cov@k$ ↑ | |
|-------|--------|---------|------|--------|------|--------|-------|-------|-------|-------|
| | | syntax | exe | assert | line | branch | $k=1$ | $k=5$ | $k=1$ | $k=5$ |
| 0.9 | NUCLEUS SAMPLING | **93.45** | 1.19 | 1.12 | 4.15 | 3.70 | 4.10 | 4.14 | 3.59 | 3.66 |
| | STARS_0.5 | 78.81 | **3.93** | **3.55** | **35.73** | **31.62** | **34.25** | **34.88** | **29.79** | **30.47** |
| 0.95 | NUCLEUS SAMPLING | **92.57** | 1.21 | 1.19 | 5.09 | 4.88 | 5.07 | 5.09 | 4.73 | 4.84 |
| | STARS_0.5 | 79.36 | **4.40** | **3.74** | **35.69** | **31.65** | **34.13** | **34.99** | **29.47** | **30.65** |

## B.5 IMPACT OF TEMPERATURE ON STARS PERFORMANCE

We evaluate STARS's robustness to temperature variations by testing across a range from 0.2 to 2.0 on `TESTEVAL` using Gemma-1.1-2b-it (Table 11). Across all temperatures, temperature sampling achieves high syntax correctness but minimal coverage. While STARS achieves 32-40% line

coverage and 29-36% branch coverage, trading modest syntax correctness (78-80%) for substantial diversity.

Table 11: Experimental results on `TESTEVAL` across different sampling temperatures. All the numbers are in percentages. Best-performing methods are highlighted in bold.

| | T | Method | Correctness ↑ | | | Overall cov. ↑ | | Line $cov@k$ ↑ | | Branch $cov@k$ ↑ | |
|---|---|---|---|---|---|---|---|---|---|---|---|
| | | | syntax | exe | assert | line | branch | $k=1$ | $k=5$ | $k=1$ | $k=5$ |
| GEMMA-1.1-2B-IT | 2.0 | SAMPLING | **68.86** | 1.05 | 0.69 | 11.74 | 10.34 | 11.41 | 11.57 | 9.89 | 10.10 |
| | | STARS_0.1 | 68.71 | **1.19** | 1.07 | 13.23 | 11.91 | 13.00 | 13.18 | 11.53 | 11.82 |
| | | STARS_0.5 | 58.26 | 1.64 | **1.36** | **16.51** | **14.44** | **16.04** | **16.15** | **13.82** | **13.98** |
| | 1.8 | SAMPLING | 75.76 | 0.95 | 0.83 | 12.45 | 11.49 | 12.38 | 12.43 | 11.31 | 11.45 |
| | | STARS_0.1 | **78.00** | 0.93 | 0.81 | 11.86 | 10.71 | 11.81 | 11.86 | 10.54 | 10.68 |
| | | STARS_0.5 | 66.76 | **2.29** | **1.90** | **24.51** | **21.48** | **23.11** | **23.97** | **19.79** | **20.76** |
| | 1.6 | SAMPLING | **82.21** | 1.10 | 0.95 | 9.30 | 8.68 | 9.21 | 9.29 | 8.51 | 8.63 |
| | | STARS_0.1 | 80.90 | 1.26 | 1.07 | 11.84 | 10.85 | 11.72 | 11.79 | 10.54 | 10.71 |
| | | STARS_0.5 | 71.26 | **2.48** | **2.17** | **27.23** | **24.11** | **25.26** | **26.57** | **21.91** | **23.34** |
| | 1.4 | SAMPLING | 85.50 | 1.26 | 1.14 | 8.76 | 8.00 | 8.69 | 8.76 | 7.85 | 7.94 |
| | | STARS_0.1 | **83.64** | 1.45 | 1.31 | 13.91 | 13.31 | 13.82 | 13.90 | 13.04 | 13.25 |
| | | STARS_0.5 | 74.69 | **2.88** | **2.52** | **28.94** | **25.91** | **27.02** | **28.13** | **23.47** | **24.87** |
| | 1.2 | SAMPLING | **88.48** | 1.36 | 1.19 | 8.70 | 7.85 | 8.63 | 8.66 | 7.69 | 7.78 |
| | | STARS_0.1 | 85.29 | 1.43 | 1.26 | 11.32 | 10.43 | 10.99 | 11.28 | 9.89 | 10.33 |
| | | STARS_0.5 | 76.95 | **3.93** | **3.60** | **37.04** | **33.78** | **34.72** | **36.16** | **30.91** | **32.66** |
| | 1.0 | SAMPLING | **89.90** | 1.02 | 0.95 | 5.36 | 5.06 | 5.32 | 5.35 | 4.98 | 5.04 |
| | | STARS_0.1 | 86.48 | 1.43 | 1.17 | 10.03 | 9.32 | 9.97 | 10.02 | 9.12 | 9.25 |
| | | STARS_0.5 | 79.76 | **3.83** | **3.45** | **34.76** | **30.93** | **33.81** | **34.45** | **29.58** | **30.40** |
| | 0.8 | SAMPLING | **91.79** | 1.19 | 0.00 | 3.54 | 3.37 | 3.46 | 3.51 | 3.21 | 3.30 |
| | | STARS_0.1 | 87.83 | 1.45 | 1.12 | 10.77 | 9.88 | 10.64 | 10.72 | 9.65 | 9.78 |
| | | STARS_0.5 | 79.43 | **3.79** | **3.45** | **32.96** | **30.02** | **31.70** | **32.59** | **28.23** | **29.43** |
| | 0.6 | SAMPLING | **93.45** | 1.29 | 0.00 | 3.01 | 2.84 | 2.94 | 2.99 | 2.70 | 2.77 |
| | | STARS_0.1 | 89.19 | 1.71 | 1.43 | 12.10 | 11.22 | 12.03 | 12.10 | 10.99 | 11.15 |
| | | STARS_0.5 | 79.24 | **3.86** | **3.24** | **34.07** | **30.38** | **32.04** | **33.22** | **28.00** | **29.46** |
| | 0.4 | SAMPLING | **95.40** | 1.36 | 1.26 | 2.33 | 2.31 | 2.29 | 2.32 | 2.20 | 2.27 |
| | | STARS_0.1 | 90.43 | 1.48 | 1.43 | 8.44 | 7.69 | 8.41 | 8.44 | 7.58 | 7.63 |
| | | STARS_0.5 | 80.31 | **4.38** | **4.00** | **40.03** | **35.96** | **38.39** | **39.36** | **33.73** | **35.07** |
| | 0.2 | SAMPLING | **95.64** | 1.36 | 1.36 | 1.44 | 1.41 | 1.37 | 1.42 | 1.27 | 1.35 |
| | | STARS_0.1 | 91.52 | 1.45 | 1.36 | 7.28 | 6.50 | 7.23 | 7.25 | 6.36 | 6.40 |
| | | STARS_0.5 | 78.93 | **4.38** | **4.02** | **39.03** | **35.05** | **37.38** | **38.67** | **32.94** | **34.39** |

## B.6 STATISTICAL ANALYSIS ACROSS DIFFERENT RANDOM SEEDS

Table 12 shows the results on the full `LiveIdeaBench` dataset across different sampling temperatures on Qwen2.5-3B. Random seeds are 1, 2, and 42. The results demonstrate that STARS exhibits strong stability across different random seeds, as evidenced by small standard deviations (mostly $< 0.1$) for most metrics, and consistently outperforms baseline sampling across all temperature settings.

Table 12: Experimental results on `LiveIdeaBench` across sampling temperatures. Values represent mean ± std over three random seeds (1, 2, 42) for Qwen2.5-3B. All values are reported on a scale with a maximum score of 10. Best-performing methods are highlighted in bold.

| T | Method | Ori. ↑ | Feas. ↑ | Clar. ↑ | Flu. ↑ | Flex. ↑ | Avg. ↑ |
|---|---|---|---|---|---|---|---|
| 1.0 | SAMPLING | **6.01 ± 0.03** | **5.90 ± 0.06** | 6.15 ± 0.06 | 5.27 ± 0.12 | 5.73 ± 0.06 | 5.81 ± 0.05 |
| | STARS_0.1 | **6.01 ± 0.05** | 5.85 ± 0.04 | **6.17 ± 0.02** | 5.47 ± 0.18 | **5.79 ± 0.07** | **5.86 ± 0.04** |
| | STARS_0.5 | 5.75 ± 0.03 | 5.88 ± 0.04 | 6.05 ± 0.03 | **5.75 ± 0.13** | 5.73 ± 0.06 | 5.83 ± 0.04 |
| 0.8 | SAMPLING | 5.96 ± 0.01 | 5.95 ± 0.05 | 6.14 ± 0.01 | 5.00 ± 0.03 | 5.67 ± 0.04 | 5.74 ± 0.03 |
| | STARS_0.1 | **5.98 ± 0.06** | **5.91 ± 0.07** | **6.21 ± 0.03** | 5.27 ± 0.07 | **5.73 ± 0.02** | **5.82 ± 0.02** |
| | STARS_0.5 | 5.70 ± 0.07 | 5.86 ± 0.08 | 6.06 ± 0.01 | **5.69 ± 0.31** | 5.69 ± 0.08 | 5.80 ± 0.08 |
| 0.6 | SAMPLING | 5.88 ± 0.04 | **5.90 ± 0.04** | 6.15 ± 0.05 | 4.49 ± 0.21 | 5.52 ± 0.08 | 5.59 ± 0.06 |
| | STARS_0.1 | **5.93 ± 0.05** | 5.89 ± 0.03 | **6.21 ± 0.01** | 4.85 ± 0.26 | **5.59 ± 0.07** | **5.69 ± 0.07** |
| | STARS_0.5 | 5.60 ± 0.07 | 5.88 ± 0.03 | 6.03 ± 0.03 | **5.30 ± 0.12** | 5.57 ± 0.03 | 5.67 ± 0.04 |
| 0.4 | SAMPLING | 5.84 ± 0.04 | **5.98 ± 0.08** | **6.21 ± 0.02** | 3.50 ± 0.07 | 5.27 ± 0.02 | 5.36 ± 0.00 |
| | STARS_0.1 | **5.86 ± 0.03** | 5.97 ± 0.11 | **6.21 ± 0.09** | 4.67 ± 0.15 | **5.58 ± 0.00** | 5.66 ± 0.01 |
| | STARS_0.5 | 5.62 ± 0.04 | 5.94 ± 0.04 | 6.08 ± 0.00 | **5.18 ± 0.28** | 5.57 ± 0.06 | **5.68 ± 0.07** |
| 0.2 | SAMPLING | **5.83 ± 0.03** | **6.05 ± 0.04** | **6.26 ± 0.06** | 2.60 ± 0.07 | 5.05 ± 0.06 | 5.16 ± 0.03 |
| | STARS_0.1 | **5.83 ± 0.08** | 5.98 ± 0.06 | 6.25 ± 0.04 | 3.99 ± 0.26 | 5.41 ± 0.06 | 5.49 ± 0.06 |
| | STARS_0.5 | 5.56 ± 0.05 | 5.91 ± 0.04 | 6.07 ± 0.01 | **5.08 ± 0.15** | **5.54 ± 0.00** | **5.63 ± 0.02** |

### B.7 NUMERICAL COMPARISON OF ALGORITHM 2 AND ALGORITHM 3

This section presents a numerical comparison between the Riemannian gradient descent with line-search (Algorithm 2) and the one-step update algorithm (Algorithm 3) for solving the steering-vector computation problem (2) across different dimensional settings. We run Algorithm 2 for 100 iterations with hyperparameters $\rho = 0.2$, $c = 10^{-4}$ and $\bar{\eta} = 100$. We repeat all experiments for 50 times. For each run, we use the final loss of Algorithm 2 as a lower bound and compute the relative gap for both methods at every iteration. Figure 2 reports the average results across different dimensional settings. We also provide the computation time in Table 13. Notably, Algorithm 3 reduces the relative optimality gap from 7% to about 2% in a single update while using only about 3% of the runtime of Algorithm 2.

Table 13: Computational time of Algorithm 2 and Algorithm 3 (in seconds), average over 50 runs.

| $(d, N)$ | Algorithm 2 | Algorithm 3 |
|---|---|---|
| $(1024, 8)$ | 1.206 | 0.015 |
| $(1024, 20)$ | 3.299 | 0.024 |
| $(2048, 8)$ | 1.575 | 0.058 |
| $(2048, 20)$ | 4.322 | 0.103 |

### B.8 RUN-TIME COMPARISON

Table 14 reports the average per-question runtime across different models and tasks. The results show that Algorithm 3 introduces only minor absolute overhead (no more than 2 seconds per question on average) and modest relative overhead in most cases. Thus, the additional computational burden associated with Algorithm 3 appears negligible in practice.

Table 14: Runtime per question of baseline and Algorithm 3 (in seconds).

| Task | Model | Temperature Sampling | Algorithm 3 |
|---|---|---|---|
| Test Case Generation | GEMMA-1.1-2B-IT | 4.53 | 4.63 |
| | QWEN3-1.7B | 9.01 | 9.97 |
| Scientific Discovery | QWEN2.5-3B-INSTRUCT | 3.02 | 5.01 |
| | LLAMA-3.2-3B-INSTRUCT | 4.21 | 4.33 |

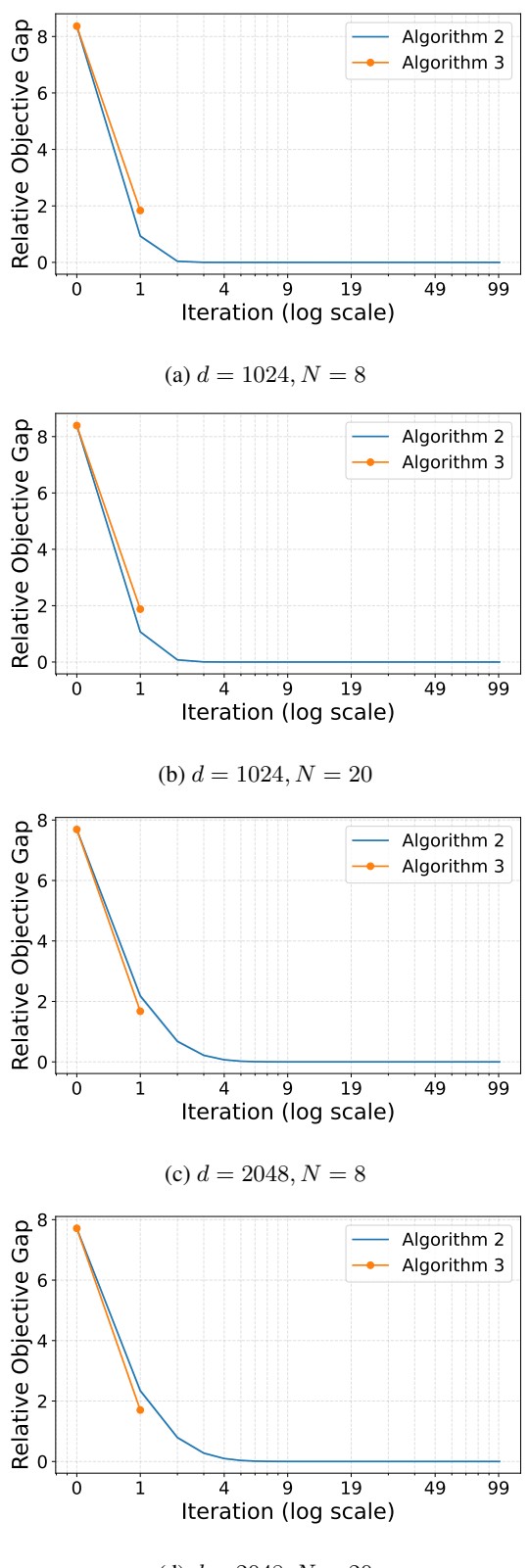

(a) $d = 1024, N = 8$

(b) $d = 1024, N = 20$

(c) $d = 2048, N = 8$

(d) $d = 2048, N = 20$

Figure 2: Optimality gap comparison of Algorithm 2 and Algorithm 3. The y-axis is the average relative gap (in percentage) across runs, and the x-axis denotes the iteration number on a logarithmic scale. The standard deviation is very small relative to the average relative gap and is therefore not visually apparent.

### B.9 CASE STUDY

Here we provide a generation example from `LiveIdeaBench` dataset with the keyword **periodic table**. The four paths of SAMPLING are

- Develop an interactive periodic table app that uses machine learning to predict chemical reactions based on user input of elements. This tool would enhance chemistry education by providing real-time predictions and explanations, making complex concepts more accessible and engaging for students.

- Develop an interactive periodic table app that allows users to visualize and manipulate chemical elements based on their electronic configurations. This tool would help educators and students better understand atomic structure and bonding principles through dynamic visualizations, potentially enhancing learning outcomes and engagement.

- Develop an interactive periodic table app that uses AI to predict chemical reactions based on user input of elements. This tool would enhance chemistry education by providing real-time reaction predictions, aiding students in understanding complex chemical interactions.

- Develop an interactive periodic table app that uses augmented reality to visualize chemical elements in real-world environments. This tool would enhance learning by allowing users to see how elements interact in their surroundings, potentially sparking new insights into chemistry and materials science.

The four generations from STARS are

- Designing a periodic table with quantum states as elements could revolutionize our understanding of chemical reactions and materials science. By mapping quantum states alongside traditional elements, we can predict new compounds and optimize energy storage solutions. This innovative approach would bridge the gap between quantum mechanics and chemistry, offering unprecedented insights into material properties and potential applications in renewable energy technologies. This concept aims to explore the relationship between quantum states and chemical elements, potentially leading to breakthroughs in materials science and energy technology.

- Develop an AI system that predicts chemical reactions using the periodic table as a foundational dataset. This system would analyze patterns and relationships between elements based on their positions in the periodic table, aiming to improve reaction prediction accuracy and potentially discover new chemical compounds.

- A periodic table app that allows users to visualize and interact with elements in real-time, enhancing understanding of chemical properties and periodic trends. This idea involves developing an educational tool for chemistry enthusiasts and students. By creating an interactive periodic table app, users can explore the relationships between elements, their atomic structures, and how they relate to each other based on their positions in the table. This could include features like real-time element animations, comparisons of physical properties, and visual representations of chemical reactions involving different elements. The app would also allow for customization, enabling users to choose specific elements or groups to study in depth.

- Utilizing the periodic table for personalized medicine by mapping genetic variations to corresponding medical treatments. This idea aims to revolutionize personalized medicine by creating a comprehensive map that links specific genetic variations with effective medical treatments, based on the periodic table's structure of elements and their properties. This could significantly improve treatment efficacy and reduce side effects by tailoring therapies to individual genetic profiles.

To summarize, temperature sampling collapses to near-duplicates of an "interactive periodic-table app with AI reaction prediction/AR" theme, whereas STARS yields four conceptually distinct but still topical and grammatical ideas: (i) a quantum-state–augmented periodic table for materials discovery; (ii) an AI reaction-prediction system grounded in periodic structure; (iii) an educational interactive table; and (iv) a (cross-domain) "periodic-table of personalized medicine" mapping genotypes to treatments.

### B.9.1 QUALITATIVE ANALYSIS OF A SIMPLE PROMPT 'A HOUSE'

Here we provide a simple example with the prompt 'A house'. We test with QWEN2.5-3B and temperature = 0.6. and max_tokens = 48.

With $N = 4$, the temperature sampling gives

- Sure! What would you like to know about a house? Here are some common questions and information related to houses: 1. **Types of Houses**: There are many types of houses including detached houses (single-family homes), semi-detached houses

- Sure! A house is a structure designed to provide shelter and living space for people or animals. Houses can vary greatly in size, style, and features depending on their intended use, location, and the preferences of the homeowner. Here are some

- Sure! I'd be happy to discuss various aspects of a house. What specifically would you like to know about it? Here are a few topics we could explore: 1. **Types of houses**: Different styles such as modern, traditional,

- Sure! A house is a structure designed for human habitation that typically includes rooms for living, sleeping, and sometimes eating or entertaining. Houses can vary greatly in size, style, and features depending on their location, the culture of the region

With $N = 8$, the temperature sampling gives

- A house is a structure designed to provide shelter and living space for humans or animals. It can vary greatly in size, style, and location, from simple cottages in rural areas to luxurious mansions in urban centers. Houses typically consist of

- A house is a structure designed to provide shelter for people or animals. It can vary greatly in size, style, and location, but generally includes rooms for living, sleeping, eating, and sometimes storage. Houses can be made of various materials

- A house is a structure designed for human habitation, typically consisting of one or more rooms and a roof to protect from weather conditions. Houses can vary greatly in size, style, and location, depending on the region, culture, and personal

- Sure, I can describe a typical house for you! A house generally consists of several key components: 1. **Structure**: This includes the foundation, walls, and roof. The foundation is what supports the entire structure and ensures it stays stable

- Sure, I can provide information about a house! Do you have a specific question or detail in mind? Here are some common topics related to houses: 1. Types of houses (e.g., single-family homes, apartments, townhouses)

- A house is a structure designed for human habitation. It typically includes rooms for living, sleeping, eating, and other activities. Houses can vary greatly in size, style, and location depending on the region and culture. They often include features

- Sure, I can provide information about a house! Do you have any specific questions or details in mind? For example: 1. What type of house are you referring to (e.g., single-family home, apartment, townhouse)? 2

- A house is a structure designed to provide shelter for people and their belongings. It can vary greatly in size, style, and location, but typically includes rooms for living, sleeping, eating, and other activities. Houses can be made from various

When $N = 4$, the STARS gives

- A house is a structure designed for human habitation. It can vary greatly in size, style, and location depending on the region and culture. Houses typically include rooms for living, sleeping, eating, and working, as well as facilities like

- Sure! I'd be happy to help describe a house. Could you please provide more details? For example, what type of house is it (modern, traditional, etc.), where is it located, or any other specific features you'd like

- A house is a structure or building that serves as a dwelling for people or other living beings. It can be made of various materials such as wood, brick, stone, and more recently, with the advancement of technology, also includes eco-friendly

- Sure! Could you please provide more details about the house? What specific information would you like to know or discuss about it? For example, do you want to know about its size, location, features, or history?

When $N = 8$, the STARS gives

- Hello! How can I assist you about a house? Do you want information on houses, house prices, or perhaps you need help with something related to building or decorating a house? Feel free to ask any specific questions you might have, and

- Sure! I can describe a house for you. Would you like details about a specific type of house or general information? If you have a particular question in mind, feel free to ask!

- Sure! A house is a structure where people live and can be made of various materials such as wood, brick, or even bamboo. It can also come in different sizes and styles. If you'd like more details about houses or have any

- A house is an object, a structure designed to provide shelter and comfort for individuals and families. Describing it more comprehensively, I can share some information: A house typically consists of walls, floors, a roof, and windows. It

- A house is more than just a structure; it's often a home where memories and stories are made. I can certainly help you generate ideas or facts about houses if you're interested in that topic. However, to be more specific, could

- Let me describe a house: A house is a structure where people live. It could be a small house or a large mansion. The materials used to build a house can vary widely, from wood, brick, stone, and other natural materials

- Sure, I can provide information about houses if you give me more details such as the location or type of house you're interested in. For example, are you looking for a specific type of house, or do you have any other questions?

- Sure. A house is a very good start for a conversation, but I need more context to provide a useful response. Could you clarify if this is what you wanted to ask about? If you have a specific question or need assistance with something

We now increase $C$ from 1.0 (as in the previous samples) to 3.0. The following are the 4-generation results of STARS with $C = 3.0$. With stronger steering, STARS yields more diverse responses, including different starting modes.

- Sure, I can describe a house for you. You want me to create a description or do something with the house? Or would you like to provide more details on what kind of house you have in mind? If you give me more context

- Here is some information about a house: Location: A house can be found in many places around the world, and you can choose to describe a specific location if needed. Size: A house can vary in size. I can give you more

- A house is an object which can refer to many things depending on the context. If you're thinking about a house in the context of "house and buildings," then there could be many houses around the world. If you're

- For example, let me describe a house: A house has four walls made of brick and mortar, each wall facing different directions. The roof over the house is made of tiles or metal sheets. Houses can vary in size, shape, and

### B.9.2 Uniform Distribution Recovery in Dice Rolling

We run the dice experiment with the following prompt:

> **Roll A Die Prompt**
>
> Roll a die.
> Output ONLY one integer from 1 to 6.
> Do not add any explanation or text.
> Output format: $< number >$

For each generation, the output logits are passed through softmax to obtain probabilities, and the probability P(token = d) for each digit is recorded by indexing the corresponding token ID. The experiment ran 10 iterations, each with 20 rolls (200 total samples). Table 15 presents the probability distributions obtained from temperature sampling and STARS. The KL column provides the KL divergence between the sampled distribution and the uniform distribution. Under standard temperature sampling, the model concentrates most of its probability mass on just two outcomes: 2 and 3. In contrast, as we increase the STARS strength from 3 to 7, the probability mass becomes much more spread out across all six faces, although it does not become fully uniform. We believe this behavior warrants further investigation.

Table 15: Probability distributions across dice outcomes (1-6) under temperature sampling and STARS with varying $C$. KL divergence measures the distance from the uniform distribution (lower is better).

|  | 1 | 2 | 3 | 4 | 5 | 6 | KL $\downarrow$ |
|---|---|---|---|---|---|---|---|
| Sampling | 0.000056 | 0.405526 | 0.450347 | 0.032215 | 0.111854 | 0.000001 | 0.71 |
| STARS_3.0 | 0.103587 | 0.452858 | 0.366496 | 0.023672 | 0.046169 | 0.007218 | 0.56 |
| STARS_5.0 | 0.380046 | 0.245687 | 0.108884 | 0.111879 | 0.103058 | 0.050445 | 0.20 |
| STARS_7.0 | 0.355143 | 0.130725 | 0.192331 | 0.184511 | 0.103763 | 0.033526 | 0.18 |

