# OpenReview forum: "Exploring Diverse Generation Paths via Inference-time Stiefel Activation Steering"
_ICLR.cc/2026/Conference — ICLR 2026 Poster_

### Official Review · Reviewer_a1Ka · 2025-10-19

**Soundness:** 3
**Presentation:** 2
**Contribution:** 2
**Rating:** 4
**Confidence:** 3

**Summary:**

This paper studies batch inference with the goal of maximizing diversity amongst the generated responses. The author's propose a training-free inference-time intervation method, cointed STAR, which modifies intermediate activations. Namely, for each of the N generations, STAR collects the hidden activations across each of the N generations and solves optimization problem to learn a set of perturbation vectors that when added to the activations, maximizes the volume of the corresponding set of N hidden states. The authors show that this optimization problem can be solved using a Riemannian gradient descent algorithm and provide convergence guarantees, along with a more loghtweight one-step update algorithm. Experiments using the one-step optimization algorithm across test-case generation and scientific discovery benchmarks show that this method outperforms standard sampling methods.

**Strengths:**

- The paper is well-written and organized. The authors do a good job of motivating the problem at hand, and I believe the problem studied is of interest to the ML community
- The authors complement their experimental results with theory, proving the feasibility of their proposed optimization procedure.

**Weaknesses:**

My biggest issue with this paper is its lack of clarity. I summarize my concerns below.

- **Some of the algorithmic details were unclear to me**. For a particular layer, are the hidden states for all the tokens modified or just one token? Is steering done only once on a pre-determined layer (with the modified hidden states propagated forward)? Or, do you do steering at several different layers? How do you pick the layers for steering?

- **The evaluation metrics are unclear to me**. In particular, the method proposed by the authors produces a collection of N responses. However, it seems to me that the metrics in Tables 1 and 2 are for specific responses? If so, how are these metrics aggregated across the N responses and shouldn't you be using a metric that evaluates the collection of N responses as a whole? Overall, the authors should make very clear whether the metrics in Table 1 and 2 are batch- or  individual-level. In my opinion, this is the biggest weakness, because the authors should be using a batch-level diversity metric (since that's what their method optimizes for) along with an individual-level quality metric.

My other concern is with the lack of comparisons to the related methods the authors summarize in Section 2. It would be great if the authors can explain why they didn't compare their method with these ones (and include this explanation in the final version).

**Questions:**

See Weaknesses.

---

> ### Author Response · Authors · 2025-11-22
> **Response to Reviewer a1Ka**
>
> We thank the reviewer for taking the time to evaluate our paper and provide positive feedback. We address the reviewer’s questions as follows.
>
> > Some of the algorithmic details were unclear to me.
>
> We steer **every decoding step** at a single chosen layer $l$, adding per‑head vectors $v^{l,j}$ at the attention‑head outputs before the FFN (see Eq. 1). The modified hidden state is propagated through the rest of the layer stack and the next token is sampled; at the next step we recompute steering vectors from the new hidden states and repeat until EOS.
>
> The steering layer is selected in advance, as shown in Appendix B.2. On a 10% validation slice, we evaluate candidate layers and rank them based on their average rank across correctness and coverage metrics. We then select the best average rank layer for full runs. The ranking results are shown in Tables 3 and 4.
>
>
>
>
> > The evaluation metrics are unclear to me.
>
> For TESTEVAL (Table 1), the coverage metrics (“overall coverage line/branch”, “line/branch cov@k”) are batch-level by construction: for each program, we consider the union of coverage over the N generated tests (or the top-k subset) and compute the fraction of lines/branches covered by at least one test. The correctness metrics (syntax/execution/assertion correctness) are computed per-test but averaged over the full set of generated tests, thereby summarizing the quality of the entire collection rather than a single “best” response.
>
> For LiveIdeaBench (Table 2), originality, feasibility, and clarity are scored per idea and then averaged to capture typical quality within the batch. Fluency, following [1], is explicitly a diversity measure: it is derived from GPT-4.1-mini’s judgments of how similar/different the multiple ideas are, and thus evaluates the collection rather than any single idea. Flexibility is computed from the distribution of the combined scores (averaging originality, feasibility, clarity, and fluency) across various keywords.
>
> [1] Ruan, Kai, et al. "LiveIdeaBench: Evaluating LLMs' Divergent Thinking for Scientific Idea Generation with Minimal Context."  arXiv:2412.17596 (2024).
>
>
>
> > My other concern is with the lack of comparisons to the related methods the authors summarize in Section 2.
>
> The following table compares our method STAR, with nucleus sampling[1] on the TESTEVAL dataset using Gemma-1.1-2b-it at temperature 0.8 with two nucleus sampling configurations (top p = 0.9 and 0.95). Generally, STAR outperforms the nucleus sampling across settings. We added the detailed results and discussions in Appendix B.4.
>
>
> | Top_p | Method           | Correctness syntax ↑ | Correctness exe ↑ | Correctness assert ↑ | Overall cov. line ↑ | Overall cov. branch ↑ | Line cov@1 ↑ | Line cov@5 ↑ | Branch cov@1 ↑ | Branch cov@5 ↑ |
> | ----- | ---------------- | -------------------- | ----------------- | -------------------- | ------------------- | --------------------- | ------------ | ------------ | -------------- | -------------- |
> | 0.9   | Nucleus sampling | **93.45**                | 1.19              | 1.12                 | 4.15                | 3.70                  | 4.10         | 4.14         | 3.59           | 3.66           |
> |       | STAR_0.5         | 78.81            | **3.93**          | **3.55**             | **35.73**           | **31.62**             | **34.25**    | **34.88**    | **29.79**      | **30.47**      |
> | 0.95  | Nucleus sampling | **92.57**                | 1.21              | 1.19                 | 5.09                | 4.88                  | 5.07         | 5.09         | 4.73           | 4.84           |
> |       | STAR_0.5         | 79.36            | **4.40**          | **3.74**             | **35.69**           | **31.65**             | **34.13**    | **34.99**    | **29.47**      | **30.65**      |
>
>
> [1] Ari Holtzman, Jan Buys, Li Du, Maxwell Forbes, and Yejin Choi. The curious case of neural text
> degeneration. arXiv preprint arXiv:1904.09751, 2020.

---

> > ### Comment · Reviewer_a1Ka · 2025-11-26
> >
> > I thank the authors for their response. My concerns have been addressed and I have increased my score accordingly.

---

### Official Review · Reviewer_ZBFD · 2025-10-27

**Soundness:** 3
**Presentation:** 3
**Contribution:** 3
**Rating:** 6
**Confidence:** 4

**Summary:**

This work proposes a method to increase diversity in LLM generations. The proposed STAR approach optimizes a steering vector at inference time, trying to maximize the volume spanned by $N$ tokens at the same timestep. By repeating this procedure at each decoding step, STAR obtains diverse sequences which still preserve good quality. The authors propose to obtain the steering vectors by means of Riemannian optimization, also deriving a fast 1-step approach alternative. The experiments show that STAR is more effective than standard temperature-based sampling.

**Strengths:**

**S1:** I believe that studying diversity is somehow lacking in the current research efforts. The applications of "diversity increase" are underexplored, and could lead to improvements of generative models both at inference (reduce bias, increase creativity, etc.) and at training time. This work tackles the topic at its core, which is refreshing.

**S2:** This work proposes to use steering to induce diversity. To the best of my knowledge, this is an unexplored application of steering, which I deeply appreciated. The proposed approach trying to maximize the volume, although having practical limitations, is of interest to the community and can spark new research on this topic.

**S3:** The approach using Riemannian optimization to maximize volume while preserving the manifold of intervention vectors is sensible and well explained. I have suggested some clarifications, but overall I am confident about the proposed approach.

**S4:** The text is well written and easy to follow. The mathematical notation is clean.

**Weaknesses:**

**W1:** STAR applies to a single layer of the model by construction. This is a fundamental drawback in my opinion. First, the best layer must be found in advance, as the authors have done in Tables 3,4. Second, previous work has shown that intervening carefully on all layers is more effective ([Rodriguez et al. NeurIPS 2025](https://arxiv.org/abs/2503.10679)). Additionally, while being a common choice, intervening at the output of attention layers is less effective than intervening on the residual path. The latter is typically the choice when the intervention is only applied to 1 layer. I believe more discussion about the layer choice (beyond the experiment in Tables 3,4) is required, pointing out the pros/cons of the choices made.

NB. Tables 3,4 are not referenced in the text.

**W2:** STAR uses the same scaling for all tokens. I understand the underlying reason for that, scaling being hard to tune or adapt per token. It is true that the scaling is "normalized" by $||H||_2^2$ but still _fixed_ and not adaptive to what the generation requires. Indeed, it might happen that tokens don't need to deviate from each other at a specific time step. Have the authors considered some adaptive steering such as MERA ([Hedström et al., ICML 2025](https://arxiv.org/abs/2510.13290))?

Additionally, could $\alpha$ be optimized together with $v_i$ in Eq. 2?

**W3:** No samples are provided. As a reader, I was expecting a subjective analysis of the sentences obtained using sampling temperature, and using STAR. Also how do these sentences differ when $N$ increases. How different, and at the same time correct, they are from a subjective point of view? I encourage the authors to share generation samples, and include them in the manuscript. For example, I am really curious to see what are the generations for simple prompts like _"A house"_ while increasing $N$.

**W4:** I encourage the authors to include generation timings of STAR compared to $N$ generations using standard temperature-based sampling. I believe this is a critical aspect that has been overlooked in the manuscript.

**W5:** The experimental section is limited. I understand that it is hard to find suitable experiments for diversity (given how underexplored this area is). I may suggest leveraging the comment on L48: _For safety and alignment, a lack of diversity prevents us from discovering varied failure modes_. I believe this is an important aspect to tackle with methods like STAR. I think this work would benefit from having an experiment showing how increasing diversity reduces bias, or shows better performance for minority groups (eg. talks about different genders with more parity).

---

### **Recommendations:**

> Please take the following as just recommendations, feel free to comment on them or pushback if you feel they are not justified.

**R1:** I suggest the authors to emphasize why Riemannian optimization is useful in this setting, and which is the manifold we are trying to _preserve_ while optimizing. The latter is defined by the the constraing $V^{\top}V=\alpha I$, which might not be evident as the text is now. For the former, I defer to the authors' to complete, but probably commenting on Riemannian opt. being faster and better behaved than using Euclidean gradients + projections in this setting.

**R2:** I also suggest the authors to run at least some test applying STAR at the residual path of some layer of choice, and compare to applying STAR at the attention output. With residual path I mean right after the sum with the residual connection. This is usually easier to capture as the input to the next Transformer block.

**Questions:**

**Q1:** One aspect that I would like to discuss with the authors is the fact that STAR tries to maximize the volume spanned by the activation vectors of the $\tau$-th token of $N$ sequences. Isn't this implying somehow that there is a temporal correspondence across sentences? Otherwise, always pushing for orthogonality and max volume at each timestep might lead to sub-optimal quality sequences, right? Do the authors have ideas on how to modify STAR so it takes the whole _trajectory_ (tokens for $t\leq \tau$) into account. Could this lead to more meaningful, and at the same time diverse, sequences?

**Q2:** L201: _To encourage diversity between different generations, we require the steering vectors to be orthogonal with each other_

Is this truly required? Do we need orthogonal tokens at each timestep to ensure diversity? For a bias related example, one could argue that diverse sentences are those that convey the same message, but using all possible choices of gender. This does not imply that all tokens must be orthogonal. Also, why would gender options be orthogonal? In my opinion, this assertion is very strong. Very interested in knowing the authors' arguments.

**Q3:** L203: _A too big $\alpha$ may break the meaningful information in $h_i$, leading to generation collapse._

Using an unbounded scaling parameter for vector-based addition has been shown to not respect the activation distributions. Recent work by  [Rodriguez et al. ICLR 2025](https://arxiv.org/abs/2410.23054v1) showed that respecting distributions is key, allowing interpolation between original and intervened (linearly mapped) activations, thus having a bounded scaling between 0 and 1. In further work,  [Rodriguez et al. NeurIPS 2025](https://arxiv.org/abs/2503.10679) show how to do that with gradient descent. I suggest considering a steering approach similar to the provided works, effectively avoiding the scaling problem. I completely understand that this is a fundamental change in your work, so I am not directly asking for this, but feel the authors should at least discuss on how these approaches could improve their work.

**Q4:** L206: In Eq.2 $\alpha$ is a parameter set by the user? It might be read as it is optimized jointly with $v_i$, which is not after reading the rest of the paper. Maybe I would clarify how $\alpha$ is chosen upfront.
Update: after reading the full manuscript, I see $\alpha$ is implicitly set by the user through $C$. I still believe that this aspect should be stated earlier on in the manuscript.

**Q5:** L224: In algorithm 1, $\alpha$ is required but never used. Consider removing it, or fixing the algorithm if $\alpha$ was not included by mistake.

**Q6:** The results in Table 2 show very close numbers across methods. Could the authors provide statistical significance for this experiment? Probably the standard deviation across 3-4 runs would be enough. This would help the reader understand the real benefit of each method.

---

> ### Author Response · Authors · 2025-11-22
> **Response to Reviewer ZBFD (Part 1)**
>
> We thank the reviewer for taking the time to evaluate our paper and provide positive feedback. We address the reviewer’s questions as follows.
>
> > STAR applies to a single layer of the model by construction. This is a fundamental drawback, in my opinion.
>
> We do not think that this is a drawback of our method.
>
> STAR takes a set of $N$ vectors as input, and it solves a Riemannian optimization on the Stiefel manifold to obtain steering vectors $V$ to maximize the log-determinant objective. STAR is a general and flexible framework: by changing three lines of code
>
> (i) We can easily hook to the residual stream (instead of the attention output);
>
> (ii) We can apply STAR to any combination of layers.
>
> Regarding intervening at multiple layers: this would incur additional compute overhead and increase the latency. Our aim in this work is to show that even a single well-chosen layer already yields strong gains in diversity with minimal overhead.
>
> The reviewer suggests intervening at the residual stream. We conducted experiments on the TESTEVAL dataset with the GEMMA‑1.1‑2B‑I model. Across different temperatures,  intervening at the attention output consistently outperforms intervening at the residual stream. The following table shows the results where  STAR_res denotes STAR applied to the residual stream. We added the detailed results and discussions in Appendix B.3.
>
> | T| Method | Correctness syntax ↑ | Correctness exe ↑ | Correctness assert ↑ | Overall cov. line ↑ | Overall cov. branch ↑ | Line cov@1 ↑ | Line cov@5 ↑ | Branch cov@1 ↑ | Branch cov@5 ↑ |
> | ---- | ------------ | -------------------- | ----------------- | -------------------- | ------------------- | --------------------- | ------------ | ------------ | -------------- | -------------- |
> | 1.0  | Sampling| **89.90**| 1.02| 0.00| 0.00| 0.00 | 0.00| 0.00| 0.00| 0.00|
> || STAR_res | 87.71| **6.02**| **5.93**| 33.54| 30.10| 31.54| 32.55| 27.64| 28.94|
> || STAR| 79.76| 3.83| 3.45| **34.76**| **30.93**| **33.81**| **34.45**| **29.58**| **30.40**|
> | 0.8  | Sampling| **91.79**| 1.19| 0.00| 1.44| 1.41 | 1.37| 1.42| 1.27| 1.35|
> || STAR_res | 87.71| **6.02**| **5.93**| **33.54**| **30.10**| 31.54| 32.55| 27.64| 28.94|
> || STAR| 79.43| 3.79| 3.45| 32.96| 30.02| **31.70**| **32.59**| **28.23**| **29.42**|
> | 0.6  | Sampling| **93.45**| 1.29| 0.00| 3.01| 2.84 | 2.94| 2.77| 2.70| 2.77|
> || STAR_res | 90.74| **6.52**| **6.36**| 33.23| 29.73| 31.09| 32.46| 26.94| 28.61|
> || STAR| 79.24| 3.86| 3.24| **34.07**| **30.38**| **32.04**| **33.22**| **28.00**| **29.46**|
> | 0.4  | Sampling| **95.40**| 1.36| 1.26| 2.33| 2.31 | 2.29| 2.32| 2.20| 2.27|
> || STAR_res | 90.83| **6.55**| **6.38**| 32.45| 29.32| 30.90| 31.77| 27.35| 28.60|
> || STAR| 80.31| 4.38| 4.00| **40.03**| **35.96**| **38.39**| **39.35**| **33.73**| **35.07**|
> | 0.2  | Sampling| **95.64**| 1.36| 1.36| 1.44| 1.41 | 1.37| 1.42 | 1.27| 1.35|
> || STAR_res | 90.67| **6.33**| **6.21**| 31.09| 27.62| 30.21| 30.60| 26.33| 26.92|
> || STAR| 78.93| 4.38 | 4.02| **39.03**| **35.05**| **37.38**| **38.67**| **32.94**| **34.39**|
>
> > Have the authors considered some adaptive steering, such as MERA (Hedström et al., ICML 2025)
>
> We appreciate the pointer to MERA (Hedström et al., ICML 2025)  and agree that adaptive steering is a promising direction.  However, integrating such a steering position detection model into our setting is non-trivial. First, MERA assumes access to supervised error signals and a calibration set with ground-truth labels. In contrast, STAR targets batch-level diversity, and there is no simple per-token “error” label to regress on. Secondly, STAR optimizes batch-level diversity across all $N$ generations, rather than per-token correctness, so a single-token linear probe is insufficient.
>
> To decide whether to increase or decrease the steering strength at time $\tau$, one would need a model that monitors the **joint state of all $N$ partial trajectories** and predicts the marginal gain in batch-level reward, which is closer in spirit to a process reward model (PRM) that scores entire reasoning paths or sets of outputs. Training such a PRM-like controller would require substantial task-specific supervision and tracing infrastructure, and would significantly increase both offline and online complexity.
>
> We therefore view adaptive steering position detection as a complementary extension. It is an interesting direction for future work, but orthogonal to the core contribution of STAR.

---

> ### Author Response · Authors · 2025-11-22
> **Response to Reviewer ZBFD (Part 2)**
>
> > Additionally, could $\alpha$ be optimized together with $v_{i}$ in Eq. 2?
>
> Generally speaking, this is a non-trivial task.  If we treat $\alpha$ as a free decision variable in Eq.2, then increasing $\alpha$ simply allows $V$  to grow in norm, and the log-det term $\log\det((H+V)^{\top}(H+V))$ can be made arbitrarily large by scaling $V$. Hence, the objective $-\log\det(\cdot)$ becomes unbounded below and the optimization is ill-posed.
>
> Properly modeling the effect of $\alpha$ on downstream metrics (e.g., coverage, correctness) would require differentiating through decoding and task evaluation, which is a non-trivial and highly task-dependent process.
>
>
>
> > No samples are provided.
>
> We provide a generation example from the LiveIdeaBench dataset with the keyword *periodic table* in Appendix B.10 of the revised paper. For the keyword periodic table, temperature sampling collapses to near‑duplicates of an “interactive periodic‑table app with AI reaction prediction/AR” theme, whereas STAR yields four conceptually distinct but still topical and grammatical ideas: (i) a quantum‑state–augmented periodic table for materials discovery; (ii) an AI reaction‑prediction system grounded in periodic structure; (iii) an educational interactive table; and (iv) a (cross‑domain) “periodic‑table of personalized medicine” mapping genotypes to treatments.
>
>
> As your suggestion, we also try with the prompt “A house” in Appendix B.10. Generally, STAR yields a more heterogeneous set of behaviors even for this simple prompt: some generations give compact definitional descriptions, others focus on materials or cultural variation, and several turn the prompt into a clarifying dialogue (“Could you please provide more details about the house?”, “Do you want information on houses, house prices, or perhaps you need help with something related to building or decorating a house? ”).
>
>
> > I encourage the authors to include generation timings of STAR compared to generations using standard temperature-based sampling.
>
> The following table shows the comparison of average runtime over questions. Overall, Algorithm 3 incurs only small absolute overheads ($\leq 2$ seconds on average) and modest relative overheads for most settings, indicating it does not substantially increase runtime. We added the results and discussions in Table 3 of Section 6.3.
>
> |Task|  Model  | Temperature Sampling | Alg.3 |
> |:--------------------:|:-------------------:|:--------------------:|:----------------------------:|
> | Test Case Generation |gemma-1.1-2b-it|4.53| 4.63 |
> | |Qwen3-1.7B  |9.01| 9.97 |
> | Scientific Discovery | Qwen2.5-3B-Instruct |3.02| 5.01 |
> | | Llama-3.2-3B-Instruct|4.21|4.33|
> >  The experimental section is limited. [...] I think this work would benefit from having an experiment showing how increasing diversity reduces bias, or shows better performance for minority groups (eg. talks about different genders with more parity).
>
> We fully agree that diversity can and should be studied in the context of safety, bias, and minority-group performance, and that this is an important direction for future work. However, our current paper does **not** aim to address group fairness or demographic bias directly: we focus on **diversity in model behavior** (e.g., exploring different test cases for code and distinct ideas in idea generation), as captured by coverage-style and quality metrics on TESTEVAL and LiveIdeaBench. These benchmarks do not include demographic or group labels, and designing a rigorous setup to evaluate gender or minority parity would require dedicated datasets, annotation protocols, and fairness metrics that are orthogonal to the main geometric and algorithmic contribution of STAR and beyond the scope of this work.  To avoid overclaiming, we therefore restrict our empirical analysis to diversity as “coverage of behaviors/solutions” rather than as “parity across social groups”.
>
>
> > I suggest that the authors emphasize why Riemannian optimization is useful in this setting, and which manifold we are trying to preserve while optimizing.
>
> The idea of ``using Euclidean gradients + projections” suffers a critical drawback: stepping along the Euclidean gradient is fine, but the subsequent projection can actually increase the objective. As a consequence, the classical convex projected gradient descent theory does not apply; and we lack a clean global convergence guarantee. Even local convergence is hard to achieve: the projection can distort steps enough to slow or even stall convergence unless the step size is extremely small.
>
> The Riemannian optimization framework addresses the above concerns:
> - First, we have a sufficient decrease in the objective value in each iteration step (see Proposition 5 in Appendix A).
> - Second, Riemannian gradient descent has convergence guarantees since it follows the true steepest-descent direction on the manifold with consistent steps [see Theorem 1].

---

> ### Author Response · Authors · 2025-11-22
> **Response to Reviewer ZBFD (Part 3)**
>
> > I also suggest that the authors run at least some tests applying STAR at the residual path of some layer of choice, and compare to applying STAR at the attention output.
>
> We evaluate the effect of our intervention when applied to two loci in the network—(i) the residual stream and (ii) the attention output. The experiments are on the TESTEVAL dataset with the GEMMA‑1.1‑2B‑I model. The following table shows the results where STAR_res denotes STAR applied to the residual stream. Across conditions, STAR_res consistently outperforms temperature sampling, indicating that intervening on the residual pathway yields measurable gains. We added the detailed results and discussions in Appendix B.3.
>
>
> | T | Method | Correctness syntax ↑ | Correctness exe ↑ | Correctness assert ↑ | Overall cov. line ↑ | Overall cov. branch ↑ | Line cov@1 ↑ | Line cov@5 ↑ | Branch cov@1 ↑ | Branch cov@5 ↑ |
> | ---- | ------------ | -------------------- | ----------------- | -------------------- | ------------------- | --------------------- | ------------ | ------------ | -------------- | -------------- |
> |1.0|Sampling|**89.90**|1.02|0.00|0.00|0.00|0.00|0.00|0.00|0.00|
> ||STAR_res|87.71|**6.02**|**5.93**|33.54|30.10|31.54|32.55|27.64|28.94|
> ||STAR|79.76|3.83|3.45|**34.76**|**30.93**|**33.81**|**34.45**|**29.58**|**30.40**|
> |0.8|Sampling|**91.79**|1.19|0.00|1.44|1.41|1.37|1.42|1.27|1.35|
> ||STAR_res|87.71|**6.02**|**5.93**|**33.54**|**30.10**|31.54|32.55|27.64|28.94|
> ||STAR|79.43|3.79|3.45|32.96|30.02|**31.70**|**32.59**|**28.23**|**29.42**|
> |0.6|Sampling|**93.45**|1.29|0.00|3.01|2.84|2.94|2.77|2.70|2.77|
> ||STAR_res|90.74|**6.52**|**6.36**|33.23|29.73|31.09|32.46|26.94|28.61|
> ||STAR|79.24|3.86|3.24|**34.07**|**30.38**|**32.04**|**33.22**|**28.00**|**29.46**|
> |0.4|Sampling|**95.40**|1.36|1.26|2.33|2.31|2.29|2.32|2.20|2.27|
> ||STAR_res|90.83|**6.55**|**6.38**|32.45|29.32|30.90|31.77|27.35|28.60|
> ||STAR|80.31|4.38|4.00|**40.03**|**35.96**|**38.39**|**39.35**|**33.73**|**35.07**|
> |0.2|Sampling|**95.64**|1.36|1.36|1.44|1.41|1.37|1.42|1.27|1.35|
> ||STAR_res|90.67|**6.33**|**6.21**|31.09|27.62|30.21|30.60|26.33|26.92|
> ||STAR|78.93|4.38|4.02|**39.03**|**35.05**|**37.38**|**38.67**|**32.94**|**34.39**|
>
>
>
>
>
> > Do the authors have ideas on how to modify STAR so it takes the whole trajectory (tokens for $t \leq \tau$) into account.
>
> Because of causal masking in autoregressive transformers, the hidden state $h_{\tau,i}$ at $\tau$-th token of $i$-th path is computed from the prefix tokens, so it already summarizes that entire prefix.  Thus, even though our objective at time $\tau$ only sees activations of $\tau$-th token, these vectors already encode the full history of each trajectory up to $\tau-1$.  This local steering accumulates into trajectory-level diversity.
>
> We agree that explicitly optimizing a trajectory-level objective (e.g., based on path summaries rather than per-step activations) is a promising extension.  A direct way to do this is to include earlier states in the volume objective by stacking them with the current step. Concretely, we can replace Eq. 2 with
> $$
> \min_{ V^\top V = \alpha I} -\log\det ( [H_{1:\tau-1}, H_{\tau}+V_{\tau}]^\top [H_{1:\tau-1}, H_{\tau}+V_{\tau}] ),
> $$
> where $H_{1:\tau-1}$ the past hidden states from all runs up to $\tau-1$. One can also develop algorithms to solve the problem above, following a similar procedure to that presented in the current paper.
>
>
> > Do we need orthogonal tokens at each timestep to ensure diversity? [...] Also, why would gender options be orthogonal?
>
> We clarify that the diversity is geometrically captured by the log-determinant objective, which measures the volume spanned by the steered activations. The constraint does not ensure diversity; however, it imposes that the modifications to the activations are orthogonal to each other, which geometrically intuitively inflates the volume.
>
>
> > I suggest considering a steering approach similar to the provided works, effectively avoiding the scaling problem.
>
> STAR has a different aim from ACT/LinEAS in Rodriguez et al. ICLR 2025 and Rodriguez et al. NeurIPS 2025.  ACT/LinEAS target **controlled convergence to a known concept** (e.g., toxic to non‑toxic)  with bounded interpolation and preserved distributions.  STAR aims at **exploration under tight per‑token latency with no target distribution**, and it does not have a distributional alignment objective toward a predefined target.
>
> STAR and ACT/LinEAS have different learning patterns.  ACT/LinEAS requires offline source/target datasets and/or end‑to‑end optimization to estimate the OT mapping per layer. That assumption does not align with our goal of training-free exploration and would require substantial precomputation that must be amortized across prompts.
>
> We thank the reviewer for pointing us to Rodriguez et al. ICLR 2025 and Rodriguez et al. NeurIPS 2025. We have discussed these papers in Section 1 of the revised paper.

---

> ### Author Response · Authors · 2025-11-22
> **Response to Reviewer ZBFD (Part 4)**
>
> > In Eq.2, $\alpha$ is a parameter set by the user?
>
> Clearly state, at the introduction of Eq.2, that $\alpha$ is a **user-controlled hyperparameter** that fixes the squared norm of each steering vector. We will state this point clearly in the revised version.
>
>
>
> > L224: In algorithm 1, $\alpha$ is required but never used. Consider removing it, or fixing the algorithm if $\alpha$ was not included by mistake.
>
> Thank you for catching this. Algorithm 1 should return a scaled matrix so that $V_{0}^{\top}V_{0}=\alpha I$. Concretely, Line 4 of Alg. 1 should return $V_0=\sqrt{\alpha}[v_1,\dots,v_N]$. Our implementation utilizes this scaled version; the omission of $\sqrt{\alpha}$ in the pseudocode is a typo.
>
>
> > The results in Table 2 show very close numbers across methods. Could the authors provide statistical significance for this experiment?
>
>
> The following table shows the results on the LiveIdeaBench dataset across different sampling temperatures on Qwen2.5-3B. Random seeds are 1, 2, and 42. The results demonstrate that STAR exhibit strong stability across different random seeds, as evidenced by small standard deviations (mostly $<$ 0.1) for most metrics, and consistently outperform baseline sampling across all temperature settings. We have discussed this in Appendix B.6.
>
> | T | Method | Ori. ↑ | Feas. ↑ | Clar. ↑ | Flu. ↑ | Flex. ↑ | Avg. ↑ |
> |---|---|---|---|---|---|---|---|
> | 1.0 | Sampling | **6.01 ± 0.03** | **5.90 ± 0.06** | 6.15 ± 0.06 | 5.27 ± 0.12 | 5.73 ± 0.06 | 5.81 ± 0.05 |
> |  | STAR_0.1 | **6.01 ± 0.05** | 5.85 ± 0.04 | **6.17 ± 0.02** | 5.47 ± 0.18 | **5.79 ± 0.07** | **5.86 ± 0.04** |
> |  | STAR_0.5 | 5.75 ± 0.03 | 5.88 ± 0.04 | 6.05 ± 0.03 | **5.75 ± 0.13** | 5.73 ± 0.06 | 5.83 ± 0.04 |
> | 0.8 | Sampling | 5.96 ± 0.01 | 5.95 ± 0.05 | 6.14 ± 0.01 | 5.00 ± 0.03 | 5.67 ± 0.04 | 5.74 ± 0.03 |
> |  | STAR_0.1 | **5.98 ± 0.06** | **5.91 ± 0.07** | **6.21 ± 0.03** | 5.27 ± 0.07 | **5.73 ± 0.02** | **5.82 ± 0.02** |
> |  | STAR_0.5 | 5.70 ± 0.07 | 5.86 ± 0.08 | 6.06 ± 0.01 | **5.69 ± 0.31** | 5.69 ± 0.08 | 5.80 ± 0.08 |
> | 0.6 | Sampling | 5.88 ± 0.04 | **5.90 ± 0.04** | 6.15 ± 0.05 | 4.49 ± 0.21 | 5.52 ± 0.08 | 5.59 ± 0.06 |
> |  | STAR_0.1 | **5.93 ± 0.05** | 5.89 ± 0.03 | **6.21 ± 0.01** | 4.85 ± 0.26 | **5.59 ± 0.07** | **5.69 ± 0.07** |
> |  | STAR_0.5 | 5.60 ± 0.07 | 5.88 ± 0.03 | 6.03 ± 0.03 | **5.30 ± 0.12** | 5.57 ± 0.03 | 5.67 ± 0.04 |
> | 0.4 | Sampling | 5.84 ± 0.04 | **5.98 ± 0.08** | **6.21 ± 0.02** | 3.50 ± 0.07 | 5.27 ± 0.02 | 5.36 ± 0.00 |
> |  | STAR_0.1 | **5.86 ± 0.03** | 5.97 ± 0.11 | **6.21 ± 0.09** | 4.67 ± 0.15 | **5.58 ± 0.00** | 5.66 ± 0.01 |
> |  | STAR_0.5 | 5.62 ± 0.04 | 5.94 ± 0.04 | 6.08 ± 0.00 | **5.18 ± 0.28** | 5.57 ± 0.06 | **5.68 ± 0.07** |
> | 0.2 | Sampling | **5.83 ± 0.03** | **6.05 ± 0.04** | **6.26 ± 0.06** | 2.60 ± 0.07 | 5.05 ± 0.06 | 5.16 ± 0.03 |
> |  | STAR_0.1 | **5.83 ± 0.08** | 5.98 ± 0.06 | 6.25 ± 0.04 | 3.99 ± 0.26 | 5.41 ± 0.06 | 5.49 ± 0.06 |
> |  | STAR_0.5 | 5.56 ± 0.05 | 5.91 ± 0.04 | 6.07 ± 0.01 | **5.08 ± 0.15** | **5.54 ± 0.00** | **5.63 ± 0.02** |

---

> > ### Comment · Reviewer_ZBFD · 2025-11-26
> > **Answer to Rebuttal**
> >
> > Let me thank you for the detailed answers to my questions!
> > I would like to emphasize the following:
> > * The provided samples do clarify how STAR actually generates diversity, which in my opinion was required since “as a user” I would like to understand why STAR would be interesting.
> >   * The “A house” experiment is insightful. It seems (by only eye-balling of course) that STAR with $N=4$ generates 2 modes of responses. 2 starting with “A house is a structure” and 2 starting with “Sure!“. I would have expected 4 very distinct answers. Could the authors comment on this aspect? Is is possible that the convergence parameter $c$ in Algorithm 2 is too relaxed and volume maximization was early stopped? Or that the learning step is too high and the optimization fell in a local minimum?
> > * About the residual application of STAR. Thanks for running this. Any intuition on why the residual intervention yields better “correctness” $\forall T$, but the coverage is lower (although pretty close at $T=1$) ?
> > * I understand the authors’ answer related to the focus of the paper on behavioral diversity and not on model fairness, and the work it would require to incorporate such benchmarks might go beyond the scope of a rebuttal. I have though one suggestion for the authors, not required ofc. One could prompt STAR with “Roll a dice and tell me the result”. If an LLM is sampled iid, nothing “forces” the LLM to produce a uniform distribution of outcomes. But with STAR, such distribution should be closer to uniform? I believe some toy/synthetic task like this one could be easily ran and add some interesting side insights to your work.
> > * Let me clarify my comment on using an approach like ACT for example. I was not implying a comparison with ACT, but rather suggesting to use an affine transform in STAR (instead of vector addition) and an interpolation strength (instead of an unbounded scaling), similar to what ACT or LinEAS do.
> >
> > Looking forward to reading your thoughts about these extra comments.

---

> ### Author Response · Authors · 2025-11-28
> **Re: Answer to Rebuttal**
>
> We genuinely appreciate the depth of your feedback and have enjoyed engaging with your comments. The following are our responses to your further comments.
>
> > The “A house” experiment is insightful. [...] Could the authors comment on this aspect?
>
> We first note that, in practice, we employ Algorithm 3 rather than Algorithm 2. Because Algorithm 3 is a lightweight, single-step update procedure, it does not guarantee global convergence.
>
> Although the beginnings of the previous 4 responses fall into 2 modes, the semantic meanings of the 4 responses are not the same.
>
> We now increase $C$ from 1 (as in the previous samples) to 3. The following are the 4-generation results of STAR with $C=3.0$. With stronger steering, STAR provides more diverse responses than before, including different starting modes.
>
> - Sure, I can describe a house for you. You want me to create a description or do something with the house? Or would you like to provide more details on what kind of house you have in mind? If you give me more context
> - Here is some information about a house: Location: A house can be found in many places around the world, and you can choose to describe a specific location if needed. Size: A house can vary in size. I can give you more
> - A house is an object which can refer to many things depending on the context. If you're thinking about a house in the context of "house and buildings," then there could be many houses around the world. If you're
> - For example, let me describe a house: A house has four walls made of brick and mortar, each wall facing different directions. The roof over the house is made of tiles or metal sheets. Houses can vary in size, shape, and
>
> > Any intuition on why the residual intervention yields better “correctness” $\forall T$ , but the coverage is lower (although pretty close at $T=1$) ?
>
>
> We agree this is primarily an empirical finding, and we do not yet have a full mechanistic explanation. The following is our intuition. Steering the attention output directly changes the relative weights assigned to different tokens in the context, so for code it can effectively put more uniform or redistributed attention on different parts of the program under test. This appears to help the model explore different branches/lines (improving coverage), but it also perturbs the local token-level decisions more aggressively, which can slightly hurt the correctness rate.
>
>
> > One could prompt STAR with “Roll a dice and tell me the result”. If an LLM is sampled iid, nothing “forces” the LLM to produce a uniform distribution of outcomes. But with STAR, such distribution should be closer to uniform?
>
>
> We run the dice experiment with the following prompt:
>  ``Roll a dice.\n Output ONLY one integer from 1 to 6.\n Do not add any explanation or text.\n Output format: <number>``
> For each generation, the output logits are extracted and converted to probabilities using softmax, and the probability P(token = d) for each digit $ d \in \{1, 2, 3, 4, 5, 6 \}$ is recorded by indexing the corresponding token ID. The experiment run 10 iterations of 20 rolls each (200 total samples). The resulting table displays the probability distributions from temperature sampling and STAR. The *KL* column provides the KL divergence between the sampled distribution and the uniform distribution.
>
> Under standard temperature sampling, the model’s probability mass is heavily concentrated on just two outcomes 2 and 3. In contrast, as we increase the STAR strength from 3 to 7, the probability mass becomes much more spread out across all six faces, although it does not become fully uniform. We believe this is an interesting behavior that merits further investigation.
>
>
> |          | 1 | 2  | 3   | 4 | 5 | 6 | KL |
> |----------|---|----|-----|---|---|---|---|
> | Sampling | 0.000056 | 0.405526 | 0.450347 | 0.032215 | 0.111854 | 0.000001 |  0.71|
> | STAR_3.0 | 0.103587 | 0.452858 | 0.366496 | 0.023672 | 0.046169 | 0.007218 |0.56|
> | STAR_5.0 | 0.380046 | 0.245687 | 0.108884 | 0.111879 | 0.103058 | 0.050445 |0.20|
> | STAR_7.0 | 0.355143 | 0.130725 | 0.192331 | 0.184511 | 0.103763 | 0.033526 | 0.18|
>
>
> > Let me clarify my comment on using an approach like ACT for example. I was not implying a comparison with ACT, but rather suggesting to use an affine transform in STAR (instead of vector addition) and an interpolation strength (instead of an unbounded scaling), similar to what ACT or LinEAS do.
>
>
> In practice, the steering strength $\alpha$ should be bounded to prevent the model from collapsing or severely degrading. To identify a safe range, one can first build a dataset by collecting activations (and corresponding generations) across and marking those settings where the model collapses. Given the subset of safe activations, one can then formulate an optimization problem (e.g., compute the convex hull of these safe points) to estimate a conservative upper bound $\alpha_{\max}$ that STAR should never exceed.

---

### Official Review · Reviewer_BKyG · 2025-10-28

**Soundness:** 2
**Presentation:** 2
**Contribution:** 2
**Rating:** 2
**Confidence:** 2

**Summary:**

The paper explores methods for steering large language model activations at inference time, with the goal of diversifying reasoning trajectories.
The work is situated within the area of activation and steering vectors for LLMs.
The authors propose adding a learned steering vector to the output of an attention head,
with the vector being initialized and subsequently updated to minimize the objective described in Equation 2.

In Section 4, the paper introduces an initialization approach based on singular value decomposition (SVD),
followed by refinement using Riemannian gradient descent.
This combination allows the model to identify meaningful steering directions while maintaining stability in optimization.
In Section 5, the authors further propose a simplified, single-step variant of the method, aimed at improving computational efficiency.

The experimental results compare the proposed approach against a single diffusion-based baseline,
although the specific diffusion method used is not clearly specified.
Overall, the paper presents an interesting and technically sound contribution to the growing field of activation steering,
though the evaluation setup could benefit from additional baselines and clarity regarding the comparison method.

**Strengths:**

- The paper addresses an important and timely topic—enhancing diversity in large language model reasoning.
- The idea of using training-free steering vectors is particularly interesting, as it offers a lightweight and potentially generalizable approach to influencing model behavior without fine-tuning.

**Weaknesses:**

- The evaluation is somewhat limited, relying primarily on a single “sampling” method as the baseline. Comparing only against one approach makes it difficult to assess the broader effectiveness of the proposed method.
- Additionally, the reported results, while suggestive, are not particularly strong or conclusive. A more comprehensive experimental section would strengthen the paper’s empirical claims.

**Questions:**

- Could the authors include comparisons with other existing steering methods to better contextualize their approach?
- Additionally, have the authors considered testing their method against alternative diffusion-based approaches to provide a more complete evaluation?

---

> ### Author Response · Authors · 2025-11-22
> **Response to Reviewer  BKyG**
>
> We thank the reviewer for taking the time to evaluate our paper and provide positive feedback. We address the reviewer’s questions as follows.
>
> > Could the authors include comparisons with other existing steering methods to better contextualize their approach?
>
> Current steering methods are mostly for a specific target (e.g, toxic→non‑toxic), so we compare STAR with two heuristic steering baselines.  See Appendix B.4 in the revised paper.
>
> The following table compares our method STAR with nucleus sampling[1] on the TESTEVAL dataset with the model. STAR outperforms the nucleus sampling under different settings.
>
> | Top_p| Method|Correctness syntax ↑| Correctness exe ↑ | Correctness assert ↑ | Overall cov. line ↑ | Overall cov. branch ↑ | Line cov@1 ↑ | Line cov@5 ↑ | Branch cov@1 ↑| Branch cov@5 ↑|
> | ----- | ---------------- | -------------------- | ----------------- | -------------------- | ------------------- | --------------------- | ------------ | ------------ | -------------- | -------------- |
> |0.9| Nucleus sampling |93.45|1.19|1.12| 4.15| 3.70 | 4.10| 4.14| 3.59| 3.66|
> | |STAR_0.5| **78.81**|**3.93**|**3.55**|**35.73**| **31.62**| **34.25**| **34.88**|**29.79**|**30.47**|
> |0.95|Nucleus sampling|92.57|1.21|1.19|5.09|4.88|5.07|5.09|4.73|4.84|
> | |STAR_0.5|**79.36**|**4.40**| **3.74**|**35.69**|**31.65**| **34.13**|**34.99**|**29.47**|**30.65**|
>
>
> [1] Ari Holtzman, Jan Buys, Li Du, Maxwell Forbes, and Yejin Choi. The curious case of neural text
> degeneration. arXiv preprint arXiv:1904.09751, 2020.
>
>
> > Additionally, have the authors considered testing their method against alternative diffusion-based approaches to provide a more complete evaluation?
>
> We are aware of diffusion-style generative methods and agree they are promising for improving diversity in models that are trained as diffusion processes. However, our setting is quite different: STAR is designed as a training-free, plug-and-play steering method for standard autoregressive decoder-only LLMs. Thus, we focus our comparisons on alternative sampling and steering strategies within the same autoregressive framework.

---

### Official Review · Reviewer_D4n2 · 2025-10-28

**Soundness:** 2
**Presentation:** 4
**Contribution:** 3
**Rating:** 4
**Confidence:** 3

**Summary:**

The paper proposes an inference-time method (STAR) to diversify LLM generation by steering hidden activations from multiple concurrent decoding runs such that they maximise the geometric volume spanned by them.

The authors present a Riemannian gradient-descent solution with convergence guarantees to find the optimal steering vectors according to this objective. Based on the insight that it is too computationally heavy for acceptable latency, they propose a lightweight one-step update as approximation for real-time decoding.

The empirical evaluation is on TESTEVAL (test-case generation) using Gemma-1.1-2b-it and QWEN3-1.7B, as well as on LiveIdeaBench (scientific idea generation) using QWEN2.5-3B-INSTRUCT. STAR is compared to temperature sampling as a baseline for temperatures varying between 0.2 and 1.0. STAR significantly outperforms this baseline in these evaluations.

**Strengths:**

- The paper is very well and pedagogically written
- The method is an inference-time method and as such training free, saving compute overhead and making it accessible
- The theoretical part of the paper is strong, from guaranteeing existence of a solution to providing an algorithm for finding the optimal solution and deriving convergence guarantees
- The paper proposes a practical algorithm to approximate the algorithm guaranteed to find the optimal solution with a more lightweight approach for real-time low latency decoding
- Compared to the baseline that is included in the paper, the proposed method performs strongly

Minor:
- I appreciate the comment in lines 214 to 216 on the realism of the assumption necessary in Proposition 1.

**Weaknesses:**

- One key element of the method, namely the constraint on $V$ to be an orthogonal matrix, is not motivated too well. Only in line 201 it briefly says “To encourage diversity between different generations, we require the steering vectors to be orthogonal with each other, i.e. […]”.  But why does this constraint ensure diversity? Is it not rather the orthogonality of the columns of the resulting $H + V$ that would maximise diversity and the objective?
- In the experiments, the comparison to baselines is weak.
    - The only baseline the proposed method is compared against in the experiments is temperature sampling. At the bare minimum, the 2 baselines/ablations that I would need to see to lean towards accept would be 1) adding random vectors of same magnitude $\alpha$ (how much better than random additions are the directions you find?) 2) $v_j=(h_j - M_j * mean(H, dim=1))*s$, where $mean$ across $dim=1$ corresponds to mean across the $N$ vectors and $M_j$ is $-1$ when $H - mean$ is negative in said dimension and $+1$ when $H-mean$ is positive, while $s$ is scaling $v_j$ to be of magnitude $\alpha$ (how much better than just pushing away from the mean is the proposed steering direction).   Ideally, it would also be good to see comparisons against other methods mentioned in lines 52-62, in particular also to a training based method among those mentioned in lines 59 - 60.  Another good baseline would be to compare to the objective of maximising variance between vectors $h_j$ rather than the volume.
    - The only baseline that the model is compared against (temperature sampling) is only exploring temperatures between 0.2 and 1.0, even though in Table 1 for QWEN3-1.7B (all metrics) and in Table 2 for QWEN2.5-3B-INSTRUCT (all metrics except feasibility) have a positive trend for increasing temperature (in absolute terms and in relative terms to the proposed method). Higher temperatures should thus be tried to see if similar performance can be achieved through this.
- Also, in the experiments only one model is evaluated for Section 6.2. As someone having worked on activation steering myself I have found that the effectiveness of steering methods can vary a lot by model family so would always advise to evaluate on at least 2 different models from different model families. Furthermore, weirdly enough the one model evaluated in Section 6.2 is different from the 2 models evaluated in Section 6.1, raising doubts about what the results would have looked like for the models evaluated on the other task.
- Given that the authors put an emphasis on the need for the one-step update for smaller latency, a comparison of added latency in Section 6 would be good to characterise the trade-off between improved performance and (latency) cost of the proposed method over simple baselines such as temperature sampling

Minor:
- I would suggest using a different acronym than STAR, since there is a fairly well-known paper aiming to improve LLM generations with an almost identical acronym (*STaR: Self-Taught Reasoner - Bootstrapping Reasoning With Reasoning* by Zelikman et al. (2022))
- Typo: Line 217 is basically a repetition of lines 215/216
- Line 266/267: a small explanation why sufficient decrease prevents rank-deficiency would be helpful
- More depth could be added to the discussion in Section 6.1, e.g. by commenting on the performance of the baseline (temperature sampling), e.g. why it catastrophically fails at the task for Gemma-1.1-2b-it (incl. why higher temperature does not increase coverage but rather decreases it after T=0.6) or by explaining the u-shaped performance of the proposed method wrt temperature for most metrics for Gemma-1.1-2b-it

**Questions:**

- Line 61: “[…] and their benefits can be fragile across domains” -> Source?
- Line 66: “If multiple runs occupy nearly the same region in this space, surface-level stochasticity has little impact” -> Source?
- Figure 1: How do the particularities of ending the generations work? Does one of the N candidate sequences have to output a EOS token or all of them simultaneously or is a sequence outputting the EOS token simply removed from the algorithm until every sequence eventually outputs an EOS token?
- Section 3.1: Have the authors tried to other locations for their intervention, e.g, the residual stream that Panickssery et al. (2023) intervene on?
- Section 3.2: How is the pre-determined layer $l$ chosen?
- Algorithm 1, requirements: what is $\mathbb{R}_{++}$?
- Section 4: why do the authors resort to riemannian gradient descent if the retraction step is still necessary to ensure $V_{k+1}$ is feasible? Could you not do some sort of projected gradient descent using the Euclidean gradient directly, i.e. moving a step along the Euclidean gradient and then projecting back to the manifold of feasible solutions?
- Theorem 1: is a Riemannian gradient of 0 a necessary condition for a minimum of the objective function? Maybe a small comment on this for readers unfamiliar with Riemannian gradient descent could be helpful

---

> ### Author Response · Authors · 2025-11-22
> **Response to Reviewer D4n2 (Part 1)**
>
> We thank the reviewer for taking the time to evaluate our paper and provide positive feedback. We address the reviewer’s  questions as follows.
>
> > One key element of the method, namely the constraint on $V$  to be an orthogonal matrix, is not motivated too well. Only in line 201 it briefly says “To encourage diversity between different generations, we require the steering vectors to be orthogonal with each other, i.e. […]”.
>
> We clarify that the diversity is geometrically captured by the log-determinant objective, which measures the volume spanned by the steered activations. The constraint does not ensure diversity; however, this constraint imposes that the modifications of the activations are orthogonal to each other, which is geometrically intuitive to inflate the volume.  We agree our current wording could create confusion, and we revised this sentence as follows:
> `` The orthogonal constraint regularizes the steering vectors $v_i$, ensuring each run receives a distinct direction and preventing the trivial solution where all $v_i$ align.”
>
>
>
> > The only baseline the proposed method is compared against in the experiments is temperature sampling.
>
> We compare the two baselines you mentioned on the task TESTEVAL ,  random ( adding random vectors of same magnitude) and MEAN ( $v_j=(h_j-M_j * mean(H,dim=1))*s$).  The following table shows the experimental results of two different models. Generally, our methods still achieve better performance.
>
> - GEMMA-1.1-2B-it
> | T | Method | Correctness syntax ↑ | Correctness exe ↑ | Correctness assert ↑ | Overall cov. line ↑ | Overall cov. branch ↑ | Line cov@1 ↑ | Line cov@5 ↑ | Branch cov@1 ↑ | Branch cov@5 ↑ |
> |---|---|---|---|---|---|---|---|---|---|---|
> | 1.0 | RAND_0.1 | 89.81 | 1.14 | 1.10 | 4.89 | 4.54 | 4.83 | 4.88 | 4.40 | 4.45 |
> |  | RAND_0.5 | 89.98 | 1.07 | 0.98 | 6.48 | 6.14 | 6.44 | 6.46 | 6.05 | 6.10 |
> |  | MEAN_0.1 | 90.64 | 1.10 | 1.02 | 4.96 | 4.56 | 4.92 | 4.95 | 4.44 | 4.51 |
> |  | MEAN_0.5 | **90.76** | 1.12| 1.07|4.51|4.20| 4.43| 4.49|4.00|4.13|
> | |STAR_0.5|79.76|**3.83**|**3.45**|**34.76**|**30.93**|**33.81**|**34.45**|**29.58**|**30.40**|
> |0.2|RAND_0.1|96.24|1.33|1.31| 1.89 | 1.88 | 1.84 | 1.87 | 1.77 | 1.81 |
> |  | RAND_0.5 | 95.86|1.21|1.14| 1.89 | 1.88 | 1.83 | 1.88 | 1.77 | 1.85 |
> |  | MEAN_0.1 | 96.57|1.45|1.40| 1.89 | 1.88 | 1.87 | 1.88 | 1.77 | 1.81 |
> |  | MEAN_0.5 | **96.74** |1.29|1.29|1.41|1.36|1.41 | 1.41 | 1.31 | 1.33 |
> |  | STAR_0.5 | 78.93 | **4.38** | **4.02** | **39.03** | **35.05** | **37.38** | **38.67** | **32.94** | **34.39** |
>
>
> - QWEN3-1.7B
>
> | T | Method | Correctness syntax ↑ | Correctness exe ↑ | Correctness assert ↑ | Overall cov. line ↑ | Overall cov. branch ↑ | Line cov@1 ↑ | Line cov@5 ↑ | Branch cov@1 ↑ | Branch cov@5 ↑ |
> |---|---|---|---|---|---|---|---|---|---|---|
> | 1.0 | RAND_0.1 | 17.93 | 12.12 | 4.57 | 56.87 | 52.23 | 54.15 | 55.40 | 48.45 | 50.25 |
> |  | RAND_0.5 | 22.19 | 15.71 | 6.19 | 71.67 | 66.07 | 68.32 | 70.04 | 61.45 | 63.81 |
> |  | MEAN_0.1 | 35.05 | 24.55 | 9.02 | 68.48 | 63.31 | 64.90 | 66.38 | 58.44 | 60.69 |
> |  | MEAN_0.5 | **83.05** | **58.14** | 24.21 | 74.30 | 69.24 | 69.53 | 71.62 | 62.10 | 65.23 |
> |  | STAR_0.5 | 72.21 | 41.86 | **30.79** | **91.05** | **86.81** | **82.18** | **87.31** | **73.59** | **81.02** |
> | 0.2 | RAND_0.1 | 10.74 | 5.17 | 2.05 | 24.70 | 22.24 | 24.06 | 24.42 | 21.39 | 21.89 |
> |  | RAND_0.5 | 18.21 | 12.29 | 4.57 | 58.35 | 52.85 | 57.31 | 58.03 | 51.22 | 52.36 |
> |  | MEAN_0.1 | 27.74 | 17.90 | 6.26 | 29.13 | 26.36 | 28.82 | 28.91 | 25.89 | 26.01 |
> |  | MEAN_0.5 | **91.10** | **63.00** | 27.60 | 62.30 | 57.07 | 60.88 | 61.54 | 54.87 | 55.85 |
> |  | STAR_0.5 | 73.40 | 41.81 | **29.95** | **91.35** | **87.13** | **82.95** | **87.61** | **74.28** | **81.32** |
>
>
>
> The following table compares our method STAR with nucleus sampling[1] on TESTEVAL dataset using Gemma-1.1-2b-it at temperature 0.8 with two nucleus sampling configurations (top p = 0.9 and 0.95). Generally, STAR outperforms the nucleus sampling across setting. We added the detailed results and discussions in Appendix B.4.
>
> | Top_p | Method | Correctness syntax ↑ | Correctness exe ↑ | Correctness assert ↑ | Overall cov. line ↑ | Overall cov. branch ↑ | Line cov@1 ↑ | Line cov@5 ↑ | Branch cov@1 ↑ | Branch cov@5 ↑ |
> |---|---|---|---|---|---|---|---|---|---|---|
> | 0.9 | Nucleus sampling | 93.45 | 1.19 | 1.12 | 4.15 | 3.70 | 4.10 | 4.14 | 3.59 | 3.66 |
> |  | STAR_0.5 | **78.81** | **3.93** | **3.55** | **35.73** | **31.62** | **34.25** | **34.88** | **29.79** | **30.47** |
> | 0.95 | Nucleus sampling | 92.57 | 1.21 | 1.19 | 5.09 | 4.88 | 5.07 | 5.09 | 4.73 | 4.84 |
> |  | STAR_0.5 | **79.36** | **4.40** | **3.74** | **35.69** | **31.65** | **34.13** | **34.99** | **29.47** | **30.65** |
>
> [1] Ari Holtzman, Jan Buys, Li Du, Maxwell Forbes, and Yejin Choi. The curious case of neural text
> degeneration. arXiv preprint arXiv:1904.09751, 2020.

---

> ### Author Response · Authors · 2025-11-22
> **Response to Reviewer D4n2 (Part 2)**
>
> >  Higher temperatures should thus be tried to see if similar performance can be achieved through this.
>
> The following tables show the results of higher temperatures. STAR outperforms temperature sampling across all settings. We added the detailed results and discussions in Appendix B.5.
>
>
> | T | Method | Correctness syntax ↑ | Correctness exe ↑ | Correctness assert ↑ | Overall cov. line ↑ | Overall cov. branch ↑ | Line cov@1 ↑ | Line cov@5 ↑ | Branch cov@1 ↑ | Branch cov@5 ↑ |
> |---|---|---|---|---|---|---|---|---|---|---|
> | 2.0 | Sampling | **68.86** | 1.05 | 0.69 | 11.74 | 10.34 | 11.41 | 11.57 | 9.89 | 10.10 |
> | 2.0 | STAR_0.1 | 68.71 | **1.19** | 1.07 | 13.23 | 11.91 | 13.00 | 13.18 | 11.53 | 11.82 |
> | 2.0 | STAR_0.5 | 58.26 | 1.64 | **1.36** | **16.51** | **14.44** | **16.04** | **16.15** | **13.82** | **13.98** |
> | 1.8 | Sampling | 75.76 | 0.95 | 0.83 | 12.45 | 11.49 | 12.38 | 12.43 | 11.31 | 11.45 |
> | 1.8 | STAR_0.1 | **78.00** | 0.93 | 0.81 | 11.86 | 10.71 | 11.81 | 11.86 | 10.54 | 10.68 |
> | 1.8 | STAR_0.5 | 66.76 | **2.29** | **1.90** | **24.51** | **21.48** | **23.11** | **23.97** | **19.79** | **20.76** |
> | 1.4 | Sampling | 85.50 | 1.26 | 1.14 | 8.76 | 8.00 | 8.69 | 8.76 | 7.85 | 7.94 |
> | 1.4 | STAR_0.1 | **83.64** | 1.45 | 1.31 | 13.91 | 13.31 | 13.82 | 13.90 | 13.04 | 13.25 |
> | 1.4 | STAR_0.5 | 74.69 | **2.88** | **2.52** | **28.94** | **25.91** | **27.02** | **28.13** | **23.47** | **24.87** |
> | 1.0 | Sampling | **89.90** | 1.02 | 0.95 | 5.36 | 5.06 | 5.32 | 5.35 | 4.98 | 5.04 |
> | 1.0 | STAR_0.1 | 86.48 | 1.43 | 1.17 | 10.03 | 9.32 | 9.97 | 10.02 | 9.12 | 9.25 |
> | 1.0 | STAR_0.5 | 79.76 | **3.83** | **3.45** | **34.76** | **30.93** | **33.81** | **34.45** | **29.58** | **30.40** |
> | 0.6 | Sampling | **93.45** | 1.29 | 0.00 | 3.01 | 2.84 | 2.94 | 2.99 | 2.70 | 2.77 |
> | 0.6 | STAR_0.1 | 89.19 | 1.71 | 1.43 | 12.10 | 11.22 | 12.03 | 12.10 | 10.99 | 11.15 |
> | 0.6 | STAR_0.5 | 79.24 | **3.86** | **3.24** | **34.07** | **30.38** | **32.04** | **33.22** | **28.00** | **29.46** |
> | 0.2 | Sampling | **95.64** | 1.36 | 1.36 | 1.44 | 1.41 | 1.37 | 1.42 | 1.27 | 1.35 |
> | 0.2 | STAR_0.1 | 91.52 | 1.45 | 1.36 | 7.28 | 6.50 | 7.23 | 7.25 | 6.36 | 6.40 |
> | 0.2 | STAR_0.5 | 78.93 | **4.38** | **4.02** | **39.03** | **35.05** | **37.38** | **38.67** | **32.94** | **34.39** |
>
>
>
>
> > Also, in the experiments, only one model is evaluated for Section 6.2.
>
> We further test our method STAR with Llama-3.2-3B-Instruct on LiveIdeaBench dataset. The following table shows the results, where STAR still outperforms the temperature sampling generally. We added the results in Table 2 of Section 6.2.
>
> | Temp. | Method   | Originality ↑ | Feasibility ↑ | Clarity ↑ | Fluency ↑ | Flexibility ↑ | Avg. ↑   |
> | ----- | -------- | ------------- | ------------- | --------- | --------- | ------------- | -------- |
> | 1.0   | Sampling | 6.51          | **5.34**      | **5.89**  | 5.41      | 5.66          | 5.76     |
> |       | STAR_0.5 | **6.60**      | 5.32          | 5.85      | **5.60**  | **5.71**      | **5.82** |
> | 0.8   | Sampling | 6.47          | **5.39**      | **5.92**  | 5.07      | 5.58          | 5.69     |
> |       | STAR_0.5 | **6.56**      | 5.36          | 5.91      | **5.22**  | **5.62**      | **5.73** |
> | 0.6   | Sampling | 6.43          | 5.42          | **5.99**  | 4.59      | 5.46          | 5.58     |
> |       | STAR_0.5 | **6.53**      | **5.42**      | 5.97      | **4.81**  | **5.58**      | **5.66** |
> | 0.4   | Sampling | 6.41          | 5.42          | **6.00**  | 4.17      | 5.35          | 5.47     |
> |       | STAR_0.5 | **6.50**      | **5.44**      | 5.99      | **4.41**  | **5.48**      | **5.56** |
> | 0.2   | Sampling | 6.42          | 5.45          | **6.03**  | 3.27      | 5.19          | 5.27     |
> |       | STAR_0.5 | **6.51**      | **5.46**      | 6.02      | **4.04**  | **5.39**      | **5.48** |
>
>
>
>
> > Given that the authors put an emphasis on the need for the one-step update for smaller latency, a comparison of added latency in Section 6 would be good to characterise the trade-off between improved performance and (latency) cost of the proposed method over simple baselines such as temperature sampling
>
>
> The following table shows the comparison of the average runtime over questions. Overall, Alg.3 incurs only small absolute overheads ($\leq 2$ seconds on average) and modest relative overheads for most settings, indicating it does not substantially increase runtime. We added the detailed results and discussions in Table 3 of Section 6.3.
>
> | Task | Model | Temperature Sampling | Alg.3 |
> |:---:|:---:|:---:|:---:|
> | Test Case Generation | gemma-1.1-2b-it | 4.53 | 4.63 |
> |  | Qwen3-1.7B | 9.01 | 9.97 |
> | Scientific Discovery | Qwen2.5-3B-Instruct | 3.02 | 5.01 |
> | | Llama-3.2-3B-Instruct|4.21|4.33|

---

> ### Author Response · Authors · 2025-11-22
> **Response to Reviewer D4n2 (Part 3)**
>
> >I would suggest using a different acronym than STAR, since there is a fairly well-known paper aiming to improve LLM generations with an almost identical acronym (STaR: Self-Taught Reasoner - Bootstrapping Reasoning With Reasoning by Zelikman et al. (2022))
>
> Thank you for pointing this out. To avoid confusion with STaR (Zelikman et al., 2022), we will rename our method to STARS (**St**iefel-based **A**ctivation Steering for Diverse **R**ea**s**oning).  For consistency with the submitted manuscript, we keep the original name STAR throughout the rebuttal.
>
>
> >Typo: Line 217 is basically a repetition of lines 215/216
>
> We removed the redundant sentence at that location and carefully proofread the surrounding paragraph.
>
>
>
> >Line 266/267: a small explanation why sufficient decrease prevents rank-deficiency would be helpful
>
> The objective function $\ell(V)=-\log\det((H+V)^{\top}(H+V))$ is finite if and only if $H+V$ is full-rank. We initialize $V_0$(Proposition 1), hence $\ell(V_0)$ is finite. The backtracking line search (line 8 of Alg.2) enforces a decrease in function value, so the entire sequence $\lbrace V_k \rbrace$ stays inside the sublevel set $\lbrace V : \ell(V) \le \ell(V_0) \rbrace$, where $H+V$ is guaranteed to remain full-rank.
>
>
>
>
>
> >More depth could be added to the discussion about the temperature sampling in Section 6.1
>
>
> Thank you for pointing this out. The temperature-sampling baseline results for T = 0.8 and T = 1.0 were not properly updated in the submission table. The unexpected trends you observed were caused by these outdated values. We have now confirmed that the corrected curves follow the expected pattern. We will update the manuscript accordingly, including Tables in Section 6.1. We sincerely apologize for the confusion and appreciate your careful reading. The following is the updated results of temperature-sampling baseline results for Gemma-1.1-2b-it.
>
> | T | Method | Correctness syntax ↑ | Correctness exe ↑ | Correctness assert ↑ | Overall cov. line ↑ | Overall cov. branch ↑ | Line cov@1 ↑ | Line cov@5 ↑ | Branch cov@1 ↑ | Branch cov@5 ↑ |
> |---|---|---|---|---|---|---|---|---|---|---|
> | 1.0 | Sampling | **89.90** | 1.02 | 0.95 | 5.36 | 5.06 | 5.32 | 5.35 | 4.98 | 5.04 |
> | 0.8 | Sampling | **91.79** | 1.19 | 0.00 | 3.54 | 3.37 | 3.46 | 3.51 | 3.21 | 3.30 |
> | 0.6 | Sampling | **93.45** | 1.29 | 0.00 | 3.01 | 2.84 | 2.94 | 2.99 | 2.70 | 2.77 |
> | 0.4 | Sampling | **95.40** | 1.36 | 1.26 | 2.33 | 2.31 | 2.29 | 2.32 | 2.20 | 2.27 |
> | 0.2 | Sampling | **95.64** | 1.36 | 1.36 | 1.44 | 1.41 | 1.37 | 1.42 | 1.27 | 1.35 |
>
>
> >Line 61: “[…] and their benefits can be fragile across domains” -> Source?
>
> [1] emphasize that each model has a different optimal point along the diversity spectrum, and highlight the “model-dependent performance peak” and “parameter sensitivity” of diversity settings across tasks and models, suggesting that non-optimal diversity choices can easily lead to worse performance. In their baseline comparison, they note that several sophisticated data selection methods (including diversity-aware ones) achieve strong gains on some benchmarks but suffer from catastrophic performance drops in other domains.
>
> [1] Zhenqing Ling, Daoyuan Chen, Liuyi Yao, Qianli Shen, Yaliang Li, and Ying Shen. Diversity
> as a reward: Fine-tuning LLMs on a mixture of domain-undetermined data. In The Thirty-
> Ninth Annual Conference on Neural Information Processing Systems, 2025.
>
>
>
>
>
> >“If multiple runs occupy nearly the same region in this space, surface-level stochasticity has little impact” -> Source?
>
> [1] shows that with standard repeated sampling, outputs are similar, trapped in a local cluster. They then show that prompt perturbations which encourage candidate solutions to “span multiple clusters” in the response space reduce textual similarity across samples and significantly improve Pass@k. This directly supports the intuition that, when all runs explore nearly the same region of the solution space, additional stochastic sampling mostly yields paraphrases rather than genuinely new solutions.
> Moreover, [2] systematically compares standard parallel (ancestral) sampling with two non-independent strategies (enumeration and iterative sampling). They observe that ancestral sampling, where each response is drawn independently from the same prompt and decoding hyperparameters, suffers from “repetitions of high-probability sequences.”
> We added these references to the revised paper.
>
> [1] Tianchun Wang, Zichuan Liu, Yuanzhou Chen, Jonathan Light, Haifeng Chen, Xiang Zhang,
> and Wei Cheng. Diversified sampling improves scaling LLM inference. arXiv preprint arXiv:2502.11027, 2025
>
> [2] Sergey Troshin, Irina Saparina, Antske Fokkens, and Vlad Niculae. Asking a language model for
> diverse responses. In Proceedings of the 2nd Workshop on Uncertainty-Aware NLP (UncertaiNLP
> 2025), pp. 66–72, 2025

---

> ### Author Response · Authors · 2025-11-22
> **Response to Reviewer D4n2 (Part 4)**
>
> > Figure 1: How do the particularities of ending the generations work?
>
> At each token, we form the activation matrix $H$ from active sequences only; sequences that emit EOS are dropped from the active set. We resolve the remaining columns in each step until all sequences terminate.
>
>
> > Section 3.1: Have the authors tried to other locations for their intervention, e.g, the residual stream that Panickssery et al. (2023) intervene on?
>
> STAR takes a set of $N$ vectors as input, and it solves a Riemannian optimization on the Stiefel manifold to obtain steering vectors $V$ to maximize the log-determinant objective. STAR is general and flexible framework: by changing three lines of code
> (i) we can easily hook to the residual stream (instead of the attention output);
> (ii) we can apply STAR at any combination of layers
>
> We also  evaluate the effect of our intervention when applied to the residual stream. The experiments are on TESTEVAL dataset with the GEMMA‑1.1‑2B‑I model. The following table shows the results, where STAR_res denotes STAR applied to the residual stream. Across different temperatures, STAR_res consistently outperforms temperature sampling, indicating that intervening on the residual pathway yields measurable gains. This shows that STAR is robust to the intervention position. We added the detailed results and discussions in Appendix B.3.
>
> | T | Method | Correctness syntax ↑ | Correctness exe ↑ | Correctness assert ↑ | Overall cov. line ↑ | Overall cov. branch ↑ | Line cov@1 ↑ | Line cov@5 ↑ | Branch cov@1 ↑ | Branch cov@5 ↑ |
> |---|---|---|---|---|---|---|---|---|---|---|
> | 1.0 | Sampling | **89.90** | 1.02 | 0.00 | 0.00 | 0.00 | 0.00 | 0.00 | 0.00 | 0.00 |
> |  | STAR_res_0.1 | 87.71 | **6.02** | **5.93** | 33.54 | 30.10 | 31.54 | 32.55 | 27.64 | 28.94 |
> |  | STAR_0.5 | 79.76 | 3.83 | 3.45 | **34.76** | **30.93** | **33.81** | **34.45** | **29.58** | **30.40** |
> | 0.8 | Sampling | **91.79** | 1.19 | 0.00 | 1.44 | 1.41 | 1.37 | 1.42 | 1.27 | 1.35 |
> |  | STAR_res_0.1 | 87.71 | **6.02** | **5.93** | **33.54** | **30.10** | 31.54 | 32.55 | 27.64 | 28.94 |
> |  | STAR_0.5 | 79.43 | 3.79 | 3.45 | 32.96 | 30.02 | **31.70** | **32.59** | **28.23** | **29.42** |
> | 0.6 | Sampling | **93.45** | 1.29 | 0.00 | 3.01 | 2.84 | 2.94 | 2.77 | 2.70 | 2.77 |
> |  | STAR_res_0.1 | 90.74 | **6.52** | **6.36** | 33.23 | 29.73 | 31.09 | 32.46 | 26.94 | 28.61 |
> |  | STAR_0.5 | 79.24 | 3.86 | 3.24 | **34.07** | **30.38** | **32.04** | **33.22** | **28.00** | **29.46** |
> | 0.4 | Sampling | **95.40** | 1.36 | 1.26 | 2.33 | 2.31 | 2.29 | 2.32 | 2.20 | 2.27 |
> |  | STAR_res_0.1 | 90.83 | **6.55** | **6.38** | 32.45 | 29.32 | 30.90 | 31.77 | 27.35 | 28.60 |
> |  | STAR_0.5 | 80.31 | 4.38 | 4.00 | **40.03** | **35.96** | **38.39** | **39.35** | **33.73** | **35.07** |
> | 0.2 | Sampling | **95.64** | 1.36 | 1.36 | 1.44 | 1.41 | 1.37 | 1.42 | 1.27 | 1.35 |
> |  | STAR_res_0.1 | 90.67 | **6.33** | **6.21** | 31.09 | 27.62 | 30.21 | 30.60 | 26.33 | 26.92 |
> |  | STAR_0.5 | 78.93 | 4.38 | 4.02 | **39.03** | **35.05** | **37.38** | **38.67** | **32.94** | **34.39** |
>
>
> >Section 3.2: How is the pre-determined layer chosen?
>
> We follow the layer selection procedure outlined in Appendix B.2. On a 10% validation slice, we evaluate candidate layers and rank them based on their average rank across correctness and coverage metrics. We then select the best average rank layer for full runs.
>
> >Algorithm 1, requirements: what is $\mathbb{R}_{++}$?
>
> We use $\mathbb{R}_{++}=\{x\in\mathbb{R}:x>0  \}$ for the set of strictly positive reals. We added this to the paper's Notations section.
>
>
> > Motivation for Riemannian gradient descent
>
> The idea you mention suffers a critical drawback: stepping along the Euclidean gradient is fine, but the subsequent projection can actually increase the objective. As a consequence, the classical convex projected gradient descent theory does not apply, and we lack a clean global convergence guarantee. Even local convergence is hard to achieve: the projection can distort steps enough to slow or even stall convergence unless the step size is extremely small.
>
> The Riemannian optimization framework addresses the above concerns:
> - First, we have a sufficient decrease in the objective value in each iteration step (see Proposition 5 in Appendix A).
> - Second, Riemannian gradient descent has convergence guarantees since it follows the true steepest-descent direction on the manifold with consistent steps (see Theorem 1).
>
> > Theorem 1: is a Riemannian gradient of 0 a necessary condition for a minimum of the objective function?
>
> Yes. On a Riemannian manifold, $V$ is a stationary point of a smooth function $\ell$ if and only if the Riemannian gradient $grad \ell(V)=0$ [1, Proposition 4.6].  We have added this discussion on page 6.
>
>  [1] Boumal, N. 2023. An Introduction to Optimization on Smooth Manifolds. Cambridge University Press.

---

> ### Author Response · Authors · 2025-11-26
> **Response to Reviewer D4n2 (Part 5)**
>
> > Also, in the experiments only one model is evaluated for Section 6.2.
>
>
> The following table shows the results on the LiveIdeaBench dataset across different sampling temperatures on Llama-3.2-3B-Instruct. Our method also outperforms temperature-based sampling on the Llama-3.2-3B-Instruct model across the data set LiveIdeaBench.
>
> | Temp. | Method | Originality ↑ | Feasibility ↑ | Clarity ↑ | Fluency ↑ | Flexibility ↑ | Avg. ↑ |
> |---|---|---|---|---|---|---|---|
> | 1.0 | Sampling | 6.51 | **5.34** | **5.89** | 5.41 | 5.66 | 5.76 |
> |  | STAR_0.1 | 6.57 | 5.31 | **5.89** | 5.45 | 5.68 | 5.78 |
> |  | STAR_0.5 | **6.60** | 5.32 | 5.85 | **5.60** | **5.71** | **5.82** |
> | 0.8 | Sampling | 6.47 | **5.39** | **5.92** | 5.07 | 5.58 | 5.69 |
> |  | STAR_0.1 | 6.49 | **5.39** | **5.92** | 5.11 | 5.59 | 5.70 |
> |  | STAR_0.5 | **6.56** | 5.36 | 5.91 | **5.22** | **5.62** | **5.73** |
> | 0.6 | Sampling | 6.43 | 5.42 | 5.99 | 4.59 | 5.46 | 5.58 |
> |  | STAR_0.1 | 6.45 | **5.46** | **6.01** | 4.78 | 5.57 | 5.65 |
> |  | STAR_0.5 | **6.53** | 5.42 | 5.97 | **4.81** | **5.58** | **5.66** |
> | 0.4 | Sampling | 6.41 | 5.42 | 6.00 | 4.17 | 5.35 | 5.47 |
> |  | STAR_0.1 | 6.44 | 5.46 | **6.02** | 4.23 | 5.43 | 5.52 |
> |  | STAR_0.5 | **6.50** | **5.44** | 5.99 | **4.41** | **5.48** | **5.56** |
> | 0.2 | Sampling | 6.42 | 5.45 | **6.03** | 3.27 | 5.19 | 5.27 |
> |  | STAR_0.1 | 6.46 | **5.48** | **6.03** | 3.60 | 5.27 | 5.37 |
> |  | STAR_0.5 | **6.51** | 5.46 | 6.02 | **4.04** | **5.39** | **5.48** |

---

### Author Response · Authors · 2025-12-03
**General Response**

Dear ACs and Reviewers,

We thank all the reviewers for taking the time to evaluate our paper and for providing valuable feedback to improve our work.

We are encouraged to see that

> **Reviewer ZBFD**: This work tackles the topic at its core, which is refreshing. [...] **is of interest to the community and can spark new research on this topic.**
>
> Reviewer ZBFD also actively engaged with us during the rebuttal phase, which we greatly appreciate.
>
> -----
>
> **Reviewer a1Ka**: The authors **do a good job** of motivating the problem at hand, and I believe **the problem studied is of interest to the ML community**.
>
> In the final reply, Reviewer a1Ka writes: 'I thank the authors for their response. My concerns have been addressed **and I have increased my score accordingly**.' https://openreview.net/revisions?id=t7KUSY1fz1




The main concerns raised by the reviewers centered on baselines, ablations, method clarity, and runtime. We have addressed these issues through new experiments, analyses, and revisions to the paper. The updated manuscript has been uploaded, and all changes are marked in blue. We summarize the key updates below:

- **Expanded baselines.** Following suggestions from Reviewers D4n2, a1Ka, and BKyG,  we added several new baselines, including random steering vectors, push-from-mean steering, and nucleus sampling. Overall, STAR **consistently outperforms these baselines across our benchmarks**.
- **Intervention positions.** As requested by Reviewers D4n2 and ZBFD, we added ablations to determine where we intervene (attention outputs vs. residual stream). The results show that STAR is **robust to the intervention position**.
- **Additional model families.** Following Reviewer D4n2’s suggestion, we evaluated STAR on Llama-3.2-3B-Instruct on LiveIdeaBench. STAR again **outperforms the baselines**, indicating that the gains are not specific to a single model family.
- **Sampling temperatures.** Also in response to Reviewer D4n2, we added ablations over different sampling temperatures. We find that **STAR remains robust across a range of temperatures**.
- **Statistical significance.** As requested by Reviewer ZBFD, we now report the average performance across different random seeds. STAR exhibits strong stability across different random seeds, as evidenced by small standard deviations (mostly $<$ 0.1) for most metrics, and **consistently outperforms baselines**.
- **End-to-end runtime.** Following comments from Reviewers D4n2 and ZBFD, we conducted an **end-to-end runtime comparison**. STAR **incurs only a small absolute overhead** (≤ 2 seconds/question on average) compared with standard temperature sampling.
- **Clarified methodology and writing.** In response to several comments from Reviewers D4n2, ZBFD, and a1Ka, we have **clarified the motivation** for the orthogonality constraint, explained more carefully why we adopt a Riemannian optimization viewpoint, and generally improved the exposition.

We hope this general response helps contextualize the revisions and addresses the reviewers’ main concerns. Thank you again for your careful consideration of our work.

---

### Meta-Review · Area_Chair_hSEj · 2025-12-03

**Summary:**

This work tackles diversity in LLM generations. For that, a method (STAR) based on activation steering is proposed, where the steering vector is optimized at generation time. STAR proposes to optimize such steering by, while generating multiple answers at the same time, the volume spanned by the generated tokens is maximal. Such optimization leverages Riemannian optimization theory, and a 1-step efficient alternative is proposed. Generations using STAR show more diversity and are able to preserve fluency on code generation and scientific discovery tasks.

**Reviewer Concerns:**

Reviewer D4n2

The authors successfully addressed most of Reviewer D4n2's concerns:

* Expanded baselines: New experiments were added comparing STAR against random steering, push-from-mean steering, and nucleus sampling, consistently showing STAR's superior performance.
*  Method robustness: Ablations confirmed STAR's effectiveness across various sampling temperatures and intervention positions (e.g., residual stream).
*  Model generality: STAR's performance was validated on an additional model family (Llama-3.2-3B-Instruct) and task, demonstrating its wider applicability.
*  Runtime analysis: End-to-end runtime comparisons showed that STAR incurs only a small overhead.
*  Methodology clarifications: Clarified the motivation for the orthogonality constraint, the reason for using Riemannian gradient descent, the layer selection process, and provided sources for claims.
*  Minor Issues: Agreed to rename the method to STARS.

Two concerns that still appear outstanding are:

1.  A comparison against a training-based diversity method.
2.  An evaluation against an alternative objective of maximizing variance between vectors (instead of geometric volume).

 Reviewer BKyG

All concerns were addressed:

*  Limited Baselines: Added nucleus sampling as a new baseline and mentioned comparing against two heuristic steering baselines as requested by D4n2, showing STAR's superior performance.
*  Weak Results: New ablations and comparisons address this issue.
*  Other Steering Methods: Clarified that most other steering methods are task-specific (e.g., toxic to non-toxic) and thus not directly comparable, focusing instead on diversity-oriented baselines within the autoregressive framework.
*  Diffusion-based Approaches: Clarified that STAR is designed for autoregressive LLMs, not diffusion models, making such comparisons outside the scope of their method.

Reviewer ZBFD

All of Reviewer ZBFD's concerns appear to have been addressed by the authors' rebuttal.

* Single layer intervention and residual path: Clarified STAR's flexibility, provided experiments comparing attention output vs. residual stream intervention, and explained their rationale for single-layer intervention.
* Limited experimental section (bias/fairness): Acknowledged this as important future work but clarified it was beyond the scope of the current paper. However, they did run an experiment requested by the reviewer about “rolling a dice” which relates to bias and equal distribution at generation.
*  Riemannian optimization utility: Provided a detailed explanation of the benefits of Riemannian optimization over Euclidean gradients with projections.
* Temporal correspondence in volume maximization: Explained how hidden states implicitly capture trajectory history and suggested an extension for explicit trajectory-level optimization.
*  Statistical significance: Provided results with standard deviations across multiple runs, demonstrating STAR's stability.
* Generation samples: The authors provided specific generation examples for different prompts in an appendix. Specifically, provided further insights and empirical results for the "A house" experiment which clarifies the utility of the method.
* Discussed why adaptive steering (like MERA) is not directly applicable to their current setup and explained why optimizing $\lambda$ would make the objective ill-posed.
* Provided a table comparing runtime for STAR and temperature sampling, showing minimal overhead.

Reviewer a1Ka

All of Reviewer a1Ka's addressed.

*  Algorithmic details: Clarified that steering occurs at every decoding step on a single pre-determined layer (chosen via a validation slice), modifying per-head vectors at the attention-head outputs.
*  Evaluation metrics: Provided a detailed explanation of how metrics are calculated, distinguishing between batch-level (e.g., coverage, fluency) and individual-level (e.g., correctness, originality, feasibility, clarity, then averaged) evaluations.
*  Lack of comparisons to related methods: Added a comparison to nucleus sampling and referred to other baselines (random steering, push-from-mean steering) mentioned in the general response.

The reviewer explicitly stated, "My concerns have been addressed and I have increased my score accordingly."

**Reviewer Scores:**

* Reviewer D4n2 (4 → 6): The strong rebuttal, new experiments and thorough answers make me think this reviewer would have upgraded to 6.
* Reviewer BKyG (2 → ?): This review is poor and most likely fully AI generated (my own opinion, also supported by online analysis from https://iclr.pangram.com. This review does not meet ICLR good practice, and will be discounted from the assessment.
* Reviewer ZBFD (6 → 8): This reviewer engaged during the rebuttal, proposed new experiments (done) and ackowledged the work.
* Reviewer a1Ka (4 → 6): This reviewer already mentioned upgrading the score, most likely from 4 to 6.

---

### Decision · Program_Chairs · 2026-01-26

Accept (Poster)